# Neural Optimal Transport Meets Multivariate Conformal Prediction

**Vladimir Kondratyev**
Department of ML, MBZUAI, UAE
`vladimir.kondratyev@mbzuai.ac.ae`

**Alexander Fishkov**
Department of ML, MBZUAI, UAE

**Mahmoud Hegazy**
CMAP, École polytechnique, France

**Nikita Kotelevskii**
Department of ML, MBZUAI, UAE

**Rémi Flamary**
CMAP, Ecole Polytechnique, Palaiseau, France

**Maxim Panov**
Department of ML, MBZUAI, UAE

**Eric Moulines**
CMAP, Ecole Polytechnique, Palaiseau, France
`eric.moulines@mbzuai.ac.ae`

## Abstract

Classical quantile regression does not extend naturally to multivariate responses, while existing approaches often ignore the geometry of joint distributions. We propose a framework for *vector quantile regression* (VQR) that combines neural optimal transport with amortized optimization, and apply it to multivariate conformal prediction. We then exploit the induced multivariate ranks for conformal prediction, constructing distribution-free predictive regions with finite-sample validity. Unlike coordinatewise methods, our approach adapts to the geometry of the conditional distribution, producing tighter and more informative regions. Experiments on benchmark datasets show improved coverage–efficiency trade-offs compared to baselines, highlighting the benefits of integrating neural optimal transport with conformal prediction.

## 1 Introduction

Quantile regression has long been a cornerstone for modeling heterogeneous conditional distributions in the univariate setting (Koenker & Bassett, 1978; Koenker, 2005). By estimating conditional quantiles rather than conditional means, it captures heteroscedasticity, asymmetry, and tail behavior, and the collection of conditional quantiles provides a detailed description of the conditional law of a response variable. This perspective has supported a broad range of advances in econometrics, statistics, and machine learning.

Extending these ideas to multivariate responses, however, remains challenging. Unlike the scalar case, $\mathbb{R}^d$ lacks a natural total ordering. As a result, early multivariate notions of quantiles—based on projections, spatial medians, or depth functions—provide useful summaries but generally inherit only part of the desirable scalar properties (Chaudhuri, 1996; Hallin et al., 2021).

Recent progress in optimal transport has offered a principled definition of multivariate ranks and quantiles (Chernozhukov et al., 2017; Hallin & Konen, 2024; Peyré et al., 2019). By interpreting quantiles as transport maps from a reference distribution to the law of $Y$, these approaches recover distribution-free center-outward ranks and nested quantile regions that extend univariate order statistics to high dimensions. Building on this perspective, vector quantile regression (VQR; Carlier et al., 2016; 2017) introduces conditional vector quantile functions (CVQFs), cyclically monotone maps that represent $Y$ as a transformation of latent uniform variables given covariates. CVQFs provide a rich yet tractable representation of conditional distributions, and recent work has developed practical

estimators and extensions beyond linear models (Rosenberg et al., 2023; Vedula et al., 2023b;a; Sun et al., 2022; del Barrio et al., 2025).

In parallel, conformal prediction provides a general framework for constructing predictive regions with finite-sample marginal coverage under exchangeability (Shafer & Vovk, 2008; Angelopoulos et al., 2023). Although well studied in the univariate case, multivariate extensions are less developed, and many existing constructions do not explicitly exploit the geometry of the joint conditional distribution. Existing approaches include coordinate-wise methods that treat each coordinate separately (Zhou et al., 2024; Diquigiovanni et al., 2021); reductions of the multivariate problem to one dimension via a scalar score (Dheur et al., 2025; Izbicki et al., 2019); and structured approaches that use deep generative embeddings (Dheur et al., 2025; Feldman et al., 2023). Many of these methods rely on heuristic choices of scores or shapes, with limited theoretical guidance on how to encode multivariate dependence. For an overview of recent developments in multivariate conformal regression, see (Dheur et al., 2025). Very recent work begins to bridge this gap by incorporating optimal-transport–based multivariate ranks into conformal prediction, yielding theoretically grounded multivariate prediction sets (Thurin et al., 2025; Klein et al., 2025).

In this paper, we leverage a neural optimal transport framework for learning CVQFs (Seguy et al., 2018; Makkuva et al., 2020), which allows us to estimate parametric cyclically monotone vector quantiles and multivariate ranks. Building on the resulting multivariate ranks, we use conformal prediction to produce distribution-free valid prediction regions that adapt to the geometry of conditional distributions in the multivariate setting.

We make three main contributions:

- We develop a **neural optimal transport framework** for learning **vector quantile regression (VQR)** and the associated multivariate ranks (Section 3).
- We provide a **principled integration** of **vector quantiles and multivariate ranks** into **conformal prediction**, yielding *distribution-free predictive regions* with finite-sample marginal coverage that adapt to the geometry of conditional distributions (Section 4).
- We show in numerical experiments that the resulting **conformal prediction sets** consistently *improve calibration and reduce set size* relative to coordinatewise and representation-based baselines across benchmarks (Section 6).

## 2 CONSTRUCTING MULTIVARIATE CONFIDENCE SETS

We begin by informally introducing the conditional vector quantile and rank maps, which provide a flexible representation of the conditional law of $Y$ given $X$.

**Quantiles in 1D and Confidence Sets.** Let us first consider the case of $Y \in \mathcal{Y} \subseteq \mathbb{R}$. Let $(Y, X) \sim F_{YX}$ and let $F_{Y|X}$ be the conditional distribution of $Y$ given $X$. For any $\alpha \in [0, 1]$ and any $x \in \mathcal{X}$, the quantile function $Q_{Y|X}(\cdot, x)$ returns the quantile value $Q_{Y|X}(\alpha, x) \in \mathcal{Y}$ of the conditional distribution $F_{Y|X=x}$. Knowing the quantile function is instrumental for constructing confidence sets. For example, for a given $\alpha \in (0, 1)$ one can define $\mathcal{C}_\alpha(x) = [Q_{Y|X}(\alpha/2, x), Q_{Y|X}(1 - \alpha/2, x)]$. Under standard regularity conditions (e.g., continuity of $F_{Y|X=x}$), this confidence set is conditionally valid, i.e. $\mathbb{P}(Y \in \mathcal{C}_\alpha(x) \mid X = x) = 1 - \alpha$.

The inverse map $Q_{Y|X}^{-1}$ is sometimes called a rank function: for any value $y$ it produces a value $Q_{Y|X}^{-1}(y, x) \in [0, 1]$, which can be interpreted as the rank of $y$ under the distribution $F_{Y|X=x}$. Importantly, the probability integral transform implies that the distribution of $Q_{Y|X}^{-1}(Y, X) \mid X = x$ is uniform on $[0, 1]$. In turn, the rank function gives an equivalent way to define the confidence interval, $\mathcal{C}_\alpha(x) = \{y \colon Q_{Y|X}^{-1}(y, x) \in [\alpha/2, 1 - \alpha/2]\}$. This shows two interchangeable viewpoints: one may construct $\mathcal{C}_\alpha(x)$ either from the quantile function or from the rank function.

**Multivariate Quantiles.** In the absence of a natural order on $\mathbb{R}^d$ for $d > 1$, the definition of multivariate quantiles is nontrivial. In this paper, we study multivariate quantiles based on optimal transport; see among others (Carlier et al., 2016; Hallin et al., 2021; Hallin & Konen, 2024; Peyré et al., 2019). We start by looking at a specific example, while the full exposition is given in Section 3.

Let Volume denote the Lebesgue volume normalized on $\mathcal{U} := \mathrm{B}(0,1)$ so that $\mathrm{Volume}(\mathcal{U}) = 1$. Define $r_{1-\alpha} \in \mathbb{R}_+$ such that the Euclidean ball $\mathrm{B}(0, r_{1-\alpha}) \subset \mathcal{U}$ satisfies the condition $\mathrm{Volume}(\mathrm{B}(0, r_{1-\alpha})) = 1 - \alpha$. Then, it can be shown (see Theorem 1 below) that there exists a map $Q_{Y|X}(u,x)$ and a uniform random variable $U$ over $\mathcal{U}$, independent of $X$, such that $Y = Q_{Y|X}(U, X)$ almost surely. This map is called a *vector quantile*. The corresponding inverse map $Q_{Y|X}^{-1}(y, x) \in \mathcal{U}$ is a natural analogue of the *rank function* and yields multivariate ranks that are uniform on $\mathcal{U}$ conditional on $X$.

We can directly proceed with the construction of confidence sets based on $Q_{Y|X}^{-1}$. For $x \in \mathcal{X}$, define the *pullback set*

$$\mathcal{C}_\alpha^{\mathrm{pb}}(x) := \{y \colon Q_{Y|X}^{-1}(y, x) \in \mathrm{B}(0, r_{1-\alpha})\}. \tag{1}$$

Using the key identity $Q_{Y|X}^{-1}(Q_{Y|X}(U, X), X) = U$ almost surely, we get that

$$
\begin{aligned}
&\mathbb{P}(Y \in \mathcal{C}_\alpha^{\mathrm{pb}}(X)) \\
&= \mathbb{P}_{(U,X) \sim F_U \otimes F_X}(\|Q_{Y|X}^{-1}(Q_{Y|X}(U, X), X)\| \le r_{1-\alpha}) \\
&= \mathbb{P}_{U \sim F_U}(\|U\| \le r_{1-\alpha}).
\end{aligned}
\tag{2}
$$

Hence, by the definition of $r_{1-\alpha}$, the coverage of the pullback set $\mathcal{C}_\alpha^{\mathrm{pb}}(x)$ is exactly $1 - \alpha$. Moreover, since $U$ is independent of $X$, the same argument yields $\mathbb{P}(Y \in \mathcal{C}_\alpha^{\mathrm{pb}}(x) \mid X = x) = 1 - \alpha$ for each $x \in \mathcal{X}$.

**Conformalized Confidence Sets.** In practice, we can only have access to the estimator $\widehat{Q}_{Y|X}^{-1}$ of $Q_{Y|X}^{-1}$. One can consider plug-in confidence sets constructed by replacing $Q_{Y|X}^{-1}$ with $\widehat{Q}_{Y|X}^{-1}$ in (1). However, such sets do not, in general, guarantee coverage, because typically $\widehat{Q}_{Y|X}^{-1} \neq Q_{Y|X}^{-1}$. Consequently, the coverage of $\mathcal{C}_\alpha^{\mathrm{pb}}(X)$ may be miscalibrated, motivating the use of conformal prediction (Shafer & Vovk, 2008; Romano et al., 2019; Angelopoulos et al., 2023). Conformal prediction corrects this miscalibration and provides finite-sample, distribution-free *marginal* coverage guarantees under exchangeability.

Specifically, given a calibration set $\mathcal{D}_{\mathrm{cal}} = \{(X_i, Y_i)\}_{i=1}^n$ independent of the training data, consider a score $S_i = \|\widehat{Q}_{Y|X}^{-1}(Y_i, X_i)\|, i = 1, \ldots, n$. Let $\hat{q}_{1-\alpha}$ be the empirical $(1 - \alpha)(1 + 1/n)$-quantile of $\{S_i\}_{i=1}^n$. Then split-conformal prediction constructs, for a new covariate value $X_{\mathrm{test}}$, the set

$$\hat{\mathcal{C}}_\alpha^{\mathrm{pb}}(X_{\mathrm{test}}) := \{y \in \mathcal{Y} : \|\widehat{Q}_{Y|X}^{-1}(y, X_{\mathrm{test}})\| \le \hat{q}_{1-\alpha}\}$$

such that $\mathbb{P}\{Y_{\mathrm{test}} \in \hat{\mathcal{C}}_\alpha^{\mathrm{pb}}(X_{\mathrm{test}})\} \ge 1 - \alpha$, under the assumption that $(X_1, Y_1), \ldots, (X_n, Y_n), (X_{\mathrm{test}}, Y_{\mathrm{test}})$ are exchangeable (Romano et al., 2019; Angelopoulos et al., 2023). Other choices of the score are possible, see discussion in Section 4. In what follows, we discuss various approaches to construct $\widehat{Q}_{Y|X}$ and $\widehat{Q}_{Y|X}^{-1}$ based on neural optimal transport.

## 3 VECTOR QUANTILE REGRESSION VIA OPTIMAL TRANSPORT

We recall the optimal-transport construction of conditional vector quantiles and multidimensional ranks. Our presentation follows closely the formulation of Carlier et al. (2016); Hallin et al. (2021); see also Brenier (1991); Santambrogio (2015) for background on Brenier maps and optimal transport, and Chernozhukov et al. (2017) for related OT-based multivariate ranks and quantiles.

Let $(Y, X)$ be a random vector on a complete probability space $(\Omega, \mathcal{A}, \mathbb{P})$, where $Y \in \mathbb{R}^{d_y}$ and $X \in \mathbb{R}^{d_x}$. Throughout this section, set $d = d_y$. Denote by $F_{YX}$ the joint law of $(Y, X)$, by $F_{Y|X}$ the conditional law of $Y$ given $X$, and by $F_X$ the marginal of $X$. Let $U$ be a random vector on $(\Omega, \mathcal{A}, \mathbb{P})$ with reference distribution $F_U$. We write $\mathcal{Y}, \mathcal{X}, \mathcal{U}, \mathcal{Y} \times \mathcal{X}, \mathcal{U} \times \mathcal{X}$ for the supports of $F_Y, F_X, F_U, F_{YX}, F_{UX}$, and $\mathcal{Y}_x$ for the support of $F_{Y|X=x}$. (Here, $\mathcal{Y} \times \mathcal{X}$ and $\mathcal{U} \times \mathcal{X}$ are not assumed to coincide with Cartesian products of marginal supports; they are simply notational shorthands for the supports of the corresponding joint laws.) By $U \le u$ for $U, u \in \mathbb{R}^d$ we mean a coordinatewise inequality between the vectors, that is, $U_j \le u_j$ for all coordinates $j \in \{1, \ldots, d\}$. This induces the multivariate CDF notation $F_U(u) = F_U(u_1, \ldots, u_d) = \mathbb{P}(U_1 \le u_1, \ldots, U_d \le$

$u_d) = \mathbb{P}(U \leq u)$. All norms $\|\cdot\|$ are Euclidean on $\mathbb{R}^d$. The following properties of $F_U$ and $F_{Y|X}$ are sufficient for the construction of OT-based conditional vector quantiles and ranks.

**Assumption 1.** *The reference distribution $F_U$ admits a density $f_U$ with respect to Lebesgue measure on $\mathbb{R}^d$, with convex support $\mathcal{U} \subseteq \mathbb{R}^d$.*

Typical choices for $F_U$ include the uniform distribution on $[0, 1]^d$, the Gaussian $\mathcal{N}(0, I_d)$, or any distribution on $\mathbb{R}^d$ with convex support.

**Assumption 2.** *For each $x \in \mathcal{X}$, the conditional law $F_{Y|X}(\cdot, x)$ has a density $f_{Y|X}(\cdot, x)$.*

Our aim is to construct a measurable map that transports $F_U$ to the conditional law of $Y$ given $X$, with a multivariate monotonicity property. In dimension $d > 1$, the appropriate notion of monotonicity is given by gradients of convex functions (equivalently, cyclical monotonicity for the quadratic cost), which motivates the *conditional vector quantile function* (CVQF).

**Theorem 1** (Carlier et al. (2016), Theorems 2.1 & 2.2). *Suppose Assumption 1 holds. Then:*

  (i) *For each $x \in \mathcal{X}$, there exists a measurable map $u \mapsto Q_{Y|X}(u, x)$, unique $F_U$-a.e., which is the gradient of a convex function and pushes $F_U$ forward to $F_{Y|X=x}$.*
  (ii) *Consequently, there exists $U \sim F_U$ such that $Y = Q_{Y|X}(U, X)$ a.s. with $U \mid X \sim F_U$.*
  (iii) *Additionally, if Assumption 2 holds, then there exists a measurable inverse map $Q_{Y|X}^{-1}(y, x) \in \mathcal{U}$ such that $Q_{Y|X}^{-1}(Q_{Y|X}(u, x), x) = u$ for $F_U$-a.e. $u$, and $\mathbb{P}(Q_{Y|X}^{-1}(Y, X) \leq u \mid X = x) = F_U(u)$.*

The map $y \mapsto Q_{Y|X}^{-1}(y, x)$ is the *conditional vector rank*. When $d = 1$ and $F_U$ is the uniform distribution on $[0, 1]$, this rank coincides (a.e.) with the conditional CDF. For $d > 1$, there is no analogous identification with a coordinatewise conditional CDF, and the rank is intrinsically vector-valued (Hallin et al., 2021; Hallin & Konen, 2024; del Barrio et al., 2025).

Finally, the following moment condition ensures that the quadratic OT objective is well-defined and that the duality-based characterization below applies.

**Assumption 3.** *$Y$ and $U$ have finite second moments: $\mathbb{E}[\|Y\|^2] < \infty$ and $\mathbb{E}[\|U\|^2] < \infty$.*

Under Assumptions 1 and 3, the CVQF admits the following optimal-transport characterization:

$$\min_V \mathbb{E}[\|Y - V\|^2] \quad \text{s.t. } V \mid X \sim F_U,$$

The dual program is

$$\min_{\psi, \varphi} \mathbb{E}[\varphi(V, X)] + \mathbb{E}[\psi(Y, X)] \quad \text{s.t.} \quad \varphi(u, x) + \psi(y, x) \geq u^\top y, \tag{3}$$

where $V$ is any vector such that $V \mid X \sim F_U$, and the inequality is required for all $x \in \mathcal{X}$, $u \in \mathcal{U}$, and $y \in \mathcal{Y}_x$. The following properties of the solution to (3) hold.

**Theorem 2** (Carlier et al. (2016), Theorem 2.3). *Suppose Assumptions 1–3 hold. Then,*

  (i) *There exist potentials $\varphi(u, x)$ and $\psi(y, x) = \varphi^*(y, x)$ solving (3), where for each $x$, $u \mapsto \varphi(u, x)$ and $y \mapsto \psi(y, x)$ are convex and Legendre conjugates:*

$$\begin{aligned} \varphi(u, x) &= \sup_{y \in \mathcal{Y}} \{u^\top y - \psi(y, x)\}, \\ \psi(y, x) &= \varphi^*(y, x) = \sup_{u \in \mathcal{U}} \{u^\top y - \varphi(u, x)\}. \end{aligned} \tag{4}$$

  (ii) *The conditional vector quantile is $Q_{Y|X}(u, x) = \nabla_u \varphi(u, x)$ for $F_U$-a.e. $u$.*
  (iii) *The conditional vector rank is $Q_{Y|X}^{-1}(y, x) = \nabla_y \psi(y, x)$ for $F_{Y|X}(\cdot, x)$-a.e. $y$.*
  (iv) *These maps are inverses: for each $x$, $\nabla_y \psi(\nabla_u \varphi(u, x), x) = u$, $\nabla_u \varphi(\nabla_y \psi(y, x), x) = y$, for $F_U$-a.e. $u$ and $F_{Y|X}(\cdot, x)$-a.e. $y$.*

This characterization reduces the construction of $Q_{Y|X}$ and $Q_{Y|X}^{-1}$ to finding convex dual potentials solving (3), and it underlies computational approaches based on convex optimization and discretized optimal transport.

We now describe a practical approach for learning continuous Neural VQR models. We first rewrite the optimization problem using a single convex potential via the conditional $c$-transform. We then discuss implementable parameterizations with Partially Input Convex Neural Networks (PICNNs; Amos et al., 2017; Bunne et al., 2022) and two standard strategies to reduce training cost: amortized optimization and an entropic variant.

**Neural parameterization and semi-dual formulation.** First, following Taghvaei & Jalali (2019); Makkuva et al. (2020); Amos (2023), we reformulate the Monge-Kantorovich dual problem (3) as an optimization over a parametric family of potentials $\varphi_\theta$ with parameters $\theta$. We enforce that $\varphi_\theta$ is convex in its first argument, so that the Fenchel-Legendre conjugacy in (4) (the $c$-transform) yields a unique conjugate potential.

For each $x \in \mathcal{X}$ we introduce the conjugate of a pointwise potential $\varphi_\theta(\cdot, x) \colon \mathcal{U} \to \mathcal{Y}$ as

$$J_{\varphi_\theta(\cdot,x)}(u, y) = u^T y - \varphi_\theta(u, x), \tag{5}$$

$$\varphi_\theta^*(y, x) = J_{\varphi_\theta(\cdot,x)}\big(\check{u}_{\varphi_\theta(\cdot,x)}(y), y\big), \tag{6}$$

$$\check{u}_{\varphi_\theta(\cdot,x)}(y) = \arg\max_{u \in \mathcal{U}} J_{\varphi_\theta(\cdot,x)}(u, y). \tag{7}$$

With these notations, the problem (3) can be reformulated as the minimization of $\mathcal{V}(\theta)$, defined as

$$\mathcal{V}(\theta) := \mathbb{E}_{(U,X)\sim F_U \otimes F_X} \left[\varphi_\theta(U, X)\right] \quad + \mathbb{E}_{(Y,X)\sim F_{YX}} \left[\varphi_\theta^*(Y, X)\right]. \tag{8}$$

Here, $F_U \otimes F_X$ denotes the product measure of $F_U$ and $F_X$, corresponding to independent sampling of $U \sim F_U$ and $X \sim F_X$. The optimal parameter can then be found by gradient steps on $\mathcal{V}(\theta)$. By Danskin's theorem (Danskin, 1967), differentiating through the conjugate only requires derivatives of $\varphi_\theta$ and yields

$$\nabla_\theta \mathcal{V}(\theta) = \mathbb{E}_{(U,X)\sim F_U \otimes F_X}[\nabla_\theta \varphi_\theta(U, X)] - \mathbb{E}_{(Y,X)\sim F_{YX}}[\nabla_\theta \varphi_\theta(u, X)|_{u=\check{u}_{\varphi_\theta(\cdot,X)}(Y)}]. \tag{9}$$

*Remark* 1. Above we focus on the dual potential $\varphi_\theta(\cdot, x)$, which is linked to $F_U$, while its conjugate $\varphi_\theta^*(\cdot, x)$ is linked to $F_{Y|X}(\cdot \mid X = x)$. Due to the symmetry of (3), one may instead parameterize $\psi_\theta(\cdot, x)$; in our experiments we consider both choices.

**Neural Quantile Regression with PICNNs (C-NQR).** Convexity of $\varphi_\theta(\cdot, x)$ with respect to the first argument can be achieved (Bunne et al., 2022) using PICNNs (Amos et al., 2017). The main computational challenge in (8) is that the conjugate $\varphi_\theta^*(\cdot, x)$ must be computed for each $x$ in the mini-batch. As a first strategy, we compute the argmax in (7) exactly with an L-BFGS solver (Liu & Nocedal, 1989), and update the PICNN parameters $\theta$ with stochastic gradient descent; see Algorithm 1 in Appendix E.6. We refer to this method as conditional neural quantile regression (C-NQR). We distinguish C-NQR$_U$ and C-NQR$_Y$ depending on whether we estimate $\varphi_\theta$ or, equivalently by symmetry, $\psi_\theta$. This approach is straightforward but can be expensive because it requires repeated inner optimization, especially for large mini-batches or high-dimensional data.

**Amortized Neural Quantile Regression (AC-NQR).** To reduce the cost of repeatedly solving (7), we use amortized optimization, following Amos (2023). We introduce an amortization model $\tilde{u}_\vartheta(y, x)$ parameterized by $\vartheta$ that predicts an approximate maximizer:

$$\tilde{u}_\vartheta(y, x) \simeq \check{u}_{\varphi_\theta(\cdot,x)}(y).$$

We focus on amortization architectures based on PICNNs, as in (Makkuva et al., 2020; Korotin et al., 2019). The amortization model is trained jointly with $\varphi_\theta$ using a quadratic loss encouraging $\tilde{u}_\vartheta(y, x)$ to match $\check{u}_{\varphi_\theta(\cdot,x)}(y)$; see Algorithm 2 in Appendix E.6. We follow a two-time-scale training scheme (Konda & Tsitsiklis, 2004; Borkar, 2008), where the amortization model is updated on a faster timescale than $\varphi_\theta$. We refer to this method as amortized conditional neural quantile regression (AC-NQR), with AC-NQR$_U$ and AC-NQR$_Y$ defined analogously to C-NQR.

**Entropic regularized Neural Quantile Regression (EC-NQR).** The two approaches above still require solving a convex optimization problem to compute (approximate) conjugates, which can become costly in high dimensions. An alternative is to use entropic regularization, which enables stochastic gradient solvers that scale better (Cuturi, 2013; Genevay et al., 2016), but introduces bias that can distort the geometry of quantile maps (Rosenberg et al., 2023). Neural parameterizations

of entropic OT dual potentials were studied in the non-conditional case by Seguy et al. (2018); we extend this idea to the conditional Neural VQR setting.

Concretely, we add an entropic regularization term to the primal OT problem. This smooths the objective and yields a closed-form surrogate for the conjugate (the argmax in (7) becomes a softmax). We refer to this method as entropic conditional neural quantile regression (EC-NQR). The resulting objective replaces inner conjugate optimization by an expectation that can be approximated by sampling; see Algorithm 3 in Appendix E.6. Additional details are provided in Appendix C and Appendix A.

*Remark* 2. By parameterizing the dual potential as a convex neural network, we restrict the model class. This raises the question of whether a more flexible non-convex generative model, such as a normalizing flow that parameterizes a bijection, could replace the convex potential while still defining a meaningful quantile operator. We show that this is not the case and provide a counterexample in Appendix G. In particular, cyclic monotonicity of the transformation is essential for a statistically meaningful notion of multivariate rank.

## 4    CONFORMAL PREDICTION WITH OT NEURAL MAPS

In this section, we demonstrate the use of our neural OT framework in constructing intrinsically adaptive confidence sets with CP. The key idea is to exploit multivariate quantile and rank maps learned by conditional neural OT as a building block for defining conformity scores and constructing calibrated prediction regions. Let $(Y, X) \sim F_{YX}$ and $\alpha \in (0, 1)$ and denote by $\widehat{Q}_{Y|X}^{-1}$ a proxy for the true associated vector rank function $Q_{Y|X}^{-1}$ as in Theorem 1.

**Generalizing conformalized quantile regression.** In the univariate case, conformalized quantile regression (CQR; Romano et al., 2019) replaces a nominal quantile with the empirical $(1 - \alpha)$-quantile of residuals, ensuring distribution-free, finite-sample coverage. The same principle extends to the plug-in pullback set in (1). Define conformity scores:

$$S_i = \|\widehat{Q}_{Y|X}^{-1}(Y_i, X_i)\|, \qquad (Y_i, X_i) \in \mathcal{D}_{\mathrm{cal}}.$$

Let $S_{(1)} \leq \cdots \leq S_{(n)}$ denote the order statistics, set $k = \lceil (n + 1)(1 - \alpha) \rceil$, and $\rho_{1-\alpha} = S_{(k)}$. The conformal set

$$\hat{\mathcal{C}}_\alpha^{\mathrm{pb}}(x) = \{y \colon \widehat{Q}_{Y|X}^{-1}(y, x) \in \mathrm{B}(0, \rho_{1-\alpha})\}$$

then guarantees $\mathbb{P}_{(Y,X) \sim F_{YX}}\left(Y \in \hat{\mathcal{C}}_\alpha^{\mathrm{pb}}(X)\right) \geq 1 - \alpha$. We now show that this construction of confidence sets is optimal when the Jacobian of the inverse transport admits a radial structure.

**Theorem 3** (Volume–optimality of pullback balls under radiality)**.** *Fix $x \in \mathcal{X}$ and reference distribution $F_U(u) = \phi(\|u\|)$ for a strictly decreasing $\phi \colon [0, \infty) \to (0, \infty)$ on $\mathcal{U}$, under the assumptions of Theorem 1, let $Q_{Y|X}$ and $Q_{Y|X}^{-1}$ be the vector quantile and multivariate rank functions. Assume that there exists $j_x$ such that for all $y$ in the support of $F_{Y|X}$, it holds*

$$\det\left[\nabla_y Q_{Y|X}^{-1}(y, x)\right] = j_x\left(\|Q_{Y|X}^{-1}(y, x)\|\right),$$

*and the function $r \mapsto \phi(r) j_x(r)$ is strictly decreasing. Let $r_\alpha > 0$ be the unique radius satisfying $\mu(B_{r_\alpha}) = 1 - \alpha$, where $\mu$ is the law corresponding to $F_U$ and $B_r = \{u \colon \|u\| \leq r\}$. Define the pullback ball $\mathcal{C}_\alpha^{\mathrm{pb}}(x) := \left\{y \colon \|Q_{Y|X}^{-1}(y, x)\| \leq r_\alpha\right\}$. Then, $\mathcal{C}_\alpha^{\mathrm{pb}}(x)$ minimizes Lebesgue volume among all sets with $x$-conditional coverage of at least $1 - \alpha$, i.e., for every measurable $A \subset \mathcal{Y}_x$ with $\mathbb{P}\{Y \in A \mid X = x\} \geq 1 - \alpha$, $\mathrm{Vol}\left(\mathcal{C}_\alpha^{\mathrm{pb}}(x)\right) \leq \mathrm{Vol}(A)$.*

Equivalently, Theorem 3 shows that $\mathcal{C}_\alpha^{\mathrm{pb}}(x)$ is the highest probability density (HPD) region for $Y \mid X = x$ at level $1 - \alpha$. A noteworthy specialization, where the assumptions of Theorem 3 are met, is the *elliptical* case (including Gaussian) with $F_{Y|X}$ and $F_U$ belonging to the same elliptical family. We defer the proof and additional details to Appendix F.

**Re-ranked pullback sets.** This construction is effective only if the scores $S_i$ capture isotropic structure. Indeed, $\hat{\mathcal{C}}_\alpha^{\mathrm{pb}}(x)$ is the preimage of a centered Euclidean ball in $\mathcal{U}$, implicitly assuming that the conditional distribution of $U = \widehat{Q}_{Y|X}^{-1}(Y, X)$ is radially symmetric. When $\widehat{Q}_{Y|X}^{-1}$ is misspecified,

however, the ranks may be anisotropic, and Euclidean radii become unreliable. We note that the vector ranks $\{U_i = \widehat{Q}_{Y|X}^{-1}(Y_i, X_i)\}_{i=1}^n$ can themselves be interpreted as multivariate score functions and as such be combined with the OT-CP approach of Thurin et al. (2025), which is designed to conformalize multivariate score functions. In particular, let $\mathbf{R}\colon \mathcal{U} \to \mathcal{U}$ be a reranking approach, designed to correct deviations from reference distribution $F_U$. Reranking induces new scores $S_i^{rpb} = \|\mathbf{R}(U_i)\|$. As before let $S_{(1)}^{rpb} \leq \cdots \leq S_{(n)}^{rpb}$ denote the order statistics, set $k = \lceil (n+1)(1-\alpha) \rceil$, and $\rho_{1-\alpha}^{uni} = S_{(k)}^{rpb}$. Then, the conformalization step applied to the adjusted scores yields a calibrated radius and the prediction set

$$\hat{\mathcal{C}}_\alpha^{\mathrm{rpb}}(x) = \big\{ y\colon \mathbf{R}\big(\widehat{Q}_{Y|X}^{-1}(y, x)\big) \in \widehat{\mathcal{Q}}(1-\alpha) \big\},$$

where $\widehat{\mathcal{Q}}(1-\alpha) = \{u\colon \|\mathbf{R}(u)\| \leq \rho_{1-\alpha}^{uni}\}$. See additional implementation details in Appendix E.7.

*Remark* 3. For completeness, we also consider a complementary construction that leverages the OT quantile and rank maps to estimate the conditional density via the change of variables formula. Using the estimated density as a conformal score, this approach yields valid regions and can capture disconnected geometry when $F_{Y|X=x}$ is multimodal, e.g. Gaussian mixture. We provide additional details and a brief discussion in Appendix F.

## 5 RELATED WORK

**Multivariate Quantiles.** Scalar quantile regression estimates conditional quantiles of $Y \in \mathbb{R}$ given $X \in \mathbb{R}^p$, typically using linear-in-features models trained via the check loss (Koenker & Bassett, 1978; Koenker, 2005). Extending this framework to the multivariate setting is challenging due to the absence of a natural order. Early generalizations include spatial quantiles (Chaudhuri, 1996) and depth-based quantiles (Hallin et al., 2021), though these lack a transport-map interpretation. From a measure-transportation perspective, multivariate quantiles are defined as optimal transport (OT) maps from a reference distribution, inducing center-outward ranks and quantile regions with desirable geometric and statistical properties (Chernozhukov et al., 2017; Hallin et al., 2021; Hallin & Konen, 2024; del Barrio et al., 2025). The conditional vector quantile function (CVQF) of Carlier et al. (2016) models the quantile map as affine in $X$ and estimates it via variational OT (Carlier et al., 2017), with subsequent extensions to nonlinear embeddings (Rosenberg et al., 2023) and nonparametric rank estimation (del Barrio et al., 2025). However, prior efforts to construct continuous VQR models (Vedula et al., 2023b;a; Sun et al., 2022) have retained the affine-in-$X$-embeddings assumption , thereby constraining the expressivity of the learned quantile maps. Moreover, these methods do not provide a principled way to estimate continuous rank functions, instead producing discrete, pointwise solutions. Finally, while scalable solvers based on entropic regularization have been developed, to the best of our knowledge, no previous work has scaled VQR using neural optimal transport, as proposed in this paper.

**Neural Optimal Transport.** High-dimensional OT is challenging due to the nonlinear dual formulation. One approach employs entropic regularization, enabling Sinkhorn iterations and stochastic gradient solvers (Cuturi, 2013; Genevay et al., 2016; Seguy et al., 2018; Carlier et al., 2022), which scale well but introduce bias that may distort the geometry of quantile maps (Rosenberg et al., 2023). A second approach parameterizes convex potentials with input-convex neural networks (IC-NNs; Amos et al., 2017; Makkuva et al., 2020; Amos, 2023), ensuring monotonicity and invertibility of the learned map. Conditional potentials (and Monge maps) have been proposed in Bunne et al. (2022) but are learned in a supervised way (from examples of conditioning and target distributions) and never from a unique joint sampling using the framework of Carlier et al. (2017) as proposed in our work.

**Multivariate Conformal Prediction.** Conformal prediction (CP) constructs distribution-free predictive sets with coverage guarantees. In the scalar case, conformalized quantile regression (CQR; Shafer & Vovk, 2008; Romano et al., 2019; Angelopoulos et al., 2023) adjusts quantile estimates to achieve valid intervals. For multivariate responses, naive coordinatewise CP yields conservative rectangles; scalarized scores via norms or maxima produce balls or boxes, but remain restrictive. Structured approaches include deep generative embeddings (Feldman et al., 2023) and copula calibrations (Messoudi et al., 2021). Dheur et al. (2025) propose conformity scores based on generative

| Dataset | EC-NQR | C-NQR$_U$ | C-NQR$_Y$ | AC-NQR$_U$ | AC-NQR$_Y$ | CPF | FN-VQR | VQR | CPQ | CVQR |
|---|---|---|---|---|---|---|---|---|---|---|
| *Star* | 0.197 | 0.184 | 0.184 | **0.182** | 0.197 | 0.247 | 0.271 | 0.270 | 0.274 | 0.443 |
| *Glasses* | **0.748** | 0.785 | 0.812 | 0.771 | 0.810 | 1.687 | 2.017 | 1.964 | 0.931 | 1.170 |
| *Banana* | 0.111 | 0.072 | 0.073 | 0.073 | 0.072 | **0.069** | 0.398 | 0.389 | 0.237 | 0.401 |
| *Convex Star* | 0.200 | **0.182** | 0.184 | 0.182 | 0.191 | 0.191 | 0.262 | 0.261 | 0.252 | 0.448 |
| *Convex Glasses* | **0.650** | 0.656 | 0.668 | 0.657 | 0.689 | 0.760 | 1.954 | 1.961 | 0.793 | 0.953 |
| *Convex Banana* | 0.103 | 0.101 | 0.071 | 0.070 | 0.070 | **0.069** | 0.397 | 0.392 | 0.211 | 0.425 |
| Training time | 10.99 sec. | 15.08 sec. | 15.09 sec. | 8.89 sec. | 12.63 sec. | - | - | - | - | - |
| Inference time | 1.71 sec. | 1.21 sec. | 1.76 sec. | 1.12 sec. | 1.34 sec. | - | - | - | - | - |

Table 1: S-W2 between the ground-truth and empirical distributions. We report the per-epoch training time, averaged across all datasets, and the average inference time required to compute the inverse $c$-transform for 8192 elements. The best result is shown in bold, and the second best is underlined.

models or aggregated $p$-values.
In contrast with previously developed methods based on generative modeling (Feldman et al., 2023; Dheur et al., 2025). We propose an explicit generative model, that approximates a canonical quantile function (Hallin et al., 2021).

Very recently, the use of OT-based ranks and quantiles has been exploited in conformal prediction. In two concurrent works, Thurin et al. (2025) define conformity scores from discrete OT ranks, while Klein et al. (2025) leverage the same construction albeit with entropy regularized discrete OT. By construction, these two approaches are not adaptive, i.e. the size of the conformal set does not depend on $X$. Thurin et al. (2025) did propose an adaptive variant based on conditional density estimation with $k-$nearest neighbors. Our direct learning of neural VQR does not depend on conditional density estimation and we expect it to perform better in high dimensionality settings.

## 6 NUMERICAL EXPERIMENTS

### 6.1 NEURAL OPTIMAL TRANSPORT

To evaluate the generative performance of our models, we conduct extensive experiments. Whenever a ground-truth operator is required, we parametrize the datasets using a convex potential function, see Appendix H.2 for details. EC-NQR, C-NQR$_U$, C-NQR$_Y$, AC-NQR$_U$, AC-NQR$_Y$ are the methods described in Section 3. We measure the generative performance against FN-VQR (Rosenberg et al., 2023), VQR (Carlier et al., 2017), CPF (Huang et al., 2021), CVQR (Vedula et al., 2023a) and CPQ (Sun et al., 2022).

**Metrics.** We employ three categories of metrics: (i) Wasserstein-2 (W2) and Sliced Wasserstein-2 (S-W2) distances; (ii) Kernel Density Estimate $\ell_1$ distance (KDE-L1) and Kernel Density Estimate Kullback–Leibler divergence (KDE-KL); and (iii) Percentage of Unexplained Variance (L2-UV; Korotin et al., 2021). Metrics in (i) and (ii) quantify the fidelity of the learned distribution to the target density, while (iii) assesses the extent to which the ground-truth quantile is recovered. Additional implementation details are provided in Appendix H.1.

**Datasets.** We evaluate on three synthetic datasets originally introduced in the discrete setting of quantile regression (Rosenberg et al., 2023): *Banana*, a parabola-shaped distribution whose curvature varies with a latent random variable; *Star*, a three-pointed star whose orientation is governed by a latent variable; and *Glasses*, a bimodal distribution with sinusoidally shifting modes. We denote convex-potential counterparts as *Convex Banana*, *Convex Star*, and *Convex Glasses*.

**Results.** The experiments in Table 1 evaluate the generative capability of the proposed neural quantile regression framework.

We evaluate the ability of our method to recover the underlying quantile mapping. We report the L2-UV metric in Table 2 evaluated on Convex Banana, Convex Star, and Convex Glasses. The results in Table 2 show that the proposed models achieve high precision in reconstructing the true quantile operator.

The experiments confirm the overall performance of the proposed quantile construction, making it a suitable tool for subsequent conformal prediction procedures. Extended experimental results

| Function | Dataset | EC-NQR | C-NQR$_U$ | C-NQR$_Y$ | AC-NQR$_U$ | AC-NQR$_Y$ | CPF |
|---|---|---|---|---|---|---|---|
| | *Convex Star* | 1.331 | **0.211** | 0.286 | 0.264 | 0.425 | 0.447 |
| $Q_{Y|X}^{-1}$ | *Convex Glasses* | 0.348 | 0.332 | **0.068** | 0.203 | 0.109 | 2.268 |
| | *Convex Banana* | 3.942 | 3.784 | 0.212 | **0.106** | 0.206 | 9.479 |
| | Convex Star | 2.746 | 0.360 | 0.351 | 0.393 | 0.525 | **0.267** |
| $Q_{U|X}$ | *Convex Glasses* | 0.678 | **0.535** | 0.732 | 0.985 | 1.096 | 1.726 |
| | *Convex Banana* | 9.400 | 7.665 | 0.660 | **0.545** | 0.569 | 16.537 |

Table 2: L2-UV of the true quantile function, evaluated on generative processes parameterized by convex potential networks. The best result is shown in **bold**, and the second best is underlined.

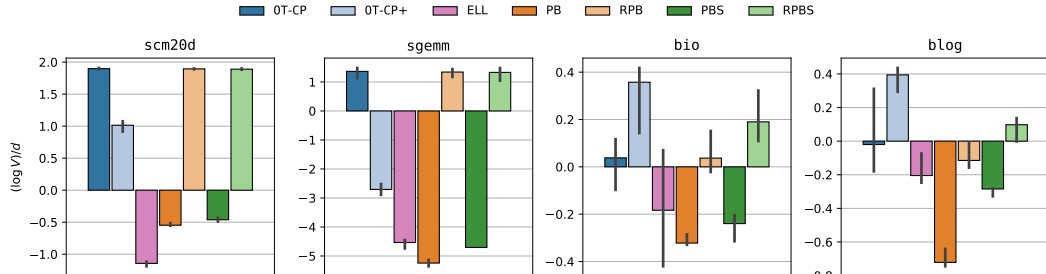

Figure 1: Log-volume of the prediction sets, normalized by $d_y$, of the resulting prediction sets for different methods. Results averaged over 10 independent data splits. Nominal miscoverage level $\alpha = 0.1$

including experiments on scalability, visualisation of learned generative model and full list of metrics on studied datasets are available in Appendix H.3.

## 6.2 CONFORMAL PREDICTION EXPERIMENTS

We further evaluate conformal prediction by constructing prediction sets on real-world datasets using the methods described in Section 4. Extended results are presented in Appendix H.

**Methods.** We use AC-NQR$_U$ as the base model to implement our two conformal methods: PB($\hat{\mathcal{C}}^{\mathrm{pb}}$) and RPB ($\hat{\mathcal{C}}^{\mathrm{rpb}}$). In addition to fitting our vector quantile regression model directly on $y$, we also fit both methods on signed residuals $s = y - \hat{f}(x)$, where $\hat{f}$ is a Random Forest regressor fit on 25% of the training data (PBS and RPBS in the plots). We consider OT-CP and OT-CP+ (Thurin et al., 2025), as well as the local Ellipsoid method (Messoudi et al., 2022) for comparison.

**Metrics.** We evaluate performance using three metrics: (i) marginal coverage, (ii) worst-slab coverage (Cauchois et al., 2021), and (iii) average prediction set volume.

**Datasets.** We evaluate on standard multi-target regression benchmarks used in previous work on uncertainty estimation (Plassier et al., 2025; Dheur et al., 2025): scm20d, sgemm, blog, and bio. For the single-target datasets blog and bio, we follow Feldman et al. (2023) and add one of the features as a second output. The resulting response dimensions are 16, 4, 2 and 2, respectively. We use preprocessing procedure of (Grinsztajn et al., 2022).

**Discussion.** PB and PBS provide competitive conditional coverage and smallest volume at the same time on three out of four datasets. The re-ranking step of RPB and RPBS allows to achieve a slightly sharper conditional coverage, but the increase in prediction sets volume make it a questionable trade-off. Overall, it shows that for our quantile regression models the split conformal calibration is enough. Our methods provide a scalable training enable building competitive conformal predictors.

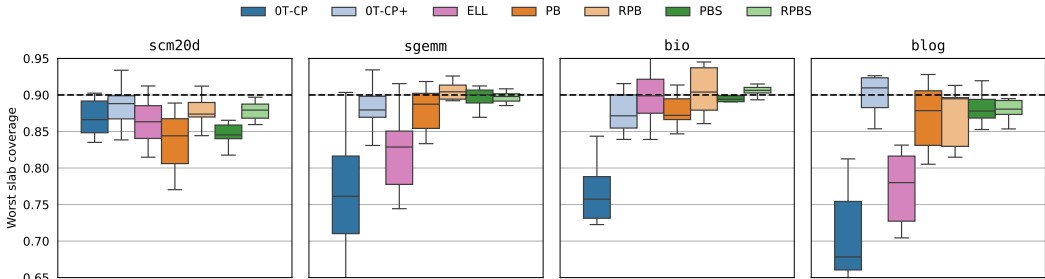

Figure 2: Worst-slab coverage for different methods. Results averaged over 10 independent data splits. Nominal miscoverage level $\alpha = 0.1$

## 7 CONCLUSION

We introduced a framework for multivariate conformal prediction based on convex potentials and optimal transport. Our approach leverages neural quantile regression with input convex neural network parameterization to construct valid and efficient prediction sets. Through experiments on synthetic benchmarks and real-world multi-target regression datasets, we demonstrated strong performance in terms of coverage and set size, while maintaining scalability in higher dimensions. Comparisons with existing baselines further highlight the robustness and flexibility of our method. Future work includes extending the framework to broader classes of generative models and exploring tighter efficiency guarantees in high-dimensional regimes.

## Usage of Large Language Models (LLMs)

LLMs were used as a general-purpose assistive tool during the preparation of this paper. Their usage fell into two categories: (i) for writing assistance, they helped improve clarity and readability of certain passages through language refinement and (ii) for coding assistance, where they provided support with code completion and debugging. LLMs were not used for research ideation, experimental design, theoretical development, or analysis of results. All substantive contributions, including the conception of ideas, methodology, and experiments, were made by the authors.

## Reproducibility Statement

We provide the full code to reproduce our experiments as supplementary material and will release it publicly upon acceptance. All experiments were conducted on publicly available datasets or datasets we created ourselves, which will be released alongside the code. We ran experiments with multiple seeds, if applicable, and report summary statistics.

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

# A  EXTENDED STATE OF THE ART

**From scalar to vector quantiles.**  Classical quantile regression (QR) estimates conditional quantiles of a scalar response $Y \in \mathbb{R}$ given features $X \in \mathbb{R}^p$, providing a flexible alternative to least squares for modeling heterogeneous effects (Koenker & Bassett, 1978; Koenker, 2005). For a quantile level $u \in (0, 1)$ and feature map $\varphi(x)$, a standard linear QR model assumes $Q_{Y|X}(u \mid x) = \beta(u)^\top \varphi(x)$, with $\beta(u)$ obtained by minimizing the check-loss. While univariate QR theory is well-developed, extending these notions to a multivariate response $Y \in \mathbb{R}^d$ is challenging due to the lack of a natural total order on $\mathbb{R}^d$. Many generalizations have been proposed, including *directional* or *projection* quantiles (reducing to scalar quantiles along particular directions) and *geometric* or *spatial* quantiles (e.g. Chaudhuri, 1996), as well as definitions based on statistical depth (e.g. Tukey's halfspace depth) that yield central regions viewed as multivariate "quantiles." However, these early notions only partially extend scalar quantile properties and generally do not yield a unique quantile *mapping* for $Y$. A recent breakthrough comes from the *measure transportation* perspective, which defines multivariate quantiles as the optimal transport map pushing a reference distribution (usually the spherical uniform, or uniform on the unit hypercube) onto the distribution of $Y$. This approach rooted in Brenier's theorem on monotone optimal transport maps (Brenier, 1991) yields well-behaved center-outward distribution and quantile functions that assign each point in $\mathbb{R}^d$ a multivariate rank and sign with distribution-free properties. The resulting quantile regions are nested, have correct probability contents, and enjoy equivariance properties generalizing the one-dimensional case. These concepts, introduced by Chernozhukov et al. (2017) and further developed by Hallin et al. (2021), provide a rigorous multivariate analog of the quantile function; see (Hallin & Šiman, 2017) for a survey of earlier definitions. Recent work continues to refine this framework: Hallin & Konen (2024) compare geometric vs. transport-based contours, and nonparametric multiple-output quantile regression methods based on center-outward ranks have been proposed (del Barrio et al., 2025).

**Vector quantile regression (VQR).**  Building on optimal transport ideas, Carlier et al. (2016) introduced the *conditional vector quantile function* (CVQF) $Q_{Y|X}(u, x)$ for $Y \in \mathbb{R}^d$. This is defined as a (a.e.) *monotone* map in $u$ — specifically, the gradient of a convex function in the $u$ argument — such that for each fixed $x$, $Q_{Y|X}(\cdot, x)$ pushes the uniform distribution on $[0, 1]^d$ forward to the conditional distribution $Y \mid X = x$. In analogy to the scalar case, one can represent $Y$ as $Y = Q_{Y|X}(U, X)$ with $U \sim \mathrm{Unif}([0, 1]^d)$ independent of $X$. This generalizes the scalar quantile relationship $Y = Q_{Y|X}(U, X)$ for $U \sim \mathrm{Unif}(0, 1)$, providing a powerful characterization of the conditional law of $Y$ by a deterministic map on the unit hypercube. In practice, VQR imposes a parametric form on the CVQF; for example, the original proposal assumes an affine structure $Q_{Y|X}(u, x) = \alpha(u) + B(u)^\top x$ (with $\alpha(u) \in \mathbb{R}^d$ and $B(u) \in \mathbb{R}^{d \times p}$) and estimates these functions by solving a large-scale optimal transport problem under empirical data constraints. The solution can be found via a convex dual formulation analogous to Koenker's linear program, ensuring the fitted $Q_{Y|X}$ is monotone in $u$ (i.e. cyclically monotonic) (Carlier et al., 2016; 2017). This yielded the first notion of "quantile regression for vectors," including strong theoretical guarantees on consistency and uniqueness under appropriate conditions. Since then, a number of extensions have been proposed: Rosenberg et al. (2023) introduce a fast nonlinear VQR model (e.g. using kernel or neural network features) while preserving monotonicity, Vedula et al. (2023b) develop a continuous VQR formulation that treats $u$ in a continuum (rather than on a finite grid of quantile levels), and fully nonparametric approaches based on center-outward quantile functions have appeared (del Barrio et al., 2025). Each of these methods seeks to balance flexibility and computational tractability while maintaining the defining property that $u \mapsto Q_{Y|X}(u, x)$ is a gradient map (hence invertible and order-preserving in the multivariate sense).

**Computation.**  Implementing VQR at scale poses significant challenges. The initial algorithms of Carlier et al. (2016) and Carlier et al. (2017) relied on discretizing the unit hypercube $[0, 1]^d$ (for a set of representative $u$ values) and solving a large linear program, which becomes computationally expensive as $d$ or the number of quantile levels grows. Two recent strategies have substantially improved the scalability of VQR. First, Carlier et al. (2022) propose an *entropic regularization* of the OT problem, which smooths the objective and leads to a differentiable dual formulation. By applying Sinkhorn-type iterations or gradient-based optimization on the regularized dual, one can efficiently approximate the CVQF without solving a huge LP, even for continuous $u$ spaces.

This regularized VQR approach yields an accuracy–computational cost trade-off controlled by the entropy penalty, and it has demonstrated orders-of-magnitude speedups on moderate-dimensional problems.

The second approach uses deep learning to represent the convex potential of the CVQF: Makkuva et al. (2020) propose to parameterize $Q_{Y|X}(u, x)$ as $\nabla_u \psi(u, x)$ where $\psi$ is an input-convex neural network in $u$. By training $\psi$ on data (using a suitable loss derived from the OT characterization), one obtains a VQR model that can handle high-dimensional $X$ and $Y$ and large sample sizes. This method, part of a broader trend of using neural networks for OT map estimation, sidesteps explicit discretization by leveraging automatic differentiation to enforce convexity in $u$. Both the entropic-OT and ICNN-based approaches have made it feasible to learn multivariate quantile functions on modern datasets, a task once thought impractical (Huang et al., 2021; Kan et al., 2022). For additional background on scalable optimal transport techniques that underlie these advances, see (Peyré et al., 2019).

**Conformal prediction.** Conformal prediction (CP) provides distribution-free predictive uncertainty sets with finite-sample coverage guarantees. In the scalar $Y$ case, it is common to combine quantile regression with conformal calibration. For example, conformalized quantile regression (CQR) uses holdout data to adjust the initially estimated interval $[\hat{Q}_{Y|X}(\alpha/2 \mid x), \hat{Q}_{Y|X}(1 - \alpha/2 \mid x)]$ so that it achieves the target coverage $1 - \alpha$ marginally. CQR and related methods yield prediction intervals that are adaptive (varying with $x$) while retaining rigorous coverage guarantees (Romano et al., 2019; Angelopoulos et al., 2023). However, extending CP to multivariate outputs has proven more complex. Naively applying conformal methods to each component of $Y = (Y_1, \ldots, Y_d)$ and taking a Cartesian product of marginal intervals yields a rectangular prediction region that is valid but often overly conservative (covering significantly more than $1 - \alpha$ of the probability). More refined strategies have been proposed to account for dependence between coordinates. One line of work defines a scalar nonconformity score from the multi-output residual, for instance using a norm $|Y_{\text{pred}} - Y_{\text{true}}|$ or the maximum deviation across coordinates; this yields prediction balls or boxes aligned to the chosen norm. While simple, such choices typically lead to symmetric or axis-aligned regions that may be suboptimal in shape and volume. For example, the PCP method of Wang et al. (2023) leverages an implicit generative model to draw random samples from $Y \mid X = x$ and constructs the prediction set as a union of Euclidean balls (of a fixed radius) centered at those samples. This approach guarantees marginal coverage and can improve sharpness over naive intervals, but using a global radius for all $x$ can lead to over-coverage in low-variability regions and under-coverage in high-variability regions. Alternatively, some works shape the prediction set as an ellipsoid by incorporating covariance structure: e.g. using a single global covariance estimate (Johnstone & Cox, 2021) or a local covariance around $x$ (Messoudi et al., 2022) to define a Mahalanobis-distance conformity score. Such ellipsoidal regions capture linear correlations in $Y$ and are typically smaller than axis-aligned boxes, but they still assume an (approximately) elliptical and unimodal error distribution, which may be inappropriate for complex multimodal targets.

Another class of methods seeks to learn a joint representation or dependency model for $Y$ before applying conformal. For example, Feldman et al. (2023) train a deep generative model to embed $Y$ into a lower-dimensional (ideally unimodal) latent space and perform conformal quantile regression in that space, producing flexible regions when mapped back to $\mathbb{R}^d$. Similarly, Messoudi et al. (2021) and subsequent works leverage copula transformations: they calibrate marginal predictive intervals at miscoverage levels chosen to optimize the volume of the resulting joint region, effectively shaping the prediction set according to the dependence structure of $Y$. In particular, Zhang et al. (2023) extend copula-based conformal prediction by allowing different significance levels for each output dimension and directly optimizing the hyperrectangle volume under the coverage constraint. Sun & Yu (2024) provide a theoretical analysis of such copula-shaped prediction sets, proving that the empirical copula approach achieves finite-sample validity under i.i.d. assumptions. These methods produce tighter joint regions than the naive Cartesian product by allocating miscoverage intelligently across coordinates, though they often rely on either simple parametric copulas or numerical search to balance the marginal intervals.

Very recently, Dheur et al. (2025) conducted a comprehensive study of multi-output conformal methods, proposing in particular two new families of conformity scores. One uses a generative model (e.g. an invertible normalizing flow) to transform $Y$ into a space where conventional CP can be applied coordinate-wise, and the other defines a computationally efficient scalar score by combin-

ing coordinate-wise conformal $p$-values (essentially summing their logarithms). Both approaches attain finite-sample marginal coverage and offer improvements in conditional coverage. Notably, a conceptually similar idea was introduced concurrently by Fang et al. (2025), who also leverage normalizing flows to define nonconformity in the latent space. Their method (CONTRA) maps high-density regions in the latent space to complex but high-coverage regions in output space, yielding non-axis-aligned prediction sets that outperform standard hyperrectangles or ellipsoids. Despite these advances, none of the above techniques exploits the full geometric structure of multivariate quantiles or ranks. This gap has been filled by two concurrent works that integrate the measure-transport perspective into conformal inference.

Thurin et al. (2025) introduce OT-CP, which uses the center-outward rank function of Hallin et al. (2021) to define multivariate order statistics. In essence, they compute the "rank" of a test point $y$ among past observations in $\mathbb{R}^d$ via the empirical center-outward distribution (obtained by optimal transport), and use the corresponding multivariate quantile level as the nonconformity score. This yields a prediction region for a new $X = x$ by including all $y$ whose center-outward rank is above a certain quantile (determined by the calibration set)—intuitively, the set of points that lie among the $(1 - \alpha)$ fraction most central (least outlying) under the conditional distribution of $Y \mid X = x$. Independently, Klein et al. (2025) develop a related approach that also relies on optimal transport to order multivariate outputs. They formalize the notion of distribution-free multivariate quantile regions and provide finite-sample coverage guarantees for both exact and approximate transport maps. These OT-based conformal methods leverage the geometry of Brenier maps (i.e. conditional Monge–Ampère transports) to construct flexible, data-dependent prediction sets in $\mathbb{R}^d$ that adapt to the local distribution of $Y \mid X = x$. By exploiting the vector-quantile structure, they can achieve tighter coverage with complex (even non-convex) regions while still guaranteeing the rigorous coverage properties that make conformal prediction attractive. However, the use of optimal transport maps can be computationally expensive in high dimensions, and in practice one might need to trade off some statistical efficiency for tractability when estimating the transport.

Finally, an alternative direction is to explicitly optimize prediction set volume subject to coverage, rather than relying on a fixed conformity score. Braun et al. (2025) propose an optimization-driven framework that learns minimum-volume covering sets for multivariate regression. In their approach, the predictive model is trained jointly with a parametric prediction set (for example, an adaptive norm-ball whose radius may vary with $x$) to minimize the volume of the set while enforcing coverage on the training data via a surrogate loss. This procedure effectively learns the shape of the prediction region that best captures a specified proportion of the data. By conformalizing the learned region (i.e. slightly expanding it to guarantee $1 - \alpha$ coverage on a holdout set), the method yields valid prediction sets that are much tighter than those from standard split-conformal methods. Such approaches highlight an exciting trend of combining machine learning and conformal inference: rather than treating the prediction algorithm as a black box, one can optimize the model and its uncertainty quantification in tandem to achieve improved efficiency (smaller, more informative prediction sets) without sacrificing the finite-sample guarantees of CP.

## B    REMARK ON CONTINUITY OF DISTRIBUTION FUNCTION

The requirement that $F_{Y|X=x}$ admit a density in Assumption 2 is stronger than needed, see (Ghosal & Sen, 2022), and is not customary for conditional vector ranks. For our results, it suffices that the source (reference) distribution $F_U$ be absolutely continuous with finite second moment (e.g., uniform on $[0, 1]^d$ or Gaussian). No density assumption on $F_{Y|X=x}$ is required; in particular, $F_{Y|X=x}$ may be discrete.

Under these conditions, for each $x$ there exists a convex potential $\varphi_x$ such that the Brenier map $Q_{Y|X}(\cdot, x) = \nabla\varphi_x$ is defined $F_U$-a.e., is unique $F_U$-a.e., and pushes $F_U$ forward to the conditional law:

$$(Q_{Y|X}(\cdot, x))_{\#}F_U \;=\; F_{Y|X=x}.$$

The (conditional) vector rank is given by the gradient of the convex conjugate, $Q_{Y|X}^{-1}(\cdot, x) = \nabla\varphi_x^*$, which is defined $F_{Y|X=x}$-a.e. (when $F_{Y|X=x}$ is discrete, interpret $Q_{Y|X}^{-1}$ as any measurable selection from $\partial\varphi_x^*$). It transports back to the reference:

$$Q_{Y|X}^{-1}(Q_{Y|X}(u, x), x) = u \text{ for } F_U\text{-a.e. } u.$$

Consequently, we could work under the standing condition that $F_U$ is absolutely continuous with finite second moment; Theorem 1 holds verbatim with the above interpretation.

## C  ENTROPY-REGULARIZED NEURAL VQR

Let $\mathcal{X}, \mathcal{Y}, \mathcal{U}$ be Polish spaces with Borel $\sigma$–algebras, and let $m$ be the marginal law of $X$, $\nu(\mathrm{d}x, \mathrm{d}y) = m(\mathrm{d}x)\,\nu_z(\mathrm{d}y)$ the joint law of $(X, Y)$, and $\mu(\mathrm{d}x, \mathrm{d}u) = m(\mathrm{d}x)\,\bar{\mu}(\mathrm{d}u)$ the joint law of $(X, U)$ (where $\bar{\mu}$ is the marginal distribution of $U$). For $\varepsilon > 0$, the entropic-regularized *conditional* OT problem reads (Carlier et al., 2022)

$$\min_{\gamma \in \mathcal{M}_+(\mathcal{X} \times \mathcal{Y} \times \mathcal{U})} \left\{ -\int u^\top y \mathrm{d}\gamma + \varepsilon \, \mathrm{KL}\big(\gamma \,\|\, \bar{\mu} \otimes \nu\big) \right\} \quad \text{s.t.} \quad \Pi_{X,Y} \# \gamma = \nu, \Pi_{X,U} \# \gamma = \mu. \quad (10)$$

This is a strictly convex problem with linear marginal constraints; KL denotes the Kullback–Leibler divergence. (10) specializes the standard entropic OT to the conditional setting by constraining the two $(X, \cdot)$ marginals of $\gamma$.

**Dual formulation via Fenchel–Rockafellar.** We introduce the dual potentials $\psi \colon \mathcal{X} \times \mathcal{Y} \to \mathbb{R}$ and $\varphi \colon \mathcal{X} \times \mathcal{U} \to \mathbb{R}$. A direct application of Fenchel–Rockafellar duality yields the (unconstrained) dual

$$\inf_{\psi, \varphi} \underbrace{\int \psi(y, x)\, \nu(\mathrm{d}x, \mathrm{d}y)}_{\text{term for } \Pi_{X,Y}} + \underbrace{\int \varphi(u, x)\, \mu(\mathrm{d}x, \mathrm{d}u)}_{\text{term for } \Pi_{X,U}}$$
$$+ \varepsilon \int \int \exp\left( \tfrac{u^\top y - \varphi(u,x) - \psi(y,x)}{\varepsilon} \right) \nu(\mathrm{d}x, \mathrm{d}y) \bar{\mu}(\mathrm{d}u), \quad (11)$$

with zero duality gap and attainment under mild assumptions (tightness and finite entropy) The inequality constraint of the unregularized dual is absorbed by the exponential term barrier in (11). This could be solved using purely stochastic optimization with NN parameterization of the two dual potentials $\psi(y, x)$ and $\varphi(u, x)$ similarly to what was proposed in (Genevay et al., 2016; Seguy et al., 2018). But from a practical perspective the exponential in the loss is particularly hard to optimize with numerical stability problems. This is why Genevay et al. (2016) proposed to remove one of the potentials using the smooth version of the $c$–conjugacy detailed below.

**KKT/first-order conditions:** *soft* $c$**–conjugacy.** Assuming $\nu$ and $\bar{\mu}$ admit densities w.r.t. the Lebesgue measure and differentiating the dual objective in (11) w.r.t. $\psi$ and $\varphi$ gives the optimality (stationarity) conditions

$$\psi_\varepsilon(y, x) = \varepsilon \log \int \exp\left( \tfrac{u^\top y - \varphi_\varepsilon(u,x)}{\varepsilon} \right) \bar{\mu}(\mathrm{d}u), \quad (12)$$

$$\varphi_\varepsilon(u, x) = \varepsilon \log \int \exp\left( \tfrac{u^\top y - \psi_\varepsilon(y,x)}{\varepsilon} \right) \nu_x(\mathrm{d}y), \quad (13)$$

which are the entropic ("soft") $c$–transforms, i.e., log-partition functions of exponential families induced by the bilinear cost $c(u, y) = -u^\top y$. At $\varepsilon \downarrow 0$, the identities (12)–(13) $\Gamma$–converge to the hard Fenchel conjugacy $\psi = \varphi^\star$, recovering the unregularized dual feasibility $\varphi(u, x) + \psi(y, x) \geq u^\top y$ with equality on the support of the optimal plan.

**Reduction to a single potential (semi-dual).** Eliminating $\psi$ in (11) via (12) yields an equivalent unconstrained problem in $\varphi$:

$$\mathcal{U}_\varepsilon(\varphi) = \mathbb{E}_{(X,U) \sim \mu}\big[\varphi(U, X)\big] + \mathbb{E}_{(X,Y) \sim \nu}\left[ \varepsilon \log \int \exp\left( \tfrac{u^\top Y - \varphi(u,X)}{\varepsilon} \right) \bar{\mu}(\mathrm{d}u) \right], \quad (14)$$

which is precisely the regularized analogue of the conjugate-based loss in the unregularized case (log-sum-exp replaces the sup). This problem is very interesting from an optimization perspective because now a unique dual potential needs to be optimized and the log-sum-exp can be implemented in a much more stable way than the exponential in the dual (11). But then the inner expectation in the right part of (14) cannot be computed exactly, which we discuss next.

**Gibbs conditionals and gradients.** Define the Gibbs conditional density (a.k.a. Schrödinger bridge "posterior")

$$\pi_\varphi(\mathrm{d}u \mid y, x) \propto \exp\left(\tfrac{u^\top y - \varphi(u,x)}{\varepsilon}\right)\bar{\mu}(\mathrm{d}u).$$

As in the not regularized case, we parameterize the potential $\varphi_\epsilon$ with a neural network. We denote by $\theta$ the parameters (weights) of this network. Using the log-partition derivative identity, we get that $\nabla_\theta \mathcal{U}_\varepsilon(\varphi_\theta)$ admits the "positive minus negative phase" form

$$\nabla_\theta \mathcal{U}_\varepsilon(\varphi_\theta) = \mathbb{E}_{(X,U)\sim\mu}\big[\nabla_\theta \varphi_\theta(X,U)\big] - \mathbb{E}_{(X,Y)\sim\nu}\,\mathbb{E}_{U\sim\pi_{\varphi_\theta}(\cdot\mid Y,X)}\big[\nabla_\theta\,\varphi_\theta(X,U)\big], \qquad (15)$$

obtained by differentiating the log-partition in (14). In practice, the inner expectation is estimated by Monte Carlo with $U$ drawn either from $\pi_{\varphi_\theta}(\cdot \mid Y, X)$ or via importance sampling from $\bar{\mu}$ with the usual exponential weights.

**Quantile and rank maps under entropic regularization.** If $u \mapsto \varphi_\varepsilon(u, x)$ is (strongly) convex and smooth, the regularized analogues of the conditional vector quantile and rank are

$$Q_{Y|X}^{(\varepsilon)}(u, x) := \nabla_u \varphi_\varepsilon(u, x), \qquad (16)$$

$$\big(Q_{Y|X}^{(\varepsilon)}\big)^{-1}(y, x) := \nabla_y \psi_\varepsilon(y, x) = \mathbb{E}_{U\sim\pi_{\varphi_\varepsilon}(\cdot\mid y,x)}[U], \qquad (17)$$

where the last identity follows by differentiating (12). Equations (16)–(17) are the entropic counterparts of the unregularized identities and reduce to them as $\varepsilon \downarrow 0$.

**Limit $\varepsilon \downarrow 0$.** As $\varepsilon \to 0$, $\varepsilon \log \int \exp((\cdot)/\varepsilon) \to \sup(\cdot)$, so

$$\mathcal{U}_\varepsilon(\varphi) \xrightarrow[\varepsilon\downarrow 0]{} \mathbb{E}_\mu[\varphi(X,U)] + \mathbb{E}_\nu\big[\varphi^\star(X,Y)\big],$$

recovering the unregularized loss with the hard Fenchel conjugate and the transition from the constrained dual (inequality) to the unconstrained conjugate form. In the same limit, $\pi_\varphi(\cdot \mid y, x)$ concentrates on the (possibly set-valued) argmax of $u \mapsto u^\top y - \varphi(u, x)$, and (16)–(17) converge to the OT maps of the unregularized problem.

## D  CONDITIONAL CONVEX POTENTIAL FLOWS

**Conditional (partially convex) construction.** Given covariates $x \in \mathcal{X}$, we model the conditional transport by a *partially* input–convex potential

$$\varphi \colon \mathbb{R}^d \times \mathcal{X} \to \mathbb{R}, \qquad u \mapsto \varphi(u; x) \text{ convex (strongly convex) for each fixed } x,$$

and define the *conditional convex potential flow* (a.k.a. *partially convex potential flow*)

$$Q_{Y|X}(u, x) := \nabla_u \varphi(u; x), \qquad U \sim f_U \rightsquigarrow Y \mid X = x \text{ via } Y = Q_{Y|X}(U, x).$$

Under absolute continuity of $f_{Y|X}(\cdot \mid x)$ (see Assumption 2), the conditional *rank* map (inverse quantile) exists and is the gradient of the conjugate:

$$Q_{Y|X}^{-1}(y, x) = \nabla_y \varphi^\star(y; x),$$

and the two maps are inverses a.e. (in $u$ and $y$) for each $x$. For any $(y, x)$ such that the inverse is well defined.

$$f_{Y|X}(y \mid x) = f_U\big(Q_{Y|X}^{-1}(y, x)\big) \det\Big[\nabla_y Q_{Y|X}^{-1}(y, x)\Big]. \qquad (18)$$

Equivalently, writing $y = \nabla_u \varphi(u; x)$ with $u = Q_{Y|X}^{-1}(y, x)$,

$$\log f_{Y|X}(y \mid x) = \log f_U(u) - \log \det\big[\nabla_{uu}^2 \varphi(u; x)\big].$$

Thus maximum likelihood amounts to estimating $\varphi$ so as to match the pullback $Q_{Y|X}^{-1}(Y, X)$ to the prior $f_U$, while penalizing the local volume change through the (log) Hessian determinant. In practice, the log-determinant and its gradients can be computed with Hessian–vector products, using stochastic Lanczos/trace estimators and conjugate-gradient solves, yielding unbiased $O(1)$–memory estimators that scale to high dimension.

**Inversion and sampling.** For any $(y, x)$, inversion is a convex program:

$$Q_{Y|X}^{-1}(y, x) = \arg \min_{u \in \mathbb{R}^d} \varphi(u; x) - y^\top u,$$

whose optimality condition $\nabla_u \varphi(u; x) = y$ recovers the required $u$. This is precisely the evaluation of $\nabla_y \varphi^\star(y; x)$ and can be carried out with off-the-shelf smooth convex solvers; batched inversions reduce to minimizing summed potentials over independent inputs.

Under mild regularity (convex support and densities), there exists a measurable conditional vector quantile $Q_{Y|X}$ that is the gradient (in $u$) of a convex potential and pushes $U$ to $Y \mid X = x$; the inverse rank is the gradient (in $y$) of the conjugate, and $Q_{Y|X}$ solves the $W_2$ OT problem conditionally on $x$. Hence the partially convex potential flow inherits both identifiability (a.e. uniqueness) and optimality properties in the conditional setting.

**Parameterization.** We instantiate $\varphi(\cdot; x)$ with partially input–convex networks (e.g., PICNN/PISCNN) to guarantee convexity in $u$ while conditioning on $x$, and add a quadratic $\frac{\alpha}{2}\|u\|^2$ when strong convexity is desired. Universality of ICNNs in approximating convex functions then lifts to distributional universality of the induced conditional flows and convergence to the conditional OT maps.

# E   NUMERICAL IMPLEMENTATION

This section details architectures, solvers, and training procedures for our neural vector quantile regression (VQR) models, both in the unregularized and entropic-regularized settings, together with the amortized conjugate prediction used to accelerate training. We emphasize implementation choices that preserve convexity/monotonicity and lead to stable gradients, and we provide concrete defaults for reproducibility.

**Notation recap.** We parameterize a *conditional convex potential* $\varphi_\theta : \mathcal{U} \times \mathcal{X} \to \mathbb{R}$ that is convex in $u \in \mathcal{U} \subset \mathbb{R}^{d_y}$ for each fixed $x \in \mathcal{X}$. The conditional vector quantile and rank maps are the gradients of $\varphi_\theta$ and its Fenchel conjugate $\varphi_\theta^\star$ (see Section 2):

$$Q_{Y|X}(u, x) = \nabla_u \varphi_\theta(u, x), \qquad Q_{Y|X}^{-1}(y, x) = \nabla_y \varphi_\theta^\star(y, x).$$

The conjugate evaluation at $(y, x)$ solves $\widehat{u}_\theta(y, x) \in \arg\max_{u \in \mathcal{U}} \{u^\top y - \varphi_\theta(u, x)\}$. By Danskin's theorem, gradients w.r.t. $\theta$ do *not* require differentiating through $\widehat{u}_\theta$; only $\nabla_\theta \varphi_\theta$ at $u = \widehat{u}_\theta$ is needed.

## E.1   PARTIALLY INPUT CONVEX NEURAL NETWORKS (PICNN)

We instantiate $\varphi_\theta$ as a *Partially Input Convex Neural Networks* (PICNNs; Amos et al., 2017) that is convex in $u$ and conditions on $x$:

$$(u, x) \longmapsto \varphi_\theta(u, x) = \text{PICNN}(u, x; \theta),$$

with layerwise updates

$$c_{i+1} = \tilde{g}_i(\tilde{W}_i c_i + \tilde{b}_i),$$
$$z_{i+1} = g_i\Big(W_i^{(z)}\big(z_i \circ \big[W_i^{(zc)} c_i + b_i^{(z)}\big]_+\big) + W_i^{(u)}\big[u \circ \big(W_i^{(uc)} c_i + b_i^{(u)}\big)\big] + W_i^{(c)} c_i + b_i\Big),$$

and output $\varphi_\theta(u, x) = z_K$. We initialize $c_0 = x, z_0 = 0$. Here $\circ$ denotes the element-wise product. We enforce elementwise nonnegativity of $W_i^{(z)}$ and $[\cdot]_+$ via a Softplus reparameterization:

$$W_i^{(z)} = \log\Big(1 + \exp\Big(\tilde{W}_i^{(z)}\Big)\Big), \quad \tilde{W}_i^{(z)} \in \mathbb{R}^{p \times k}, \tag{19}$$

$$[W_i^{(zc)} c_i + b_i^{(z)}]_+ = \log\Big(1 + \exp\Big(W_i^{(zc)} c_i + b_i^{(z)}\Big)\Big). \tag{20}$$

We use convex, non-decreasing activations for $g_i, \tilde{g}_i$, which guarantees convexity in $u$ while retaining expressive power. We optionally add a quadratic term $\frac{\alpha}{2}\|u\|_2^2$ (trainable $\alpha \geq 0$) to obtain strong convexity, improving stability of the inner argmax (Amos et al., 2017, Proposition 2). We choose

`Softplus` as non-linearity for $g_i$ and `ELU` as non-linearity for $c_i$. Following Huang et al. (2021) we utilize activation normalization `ActNorm` layers (Kingma & Dhariwal, 2018) before applying the $g_i$ non-linearity. Final architecture of one iterate hence becomes.

$$c_{i+1} = \text{ELU}(\tilde{W}_i c_i + \tilde{b}_i),$$

$$z_{i+1} = \text{Softplus}\Big( \text{ActNorm}\Big( W_i^{(z)} \big( z_i \circ \big[ W_i^{(zc)} c_i + b_i^{(z)} \big]_+ \big)$$

$$+ W_i^{(u)} \big( u \circ \big[ W_i^{(uc)} c_i + b_i^{(u)} \big] \big) + W_i^{(c)} c_i + b_i \Big) \Big),$$

**Practical tips (PICNN).**

(i) Normalize $u$ and $y$ scales (e.g. standardization) to ease optimization;
(ii) We use weight decay on $\theta$ and (if enabled) a small ridge $\alpha$ to avoid flat directions;
(iii) We clip gradients of $\varphi_\theta$ to bound the Lipschitz constant of $u \mapsto \nabla_u \varphi_\theta(u, x)$.

### E.2 PARTIALLY INPUT STRONGLY CONVEX NEURAL NETWORK (PISCNN)

$$\text{PISCNN}(u, x) = \text{PICNN}(u, x) + \frac{\alpha}{2} \|u\|_2^2,$$

which is strongly convex in $u$ and yields a *strictly concave* inner objective $u \mapsto u^\top y - \varphi_\theta(u, x)$, ensuring a unique maximizer $\widehat{u}_\theta(y, x)$ and faster, more reliable inner solves. We treat $\alpha$ as positive scalar parametrized by $e^w$, where $w$ is a trainable parameter. In all our implementations, enabling $\alpha > 0$ eliminated numerical non-uniqueness in the conjugate and reduced inner iterations.

### E.3 COMPUTING THE CONJUGATE: INNER MAXIMIZATION

Given $(y, x)$ and current $\theta$, we compute

$$\widehat{u}_\theta(y, x) \in \arg\max_{u \in \mathcal{U}} J_\theta(u; y, x), \quad J_\theta(u; y, x) := u^\top y - \varphi_\theta(u, x).$$

**Gradient and Hessian.** $\nabla_u J_\theta(u; y, x) = y - \nabla_u \varphi_\theta(u, x)$ and $\nabla_{uu}^2 J_\theta(u; y, x) = -\nabla_{uu}^2 \varphi_\theta(u, x) \preceq 0$. With PISCNN, $\nabla_{uu}^2 \varphi_\theta(u, x) \succeq \alpha I$ ensures strong concavity.

**Solver.** We minimize $-J_\theta$ with L-BFGS. For stability:

1. **Warm start.** We initialize the solver from amortized predictor $\tilde{u}_\vartheta(y, x)$ when available (see Section E.4) or otherwise initialize it at $u \sim F_U$.
2. **Domain handling.** If $\mathcal{U}$ is a ball/hypercube, we project the solution after each step: $u \leftarrow \Pi_{\mathcal{U}}(u)$.
3. **Stopping.** Terminate when $\|\nabla_u J_\theta\| \leq \varepsilon_{\text{norm}}$, $\|J_\theta(u_i; y, x) - J_\theta(u_{i+1}; y, x)\| \leq \varepsilon_{\text{obj}}$ or after $K_{\max}$ steps (defaults: $\varepsilon_{\text{norm}} = 10^{-7}, \varepsilon_{\text{obj}} = 10^{-7}, K_{\max} = 1000$).

### E.4 AMORTIZED CONJUGATE PREDICTION

To avoid expensive inner solves at every iteration, we learn a differentiable predictor $\tilde{u}_\vartheta : \mathcal{Y} \times \mathcal{X} \to \mathcal{U}$ that approximates $\check{u}_\vartheta(y, x)$ and serves as a warm start for L-BFGS solver. We parametrize $\tilde{u}_\vartheta(y, x)$ as feed forward neural network with a residual skip connection to encourage identity at initialization

$$\tilde{u}_\vartheta(y, x) = \text{MLP}_\vartheta\left( \begin{bmatrix} y \\ x \end{bmatrix} \right) + W_y y + b_y.$$

**Training losses.** Several loss functions have been explored in the literature. Objective-based losses (Dam et al., 2019; Amos, 2023) optimize the network to predict the maximum of the conjugate by maximizing $\mathcal{V}_{\text{obj}} = \mathbb{E}_{(Y,X) \sim F_{YX}}[J_\theta(\tilde{u}_\vartheta; y, x)]$. Alternatively, one may enforce the first-order condition $\nabla_u \varphi_\theta(u, x)|_{u = \tilde{u}_\vartheta(y,x)} \approx y$ via the residual loss $\mathcal{V}_{\text{res}}(\vartheta) = \mathbb{E}_{(Y,X) \sim F_{YX}}[\|\nabla_u \varphi_\theta(u, x)|_{u = \tilde{u}_\vartheta(y,x)} - y\|_2^2]$. If the true conjugate $\tilde{u}_{\varphi_\theta(\cdot, x)}(y)$ (5) is available, one can regress directly with $\mathcal{V}_{\tilde{u}} = \mathbb{E}_{(Y,X) \sim F_{YX}}[\|\tilde{u}_\vartheta(y, x) - \check{u}_{\varphi_\theta(\cdot, x)}(y)\|_2^2]$. In practice, we observe no major differences between these approaches and therefore adopt $\mathcal{V}_{\tilde{u}}$ as our loss of choice (see Algorithm 2).

### E.5 Entropic-regularized Semi-dual

When using the entropic semi-dual $U_\varepsilon(\varphi)$ (see Appendix C), we replace the hard conjugate with a log-sum-exp:

$$U_\varepsilon(\varphi_\theta) = \mathbb{E}_{(X,U)}[\varphi_\theta(U, X)] + \mathbb{E}_{(X,Y)}\left[\varepsilon \log \mathbb{E}_{U \sim F_U} \exp\left(\frac{U^\top Y - \varphi_\theta(U, X)}{\varepsilon}\right)\right].$$

**Monte Carlo and stability.** We approximate the inner expectation with $m$ i.i.d. samples $U_j \sim F_U$, using a numerically stable log-sum-exp with 64-bit accumulation. We found $m \in [512, 1024]$ adequate on our benchmarks, and we re-sample the $U_j$ each iteration. In the $\varepsilon \downarrow 0$ limit, this recovers the unregularized loss. We intentionally set high amount of samples for dual objective estimation to avoid effects related to high bias of logsumexp estimator.

**Gradients.** The gradient has a positive-minus-negative phase form using the Gibbs weights (see Appendix C and equation (15)), which we implement without storing the full batch $\times m$ tensor by streaming accumulation.

### E.6 Training Loops and Algorithms

We describe three loops: (i) Neural Vector Quantile Regression without amortization Algorithm 1, (ii) Amortized Vector Quantile Regression Algorithm 2, and (iii) Entropic Semi-dual Algorithm 3. All use `AdamW` (initial LR of $10^{-2}$, weight decay $10^{-4}$) with cosine annealing (LR decaying to 0), batch size 1024, and gradient clipping at 10. We sample $U \sim F_U$ as standard Gaussian unless otherwise noted. See Appendix H.2 for dataset-specific details. We use warm restarts for amortized network, restarting the learning rate to $10^{-2}$ each 10 epochs.

---

**Algorithm 1** Neural Vector Quantile Regression Training (C-NQR)

---

1: **Input:** dataset $\{(x_i, y_i)\}_{i=1}^n$, PICNN $\varphi_\theta \colon \mathcal{U} \times \mathcal{X} \to \mathbb{R}$
2: Sample mini-batch $\mathcal{B} \subset \{1, \ldots, n\}$
3: Initialize $\mathcal{V}_\varphi \leftarrow 0$
4: **for** each $i \in \mathcal{B}$ **do**
5:     $\tilde{u}_i \leftarrow \arg\max_{u \in \mathcal{U}} J_{\varphi_\theta(\cdot, x_i)}(u, y_i)$                   ▷ Run L-BFGS for each $y_i$ starting at $u = 0$
6:     $\widehat{\psi}_i(\theta) \leftarrow J_{\varphi_\theta(\cdot, x_i)}(\tilde{u}_i, y_i)$
7:     Sample $u_i \sim \mathcal{N}(0, I_d)$
8:     $\widehat{\varphi}_i(\theta) \leftarrow \varphi_\theta(u_i, x_i)$
9:     $\widehat{\mathcal{V}}_\varphi(\theta) \leftarrow \widehat{\mathcal{V}}_\varphi(\theta) + \widehat{\psi}_i(\theta) + \widehat{\varphi}_i(\theta)$
10: **end for**
11: Compute $\nabla_\theta \frac{1}{|\mathcal{B}|} \widehat{\mathcal{V}}_\varphi(\theta)$                         ▷ Do not propagate gradients through $\tilde{u}$
12: Update $\theta$ with Adam

---

### E.7 Conformal Methods Implementation

Here, we provide a detailed description of our implementation of the methods introduced in Section 4. For all proposed approaches, we start with an estimate $\widehat{Q}_{Y|X}^{-1}(y, x)$ that we obtain using a training set $\mathcal{D}_{\text{train}}$. All conformal methods operate on a separate held-out calibration set $\mathcal{D}_{\text{cal}}$. Since we need to replicate our uncertainty estimation experiments for multiple splits and datasets, we use the Amortized Neural Vector Quantile Regression version of our algorithm.

**Split Conformal Prediction with Monge-Kantorovich ranks.** An instance of classical split conformal prediction using a score derived from our vector quantile regressor.

**Fixed re-ranking.** To account for the misspecification of $\widehat{Q}_{Y|X}^{-1}(y, x)$ we introduce an intermediate re-ranking of $U_i$. We follow the approach of Thurin et al. (2025), but instead of a separate base model, we directly use our estimate: $S_i = U_i = \widehat{Q}_{Y|X}^{-1}(Y_i, X_i) \in \mathbb{R}^{d_y}$. We divide our calibration set into two parts: the first part is used to estimate an OT map $\mathbf{R} \colon \mathcal{U} \to \mathcal{U}'$ and the second part is

---

**Algorithm 2** Amortized Neural Vector Quantile Regression Training (AC-NQR)

---

1: **Input:** dataset $\{(x_i, y_i)\}_{i=1}^n$, PICNN $\varphi_\theta \colon \mathcal{U} \times \mathcal{X} \to \mathbb{R}$, $\tilde{u}_\vartheta(y, x) \colon \mathcal{Y} \times \mathcal{X} \to \mathcal{U}$
2: Sample mini-batch $\mathcal{B} \subset \{1, \ldots, n\}$
3: Initialize $\mathcal{V}_\varphi \leftarrow 0, \mathcal{V}_{\tilde{u}} \leftarrow 0$
4: **for** each $i \in \mathcal{B}$ **do**
5:     $\tilde{u}_i \leftarrow \tilde{u}_\vartheta(y_i, x_i)$
6:     $\check{u}_i \leftarrow \arg\max_{u \in \mathcal{U}} J_{\varphi_\theta(\cdot, x_i)}(u, y_i)$              ▷ Run L-BFGS for each $y_i$ starting at $u = \tilde{u}_i$
7:     $\widehat{\psi}_i(\theta) \leftarrow J_{\varphi_\theta(\cdot, x_i)}(\check{u}_i, y_i)$
8:     Sample $u_i \sim \mathcal{N}(0, I_d)$
9:     $\widehat{\varphi}_i(\theta) \leftarrow \varphi_\theta(u_i, x_i)$
10:    $\widehat{\mathcal{V}}_\varphi(\theta) \leftarrow \widehat{\mathcal{V}}_\varphi(\theta) + \widehat{\psi}_i(\theta) + \widehat{\varphi}_i(\theta)$
11:    $\widehat{\mathcal{V}}_{\tilde{u}}(\vartheta) \leftarrow \widehat{\mathcal{V}}_{\tilde{u}}(\vartheta) + \|\tilde{u}_i - \check{u}_i\|_2^2$
12: **end for**
13: Compute $\nabla_\theta \frac{1}{|\mathcal{B}|} \widehat{\mathcal{V}}_\varphi(\theta)$ and $\nabla_\vartheta \frac{1}{|\mathcal{B}|} \widehat{\mathcal{V}}_{\tilde{u}}(\vartheta)$        ▷ Do not propagate gradients through $\check{u}$
14: Update $\theta$ and $\vartheta$

---

**Algorithm 3** Entropic semi-dual training (EC-NQR)

---

1: **Input:** dataset $\{(x_i, y_i)\}_{i=1}^n$, PICNN $\varphi_\theta \colon \mathcal{U} \times \mathcal{X} \to \mathbb{R}$
2: Sample mini-batch $\mathcal{B} \subset \{1, \ldots, n\}$
3: initialize $\mathcal{L}_\varphi \leftarrow 0$
4: Sample i.i.d. $u_{ij} \sim F_U$
5: **for** each $i \in \mathcal{B}$ **do**
6:     $\widehat{\psi}_i(\theta) \leftarrow \epsilon \log \sum_{j=1}^m \exp\left( \frac{u_{ij}^T y_i - \varphi_\theta(u_{ij}, x_i)}{\epsilon} \right)$        ▷ $\varepsilon \in [10^{-3}, 10^{-1}]$
7:     Sample $u_i \sim F_U$
8:     $\widehat{\varphi}_i(\theta) \leftarrow \varphi_\theta(u_i, x_i)$;
9:     $\mathcal{L}_\varphi(\theta) \leftarrow \mathcal{L}_\varphi(\theta) + \widehat{\psi}_i(\theta) + \widehat{\varphi}_i(\theta)$
10: **end for**
11: Compute $\nabla_\theta \frac{1}{|\mathcal{B}|} \mathcal{L}_\varphi(\theta)$
12: Update $\theta$ with Adam

---

**Algorithm 4** Pull-back split conformal prediction

---

1: **Input:** dataset $\mathcal{D}_{\text{cal}} = \{(X_i, Y_i)\}_{i=1}^n$, trained model $\widehat{Q}_{Y|X}^{-1}(y, x)$, a new test point $(X_{\text{test}}, Y_{\text{test}})$ and the desired nominal miscoverage level $\alpha$
2: **for** each $i \in \{1, \ldots, n\}$ **do**
3:     $U_i \leftarrow \widehat{Q}_{Y|X}^{-1}(Y_i, X_i)$
4:     $S_i \leftarrow \|U_i\|$
5: **end for**
6: $\rho_{1-\alpha} \leftarrow \lceil (n+1)(1-\alpha) \rceil$-th largest $S_i$
7: $\hat{\mathcal{C}}_\alpha^{\text{pb}}(X_{\text{test}}) \leftarrow \left\{ y \colon \|\widehat{Q}_{Y|X}^{-1}(y, X_{\text{test}})\| \le \rho_{1-\alpha} \right\}$

---

used to conformalize the result. In our experiments, we follow the original authors' approach and use $\mathcal{U}' = \mathrm{U}(S^{d_y - 1})$ - uniform distribution on the unit ball. To evaluate the map $\widehat{\mathbf{R}}$ on the new point, we map it to the corresponding closest point from the first calibration part.

We use the code of Thurin et al. (2025) to estimate $\widehat{\mathbf{R}}$ (we divide the original calibration set into two equal parts). This implementation uses the renowned POT library (Flamary et al., 2021), which provides efficient implementations of the various optimal transport techniques.

---

**Algorithm 5** Re-ranked pull-back split conformal prediction

---

1: **Input:** dataset $\mathcal{D}_{\text{cal}} = \{(X_i, Y_i)\}_{i=1}^{n=n_1+n_2}$, trained model $\widehat{Q}_{Y|X}^{-1}(y, x)$, a new test point $(X_{\text{test}}, Y_{\text{test}})$ and the desired nominal miscoverage level $\alpha$
2: **for** each $i \in \{1, \dots, n_1\}$ **do**
3:     $U_i \leftarrow \widehat{Q}_{Y|X}^{-1}(Y_i, X_i)$
4: **end for**
5: Estimate $\widehat{\mathbf{R}}$ using sample $(\{U_i\}_{i=1}^{n_1}, \{U_i'\}_{i=1}^{n_1})$        $\triangleright$ $\{U_i'\}_{i=1}^{n_1}$ - reference sample from $\mathcal{U}'$
6: **for** each $j \in \{1, \dots, n_2\}$ **do**
7:     $S_j \leftarrow \left\| \widehat{\mathbf{R}}\left(\widehat{Q}_{Y|X}^{-1}(Y_j, X_j)\right) \right\|$
8: **end for**
9: $\rho_{1-\alpha} \leftarrow \lceil (n_2+1)(1-\alpha) \rceil$-th largest $S_j$
10: $\hat{\mathcal{C}}_\alpha^{\text{rpb}}(X_{\text{test}}) \leftarrow \left\{ y\colon \left\| \widehat{\mathbf{R}}\left(\widehat{Q}_{Y|X}^{-1}(y, X_{\text{test}})\right) \right\| \leq \rho_{1-\alpha} \right\}$

---

Table 3: Model hyperparameters for different datasets.

| Dataset(s) | Layer width | Layer depth | Batch size |
|---|---|---|---|
| bio | 12 | 4 | 512 |
| blog | 16 | 4 | 512 |
| sgemm | 46 | 4 | 8192 |
| scm20d | 10 | 1 | 2048 |
| *Banana, Convex Banana, Star, Convex Star* | 18 | 8 | 256 |
| *Glasses, Convex Glasses, Funnel* | 18 | 8 | 256 |

### E.8 HYPERPARAMETERS AND DEFAULT CONFIGURATION

- **Network sizes.** We typically use around 10% of available data as parameters scale. See Appendix E.8 for details.

- **Optimization.** AdamW (LR $10^{-2}$, weight decay $10^{-4}$). We use cosine warm restart for amortization network every 5k–10k steps; We clip gradients at 1.0.

- **Inner solver.** L-BFGS with Wolfe line search, $K_{\max} = 50$ (amortized) or 100 (no amortization); tolerance $10^{-5}$; domain projection when $\mathcal{U}$ is bounded.

- **Amortizer.** Amortization network copies the potential network architecture in all our experiments.

- **Entropic.** In all our experiments we fix $\varepsilon = 0.001$; $m = 1024$ Monte Carlo samples per $(x, y)$.

These defaults matched the settings used across Section 6.1 and Section 6.2 (metrics and datasets).

## F DEFERRED CONTENT FOR CONFORMAL PREDICTION

We now proceed to provide the deferred content from Section 4. We start by restating Theorem 3 and its proof. Then, we showcase a setting where the assumptions of Theorem 3 are met. Finally, we explain how the OT maps $Q_{Y|X}$ and $Q_{Y|X}^{-1}$ may be used to construct conformal sets using density estimation.

**Theorem 4** (Volume–optimality of pullback balls under radiality). *Fix $x \in \mathcal{X}$ and reference distribution $F_U(u) = \phi(\|u\|)$ for a continuous $\phi\colon [0, \infty) \to (0, \infty)$ on $\mathcal{U}$, under the assumptions of Theorem 1, let $Q_{Y|X}$ and $Q_{Y|X}^{-1}$ be the vector quantile and multivariate rank functions. Assume that there exists $j_x$ such that for all $y$ in the support of $F_{Y|X}$, it holds*

$$\det\left[\nabla_y Q_{Y|X}^{-1}(y, x)\right] = j_x\left(\|Q_{Y|X}^{-1}(y, x)\|\right),$$

*and the function $r \mapsto \phi(r) \, j_x(r)$ is strictly decreasing. Let $r_\alpha > 0$ be the unique radius satisfying $\mu(B_{r_\alpha}) = 1 - \alpha$, where $\mu$ is the law corresponding to $F_U$ and $B_r = \{u \colon \|u\| \le r\}$. Define the pullback ball $\mathcal{C}_\alpha^{\mathrm{pb}}(x) \coloneqq \left\{ y \colon \|Q_{Y|X}^{-1}(y, x)\| \le r_\alpha \right\}$. Then, $\mathcal{C}_\alpha^{\mathrm{pb}}(x)$ minimizes Lebesgue volume among all sets with $x$-conditional coverage of at least $1 - \alpha$, i.e., for every measurable $A \subset \mathcal{Y}_x$ with $\mathbb{P}\{Y \in A \mid X = x\} \ge 1 - \alpha$, $\mathrm{Vol}(\mathcal{C}_\alpha^{\mathrm{pb}}(x)) \le \mathrm{Vol}(A)$.*

*Proof.* Let $S_x(\cdot) = Q_{Y|X}^{-1}(\cdot)$. Then, by the change of variables formula for densities:

$$f_{Y|X}(y, x) = f_U(S_x(y)) \, \left| \det \left[ \nabla_y S_x(y) \right] \right|.$$

Using the assumption that $f_U(u) = \phi(\|u\|)$ and $\det \left[ \nabla_y S_x(y) \right] = j_x(\|S_x(y)\|)$. Using Carlier et al. (2016, Corollary 2.1), we note that $S_x$ is $C^1$ and the derivative of a convex function. Thus, it holds that $y \to \det \left[ \nabla_y S_x(y) \right]$ is positive and continuous, which allow for dropping absolute value to recover

$$f_{Y|X}(y, x) = \phi(\|S_x(y)\|) \, j_x(\|S_x(y)\|) =: h_x(\|S_x(y)\|).$$

As both $\phi$ and $y \to j_x(\|S_x(y)\|)$ are continuous, $h_x$ is a strictly decreasing continuous invertible function. Hence, $f_{Y|X}(\cdot, x)$ is a non-increasing function of the $U$–radius $\|S_x(y)\|$ and its superlevel sets are pullbacks of Euclidean balls: for each $t > 0$ there exists $r(t) \ge 0$ such that

$$\{y \colon f_{Y|X}(y, x) \ge t\} = \{y \colon h_x(\|S_x(y)\|) \ge t\} = \{y \colon \|S_x(y)\| \le r(t)\}.$$

We first record the probability identity. For any Borel $A \subset \mathcal{Y}_x$,

$$\mathbb{P}\{Y \in A \mid X = x\} = \mu\left(\{S_x(y) | y \in A\}\right).$$

Therefore $\mathbb{P}\{Y \in \mathcal{C}_\alpha^\star(x) \mid X = x\} = \mu(B_{r_\alpha}) = 1 - \alpha$.

For volume optimality, note that since $f_{Y|X}(y, x) = h_x(\|S_x(y)\|)$ with $h_x$ non-increasing, every HPD superlevel set $\{y \colon f_{Y|X}(y, x) \ge t\}$ is (almost surely) a pullback set of the form $\{y | \|S_x(y)\| \le r(t)\}$. Choosing $t_\alpha$ so that $\mathbb{P}\{Y \in \{f_{Y|X}(\cdot, x) \ge t_\alpha\} \mid X = x\} = 1 - \alpha$ forces $\mu(B_{r(t_\alpha)}) = 1 - \alpha$, hence $r(t_\alpha) = r_\alpha$ and the HPD set equals $\mathcal{C}_\alpha^{\mathrm{pb}}(x)$. $\qquad\square$

*Remark* 4 (Examples satisfying assumptions of Theorem 3). Fix $x$. Let the reference be spherical with radial, strictly decreasing continuous density $f_U(u) = \phi(\|u\|)$. Suppose $Y \mid X = x$ is elliptical with location $m(x)$ and a positive definite scatter matrix $\Sigma(x)$ whose whitened density uses the same radial generator as $U$, i.e.,

$$f_{Y|X=x}(y) \; \propto \; \phi\left(\left\|\Sigma(x)^{-1/2}(y - m(x))\right\|\right).$$

Then the map $S_x(y) = \Sigma(x)^{-1/2}(y - m(x))$ and $\det \left[ \nabla_y S_x(y) \right] \equiv \det\left(\Sigma(x)^{-1/2}\right)$. This setting includes the Gaussian case by taking $\phi(r) \propto e^{-r^2/2}$.

To show that this setting indeed satisfies the assumptions of Theorem 3, we start from the transport map $T_x$. By construction of the elliptical model, $T_x(u) = m(x) + \Sigma(x)^{1/2} u$ pushes $F_U$ to $F_{Y|X=x}$. Moreover, $T_x$ is the gradient of the convex quadratic

$$\psi_x(u) = \tfrac{1}{2} u^\top \Sigma(x)^{1/2} u + m(x)^\top u,$$

so by Brenier–Knott–Smith theory for the quadratic cost, $T_x$ is the optimal transport from $F_U$ to $F_{Y|X=x}$.

Denote the inverse map by $S_x(y) = \Sigma(x)^{-1/2}(y - m(x))$. Its Jacobian is constant, $\nabla_y S_x(y) = \Sigma(x)^{-1/2}$, and therefore

$$\det\left(\nabla_y S_x(y)\right) = \det\left(\Sigma(x)^{-1/2}\right) =: c_x > 0.$$

Theorem 3 assumes the existence of a function $j_x \colon [0, \infty) \to (0, \infty)$ such that

$$\det\left(\nabla_y Q_{Y|X}^{-1}(y, x)\right) = j_x\left(\|Q_{Y|X}^{-1}(y, x)\|\right).$$

In the elliptical case $Q_{Y|X}^{-1}(y,x) = S_x(y)$ and $\det(\nabla_y S_x(y)) \equiv c_x$, so we can simply set $j_x(r) = c_x = \det(\Sigma(x)^{-1/2})$ for all $r \geq 0$.

Finally, since $\phi$ is strictly decreasing and $j_x(r) = c_x > 0$ is constant, the product $r \mapsto \phi(r)j_x(r) = c_x\phi(r)$ is strictly decreasing in $r$, as required.

To show that $S_x$ is indeed the optimal transport map, note that $S_x$ is the gradient of convex quadratic function. Thus, it satisfies the Brenier optimal transport conditions for the Euclidean quadratic cost and , by Knott–Smith optimality criterion, it is the vector quantile function (Knott & Smith, 1984).

**Conformal HDP Sets using OT Parameterization.** While the CQR-like construction in Section 4 is robust and simple, its prediction sets are images of Euclidean spheres and thus topologically connected since, under Assumption 1 and Assumption 2, $Q_{Y|X}^{-1}$ is continuous by Carlier et al. (2016, Corollary 2.1). This can be inefficient if for some $x \in \mathcal{X}$, the true conditional distribution $F_{Y|X=x}$ is multimodal, for example a Gaussian mixture. To solve this problem, it is possible to construct prediction sets using the level sets of an estimated conditional density, which can naturally form disconnected regions.

This approach utilizes the change-of-variables formula and leveraging $\widehat{Q}_{Y|X}^{-1}$ to recover the plug-in conditional density estimator

$$\widehat{p}(y \mid x) = f_U\big(\widehat{Q}_{Y|X}^{-1}(y,x)\big) \det\big[\nabla_y \widehat{Q}_{Y|X}^{-1}(y,x)\big].$$

This estimator can then be used to define conformity scores. For each point $(Y_i, X_i)$ in the calibration set $\mathcal{D}_{\mathrm{cal}}$ we calculate the score $s_i = \widehat{p}(Y_i \mid X_i)$. The prediction set for a new point $X_{\mathrm{test}}$ is the superlevel set of this estimated density, where the level is calibrated to ensure coverage. If $s_{(1)} \leq \cdots \leq s_{(n)}$ are the ordered scores from the calibration set, we set the threshold $\tau = s_{(\lfloor (n+1)\alpha \rfloor)}$. Then, the HPD-style prediction region is given by:

$$\mathcal{C}_\alpha^{\mathrm{hpd}}(x) = \big\{y \in \mathcal{Y} \colon \widehat{p}(y \mid x) \geq \tau\big\}.$$

By standard arguments, this set fulfills the marginal coverage guarantee $\mathbb{P}_{(Y,X) \sim F_{YX}}(Y \in \mathcal{C}_\alpha^{\mathrm{hpd}}(X)) \geq 1 - \alpha$. Crucially, if the learned map $\widehat{Q}_{Y|X}^{-1}$ recovers the true rank map, then $\widehat{p}(\cdot \mid x)$ recovers the true conditional density, and the resulting prediction set is exactly the true HPD region.

**Related density–based approaches.** The idea of using density estimation to construct conformal sets has been exploited in recent related works. For example, in the setting with $\mathcal{Y} \subseteq \mathbb{R}$, *CD-split* partition $\mathcal{X}$ into multiple splits, leverage a conditional density estimator $\hat{f}(y \mid x)$, and perform conformal calibration in split-wise manner to improve conditional coverage (Izbicki et al., 2022). Furthermore, also in the setting with $\mathcal{Y} \subseteq \mathbb{R}$, *SPICE* learns a neural conditional density via deep splines and uses negative log-density/HPD scores to construct the conformal sets (Diamant et al., 2024).

*Remark* 5. To construct conformal sets using density estimation, the estimator of $\widehat{p}(y \mid x)$ requires the Jacobian of $\widehat{Q}_{Y|X}^{-1}$. Even if $\widehat{Q}_{Y|X}^{-1}$ approximates $Q_{Y|X}^{-1}$, $\nabla_y \widehat{Q}_{Y|X}^{-1}$ may not necessary approximate well $\nabla_y Q_{Y|X}^{-1}$. Empirically, small errors in the Jacobian can be magnified in $\det(\cdot)$, which distorts HPD superlevel sets. As shown in Section 6.1, in our experiments, $\widehat{Q}_{Y|X}$ approximated well the true quantile function. Nonetheless, we found the HDP approach of producing conformal sets empirically suboptimal w.r.t. the volume of the produced set and conditional coverage.

# G IMPORTANCE OF CONVEX POTENTIAL

The fact that the transformation is cyclically monotone is crucial for defining a statistically meaningful notion of multivariate rank. Cyclical monotonicity is the extension of monotonicity in the multidimensional setting. The definition of multivariate ranks is discussed in the pioneering work of Hallin et al. (2021) and in studies by Galichon (2018); Carlier et al. (2016; 2017; 2022); Chernozhukov et al. (2017).

### G.1 CYCLICAL MONOTONICITY

A subset $S \subseteq \mathbb{R}^d \times \mathbb{R}^d$ is said to be cyclically monotone if, for any finite collection of points $\{(\mathbf{x}_1, \mathbf{y}_1), \ldots, (\mathbf{x}_k, \mathbf{y}_k)\} \subseteq S$, the following inequality holds:

$$\langle \mathbf{y}_1, \mathbf{x}_2 - \mathbf{x}_1 \rangle + \langle \mathbf{y}_2, \mathbf{x}_3 - \mathbf{x}_2 \rangle + \cdots + \langle \mathbf{y}_k, \mathbf{x}_1 - \mathbf{x}_k \rangle \leq 0.$$

A finite subset $S = \{(\mathbf{x}_1, \mathbf{y}_1), \ldots, (\mathbf{x}_n, \mathbf{y}_n)\} \subseteq \mathbb{R}^d \times \mathbb{R}^d$ is cyclically monotone if and only if the inequality above holds for $k = n$. Equivalently, $S$ maximizes the empirical correlation $\sum_{i=1}^n \langle \mathbf{x}_i, \mathbf{y}_i \rangle$ among all pairings of $\{\mathbf{x}_1, \ldots, \mathbf{x}_n\}$ and $\{\mathbf{y}_1, \ldots, \mathbf{y}_n\}$, or, equivalently, minimizes the empirical distance $\sum_{i=1}^n \|\mathbf{y}_i - \mathbf{x}_i\|^2$. In other words, a finite subset $S$ is cyclically monotone if and only if the pairs $(\mathbf{x}_i, \mathbf{y}_i)$ form the solution to the optimal assignment problem with cost $\|\mathbf{y}_i - \mathbf{x}_i\|^2$. Rockafellar's Lemma (1966) (Rockafellar, 2015), establishes a relation between cyclical monotonicity and convex functions.

**Theorem 5** (Rockafellar (2015), Theorems 1). *The subdifferential $\partial \psi$ of a convex function $\psi$ on $\mathbb{R}^d$ enjoys cyclical monotonicity. Conversely, any cyclically monotone set $S$ of $\mathbb{R}^d \times \mathbb{R}^d$ is contained in the subdifferential $\partial \psi$ of some convex function $\psi$ on $\mathbb{R}^d$.*

**Theorem 6** (Brenier (1991)). *Let $\mu$ and $\nu$ be two probability measures on $\mathbb{R}^d$, with $\mu$ absolutely continuous with respect to the Lebesgue measure. Let $T : \mathbb{R}^d \to \mathbb{R}^d$ be a measurable map that pushes forward $\mu$ onto $\nu$, i.e.*

$$T_{\#}\mu = \nu.$$

*Then, there exists a unique (up to $\mu$-almost everywhere equality) cyclically monotone map $\nabla \varphi : \mathbb{R}^d \to \mathbb{R}^d$, where $\varphi$ is a convex function, such that $T = \nabla \varphi \circ S$ with $S : \mathbb{R}^d \to \mathbb{R}^d$ a measure-preserving map, that is $S_{\#}\mu = \mu$. Moreover, $\nabla \varphi$ is the optimal transport map pushing $\mu$ forward to $\nu$ for the quadratic cost $c(x, y) = \frac{1}{2}\|x - y\|^2$, and it minimizes $\int_{\mathbb{R}^d} \|T(x) - x\|^2 \, d\mu(x)$ among all measurable maps satisfying $T_{\#}\mu = \nu$.*

Brenier's theorem states that any measurable transport map $T$ can be factorized as:

(i) A measure-preserving rearrangement $S$, which redistributes mass within $\mu$ without altering it.
(ii) A cyclically-monotone map $\nabla \varphi$, which moves $\mu$ optimally (in the quadratic sense) to $\nu$.

When defining multivariate quantiles, we learn the cyclically monotone pushforward between the data distribution $\nu$ over $\mathbb{R}^d$ and the standard normal Gaussian distribution denoted $\mu$ (we could take the uniform distribution on the disk or the multivariate normal with 0 mean and identity covariance). We assume that $\nu$ is absolutely continuous w.r.t. the Lebesgue measure.

Assume that we stake MVN(0,I), we obtain the $(1 - \alpha)$ coverage sets, we take balls of radius $\rho_\alpha$; where $\rho_\alpha$ is the $1 - \alpha$ quantile of a $\chi^2$ distribution with $d$ degrees of freedom and compute their pre-images under this map.

If we use a transform which is not cyclically monotone, the regions that we will obtain using this method will of course have the correct coverage, but might have wild shapes, because the measure preserving transform (appearing in Brenier's polar factorization) can be arbitrarily complex. Indeed, there infinitely many pushforward map between $\mu$ and $\nu$, but only the cyclically monotone pushforward allows to define a quantile function and the associated ranks and signs that are meaningful.

### G.2 EXAMPLE OF NON CYCLICALLY-MONOTONE MAP

We illustrate this phenomenon in dimension $d = 2$.

Assume that the data distribution $\nu$ is multivariate normal with a non-singular covariance matrix $\Gamma$. We map it to the multivariate distribution with density $\frac{d\mu}{d\,\mathrm{Leb}}(u) = MVN(u; 0, I)$, where Leb is the Lebesgue measure.

The cyclically monotone transform is $\nabla \Phi(u) = \Gamma^{-1/2} u$, where $\Gamma^{-1/2}$ is the Hermitian root-square of $\Gamma$ (and $\Phi(u) = (1/2)u^\top \Gamma^{-1/2} u$ is a convex function). The Brenier polar factorization theorem shows that any pushforward $\nu \to \mu$ can be decomposed as $T = \nabla \Phi \circ \sigma$, where $\sigma$ is a measure-preserving map for $\mu = N(0, I)$. This measure preserving maps is not cyclically monotone and therefore might affect the ranks used in the construction.

There are many ways to construct measure preserving maps of $\text{MVN}(0, I)$. We provide such a construction in $\mathbb{R}^2$. Let $(x(t), y(t)_{t \geq 0}$ a two dimensional random process which evolves according to the the ODE:

$$\dot{x}(t) = v_{t,x}(x(t)), \quad \dot{y}_t = v_{t,y}(y(t)),$$

where $v_t = (v_{t,x}, v_{t,y})$ is the velocity field. Denote by $p_t$ the pdf of $(x(t), y(t))$ at time $t$. The continuity equation shows that

$$\nabla_t p_t + \nabla \cdot (p_t v_t) = 0$$

The density is invariant under the flow of the ODE is $\nabla \cdot (p_t v_t) = 0$ for all $t \geq 0$, in which case $p_t = p_0$ for all $t \geq 0$. Denote $w_t = p_t v_t$. A canonical way to ensure $\nabla \cdot w_t = 0$ is to set $w_{t,x} = \partial_y \Psi(x, y)$ and $w_{t,y} = -\partial_x \Psi(x, y)$, where $\Psi$ plays the role of an Hamiltonian. If we start from $p_0$ the pdf of $MVN(0, I)$, the flow of the ODE

$$\dot{x}(t) = \frac{1}{p_0}(x(t), y(t))\partial_y \Psi(x(t), y(t)) \quad \dot{y}(t) = -\frac{1}{p_0}(x(t), y(t))\partial_x \Psi(x(t), y(t))$$

is measure preserving for $p_0$, for any $t > 0$. In other words, if we define $(x(t), y(t)) = F_{0 \to t}(x(0), y(0))$ the flow of the solutions of the ODE, $F_{0 \to t} \# p_0 = p_0$. Along the flow of the solutions,

$$\frac{d}{dt}\Psi(x(t), y(t)) = \partial_x \Psi(x(t), y(t))\dot{x}(t) + \partial_y \Psi(x(t), y(t))\dot{y} = 0$$

so the Hamiltonian $\Psi$ is constant along the orbit: $\Psi(x(t), y(t)) = C$. We illustrate this with the mixing

$$\Psi(x, y) = (x^2 + y^2)^{3/2}(1 - x^2 - y^2)y$$

Results are presented in Figure 3. In (A., B.), we show the level set obtained by applying the cyclically monotone transform. In (C.,D.) we show the level sets obtained using a non cyclically monotone transformation $T$, obtained by integrating the Hamiltonian flow until tile $t = 1.5$. (E., F.) show empirical density estimate from $10^6$ points, sampled from two different operators.

It is clear that a transformation that is not cyclically monotone does not define a notion of ranks similar to the one described in Hallin et al. (2021), and cannot be used to construct a conformalization procedure similar to ours.

## H    DETAILED EXPERIMENTAL RESULTS

### H.1    OPTIMAL TRANSPORT METRICS

- **Wasserstein distances.** We compute Wasserstein-2 and Sliced Wasserstein distances using the *POT* library Flamary et al. (2021).

- **KDE-L1.** To estimate the $L^1$ distance between kernel density estimators, we draw 1000 samples from both $Q_{Y|X}^{-1}$ and its approximation $\widehat{Q}_{Y|X}^{-1}$. We then fit Gaussian kernel density estimates to each sample set and report the average pointwise $L^1$ difference between the two densities, evaluated at points drawn from $Q_{Y|X}^{-1}$.

- **KDE-KL.** The Kullback–Leibler divergence is computed following the same procedure as KDE-L1. We report the average pointwise KL divergence between the fitted densities at points drawn from $Q_{Y|X}^{-1}$.

- **L2-UV.** To compute the unexplained variance ratio, we sample $n_u$ points from $u_{\text{test}} \sim F_U$ and $n_x$ points from $x_{\text{test}} \sim F_X$. The L2-UV distance is then defined as

$$\frac{1}{n_x + n_u} \sum_{x_{\text{test}}, u_{\text{test}}} \frac{\|Q_{U|X}(u_{\text{test}}, x_{\text{test}}) - \widehat{Q}_{U|X}(u_{\text{test}}, x_{\text{test}})\|_2}{\left\|\frac{1}{n_u} \sum_{u_{\text{test}}} Q_{U|X}(u_{\text{test}}, x_{\text{test}}) - Q_{U|X}(u_{\text{test}}, x_{\text{test}})\right\|_2}.$$

### H.2    OPTIMAL TRANSPORT EXPERIMENTS DATASETS

**Banana Dataset.**    This dataset is largely used in vector quantile estimation for testing the non-linearity of estimators. It was introduced in (Feldman et al., 2023) and used in (Carlier et al., 2017; Rosenberg et al., 2023). It represents a banana-shaped random variable in $\mathbb{R}^2$, changing its position

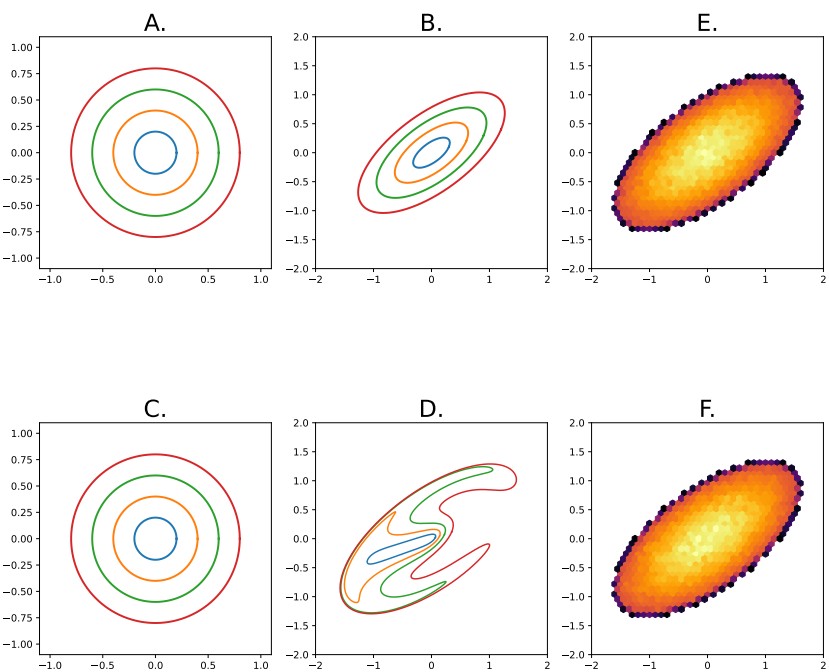

Figure 3: Two pushforward operators. (A., B.) Level sets of latent distribution chosen to be standard Gaussian. (B.) Pushforward of level sets by cyclically monotone operator. (D.) Pushforward of level sets by non cyclically operator, generated by flow. (E., F.) Empirical Density Estimates, based on points sampled from cyclically monotone operator (E.) and non cyclically monotone operator (F.).

and skewness based on latent random variable from $\mathbb{R}^1$. Data generative process can be described as:

$$X \sim \mathcal{U}[0.8, 3.2], \quad Z \sim \mathcal{U}[-\pi, \pi], \quad \varphi \sim \mathcal{U}[0, 2\pi], \quad r \sim \mathcal{U}[-0.1, 0.1],$$

$$\hat{\beta} \sim \mathcal{U}[0, 1]^k, \quad \beta = \frac{\hat{\beta}}{\|\hat{\beta}\|_1},$$

$$Y_0 = \tfrac{1}{2}\left(-\cos(Z) + 1\right) + r\sin(\varphi) + \sin(X),$$

$$Y_1 = \frac{Z}{\beta X} + r\cos(\varphi),$$

$$\mathbf{X} = X, \mathbf{Y} = \begin{bmatrix} Y_0 \\ Y_1 \end{bmatrix}.$$

We take $\mathbf{X}$ as and $\mathbf{Y}$ as observed random variables.

**Rotating Star.** This dataset is inspired by (Rosenberg et al., 2023) rotating star example. Observed random variable represents a three point star in $\mathbb{R}^2$ that rotates based on latent variable from $\mathbb{R}$. Data generative process can be described as:

$$(u_0, u_1) \sim \mathcal{N}(0, I), \quad X \sim \mathcal{U}\left[0, \frac{2}{3}\right],$$

$$\theta = \arctan\left(\frac{u_1}{u_0}\right), \quad s(\theta) = 1 + 3\cos(3\theta),$$

$$\mathbf{R}(\varphi) = \begin{bmatrix} \cos\varphi & -\sin\varphi \\ \sin\varphi & \cos\varphi \end{bmatrix},$$

$$\mathbf{Y} = \mathbf{R}(\varphi)\left(s(\theta)u_0, s(\theta)u_1\right)^\top, \mathbf{X} = X,$$

where $\varphi$ is a rotation angle. We take $\mathbf{X}, \mathbf{Y}$ as observed variables.

**Glasses.** This dataset is introduced in (Brando et al., 2022). It represents two modal distribution, where random variable is in $\mathbb{R}$. With $X \sim \mathcal{U}[0, 1]$, data generative process can be described as:

$$z_1 = 3\pi X, \quad z_2 = \pi(1 + 3X), \quad \epsilon \sim \text{Beta}(\alpha = 0.5, \beta = 1),$$

$$Y_1 = 5\sin(z_1) + 2.5 + \epsilon, \quad Y_2 = 5\sin(z_2) + 2.5 - \epsilon,$$

$$\gamma \sim \text{Categorical}(0, 1),$$

$$\mathbf{Y} = (1 - \gamma)Y_1 + \gamma Y_2.$$

We take $\mathbf{X}, \mathbf{Y}$ as observed variables.

**Neal's funnel distribution.** The classical funnel distribution (Neal, 2003) is defined on $\mathbb{R}^{d+1}$ as

$$v \sim \mathcal{N}(0, \sigma^2), \qquad x_i \mid v \sim \mathcal{N}\left(0, \exp(v)\right), \quad i = 1, \ldots, d,$$

so that the joint density of $(v, x_1, \ldots, x_d)$ is

$$p(v, x) = \frac{1}{\sqrt{2\pi\sigma^2}} \exp\left(-\frac{v^2}{2\sigma^2}\right) \prod_{i=1}^{d} \frac{1}{\sqrt{2\pi e^v}} \exp\left(-\frac{x_i^2}{2e^v}\right).$$

For large negative values of $v$, the conditional variance of the $x_i$'s shrinks, yielding a narrow region (the "neck" of the funnel), whereas large positive $v$ produces very diffuse $x_i$'s (the "mouth"). This strong nonlinearity makes the distribution challenging for MCMC methods.

**Multidimensional funnel.** A natural generalization introduces a $k$-dimensional scale vector $v = (v_1, \ldots, v_k)$ with

$$v_j \sim \mathcal{N}(0, \sigma^2), \qquad x_{j,\ell} \mid v_j \sim \mathcal{N}\left(0, \exp(v_j)\right), \quad \ell = 1, \ldots, m,$$

so that each $v_j$ controls a block of $m$ Gaussian variables. The joint distribution then lives in dimension $k(1 + m)$ and exhibits multiple funnel directions simultaneously. This high-dimensional geometry is frequently used as a stress test for MCMC and normalizing flow methods.

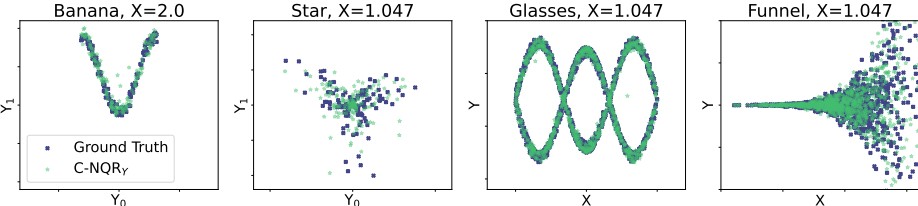

Figure 4: Example of points sampled from reference distribution of all the datasets we study and points sampled from approximation constructed by C-NQR$_Y$ method.

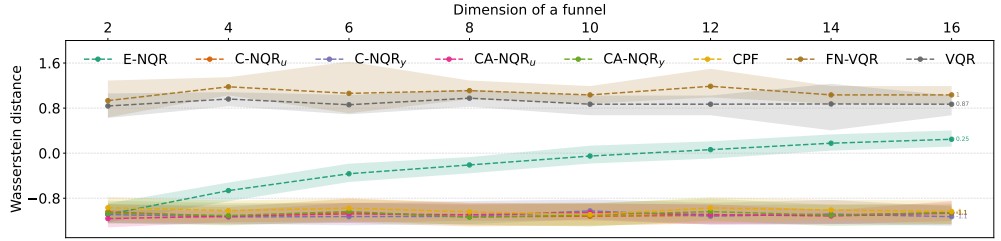

Figure 5: Log values of S-W2 on Neal's funnel distribution. We scale the dimension of a funnel from 2 to 16.

## H.3  EXTENDED RESULTS ON OPTIMAL TRANSPORT

Here we give extended results on our internal experiments of a proposed quantile regression estimate.

**Generative Model** In Figure 4 we plot examples of samples from different datasets. Dark blue points are sampled from underlying ground truth generative model. Green points represent samples from C-NQR$_Y$ generative model.

**Scalability** To evaluate scalability, Figure 5 reports the S-W2 metric on Neal's Funnel as the target dimension increases from 2 to 16. We observe that methods leveraging explicit $c$-transform computation (NQR$_U$, NQR$_Y$, AC-NQR$_U$, AC-NQR$_Y$) scale robustly with dimensionality and maintain high generative accuracy, while entropically relaxed variants (EC-NQR) fail to scale effectively. Furthermore, the proposed framework consistently achieves superior S-W2 metric compared to prior approaches (Rosenberg et al., 2023; Carlier et al., 2016), demonstrating its performance in high-dimensional quantile estimation.

**Full set of metrics** Lastly we present full set of metrics for studied datasets:

- Metrics for Glasses dataset is accessible at Figure 11.
- Metrics for convex potential, that was trained on Glasses dataset can be found at Figure 12
- Metrics for Star dataset is accessible at Figure 9.
- Metrics for convex potential, that was trained on Star dataset can be found at Figure 10
- Full set of metrics for Banana dataset is accessible at fig. 7. Metrics for convex potential, that was trained on Banana dataset can be found at fig. 8.

## H.4  DETAILED RESULTS OF THE CONFORMAL PREDICTION EXPERIMENTS

We present more detailed results on conditional coverage on real datasets, involving more variations of our methods and more nominal levels $\alpha$.

**Methods.** We include the HPD variant of our method as well as models estimating either the forward (U) or the inverse (Y) quantile map.

For methods labeled with Y, we model the function $\psi$ with a neural network and have $\widehat{Q}_{Y|X}^{-1}(y, x) = \nabla_y \psi(y, x)$. For methods labeled with U we model function $\varphi$ and get $\widehat{Q}_{Y|X}(y, x) = \nabla_u \varphi(u, x)$.

Method Quantile corresponds to using the Monge-Kantorovich rank to construct the predictive regions, assuming that we have found exactly the mapping to the reference standard multidimensional normal distribution. In this particular case, the squared ranks follow the Chi-square distribution and the corresponding radius for the construction of the pullback-type prediction set can be found exactly.

The methods labeled with RF correspond to fitting our model to the residuals of $s = y - \hat{f}(x)$ of a base Random Forest predictor $\hat{f}$. Base predictor uses 25% of the training data and remainder is used to train our model.

**Implementation details.** For baseline methods we use the original authors implementation, where available and their suggested values for hyperparameters. For our methods, we select the number of parameters for neural networks to be roughly 10% of the number of training samples. We tune the other hyperparameters for each dataset using a separate data split and utilize the mean coverage error of the pullback sets at different levels of $\alpha$ as a performance measure. All experiments were replicated using 10 random splits of the data into training, calibration, and test parts.

**Discussion.** The Quantile method fails to achieve the nominal levels of conditional coverage, which suggests that a supporting measure like conformal prediction is indeed required. Unfortunately, HPD approaches do not perform well on many occasions, proving that density estimation in multiple dimensions is still a difficult to solve problem.

Using a base model and fitting quantile regression to the residuals instead of directly $Y$ provides less variable results, but does not always improve performance of our methods.

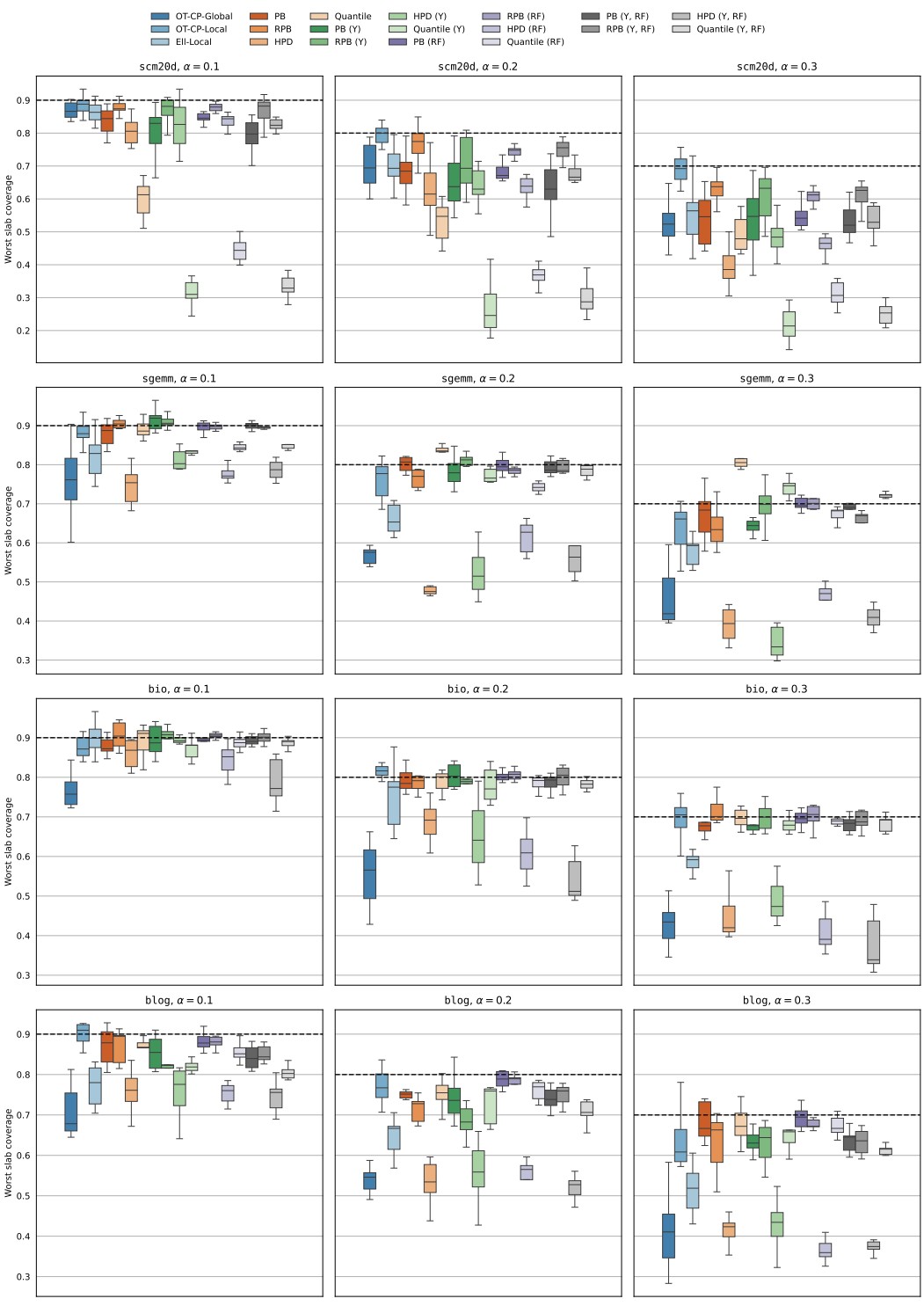

Figure 6: Worst slab coverage at different nominal miscoverage $\alpha$ levels for conformal prediction methods, achieved on large datasets.

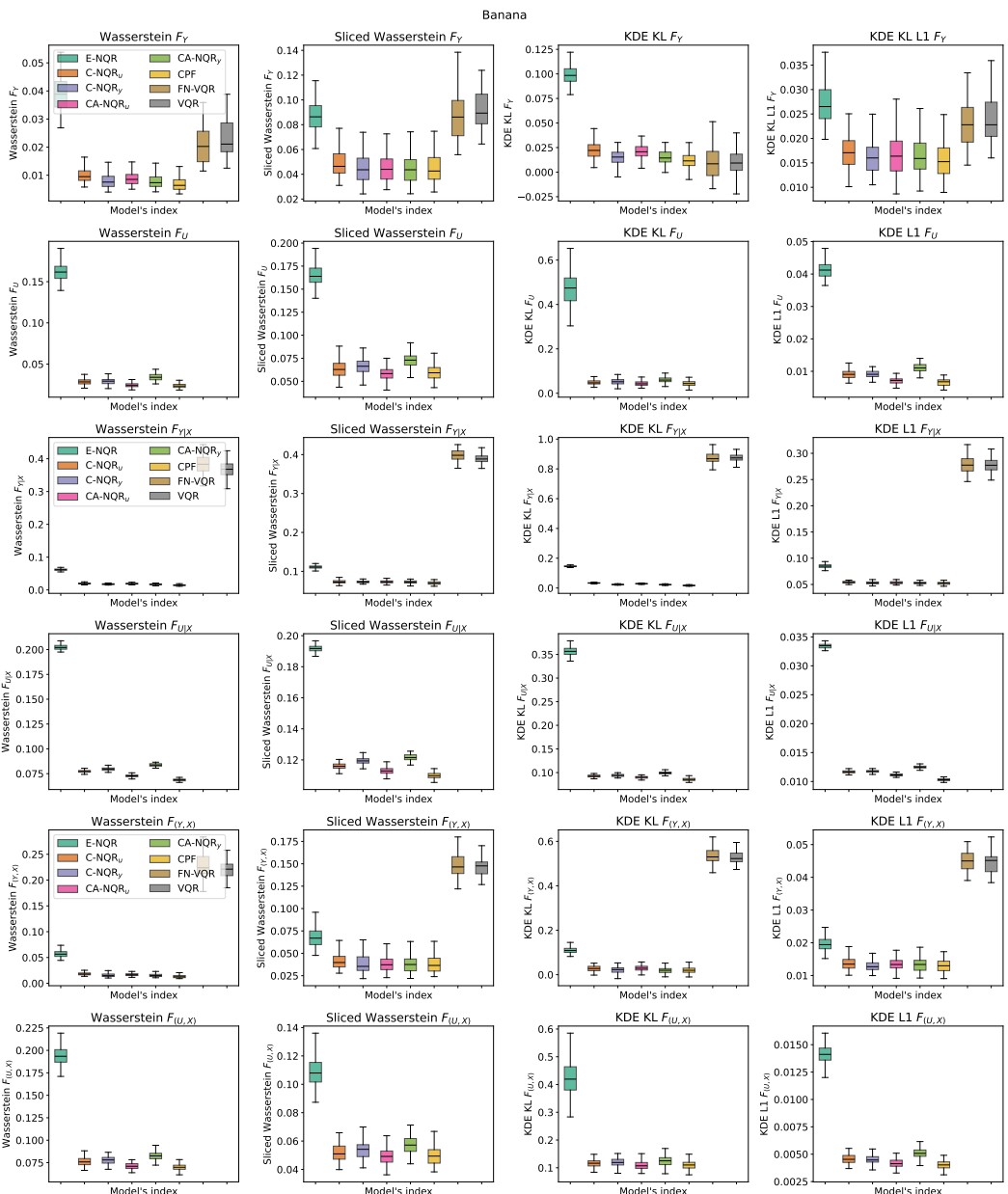

Figure 7: Full set of metrics for Banana dataset.

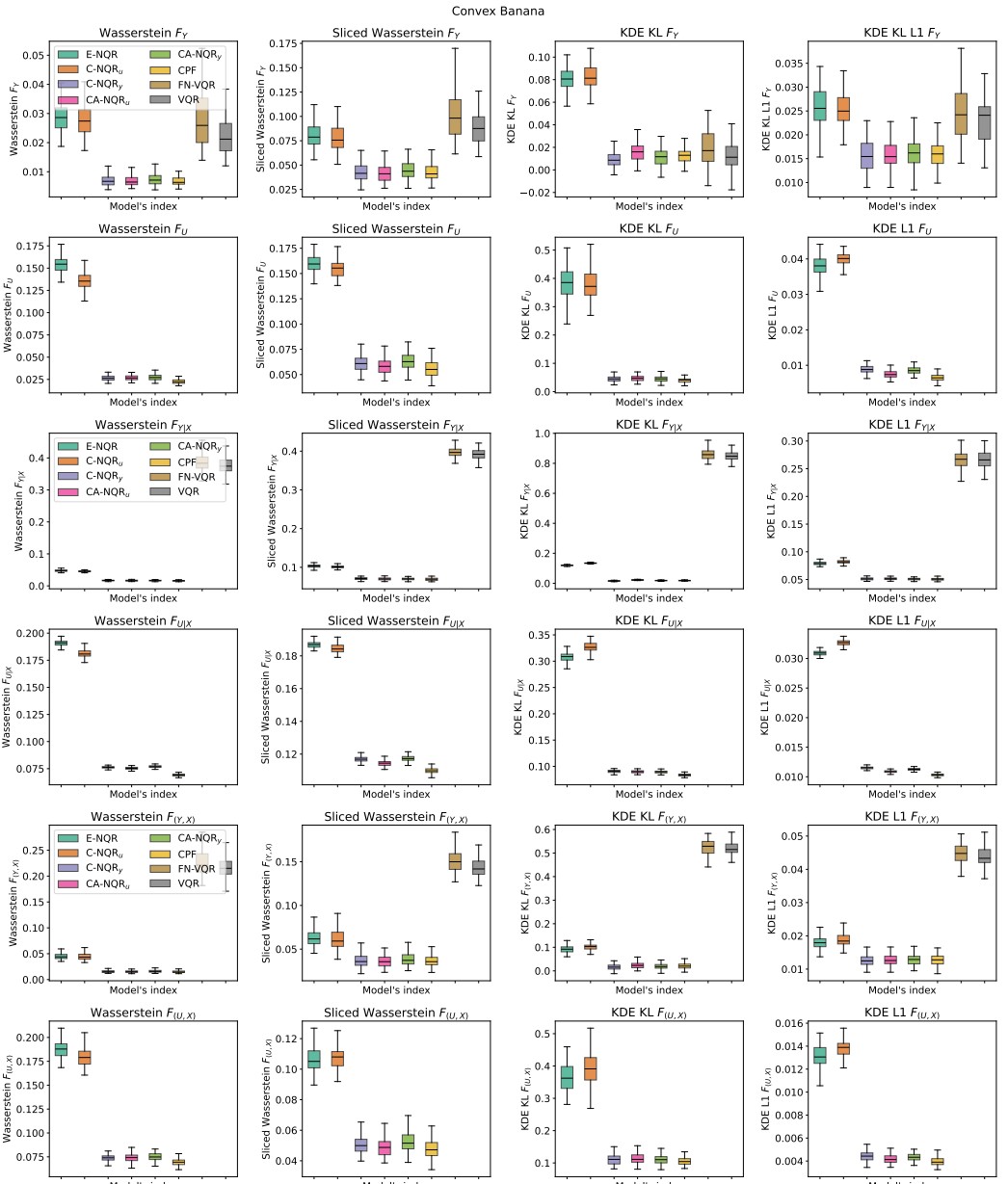

Figure 8: Full set of metrics for Banana dataset.

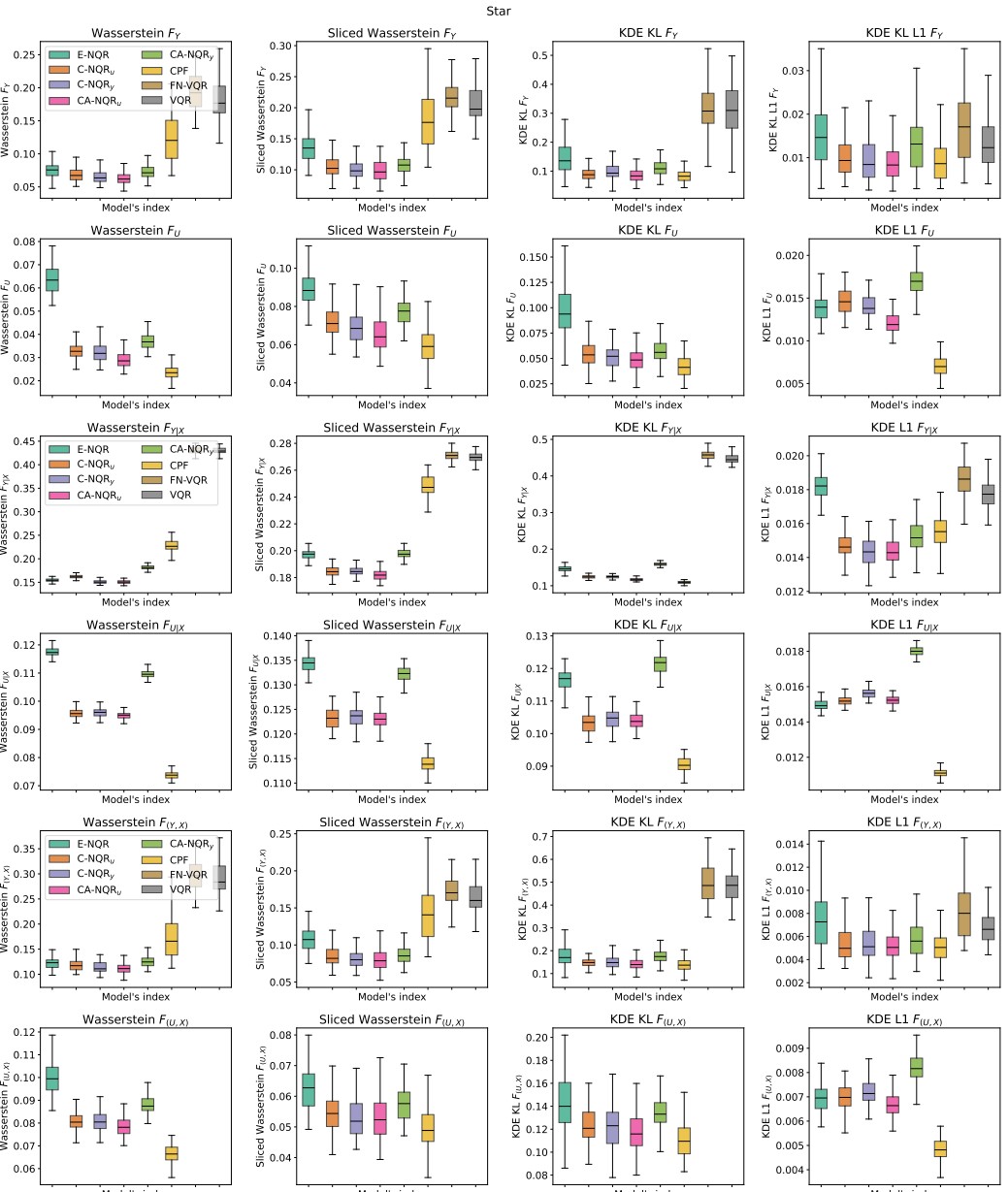

Figure 9: Full set of metrics for Star dataset.

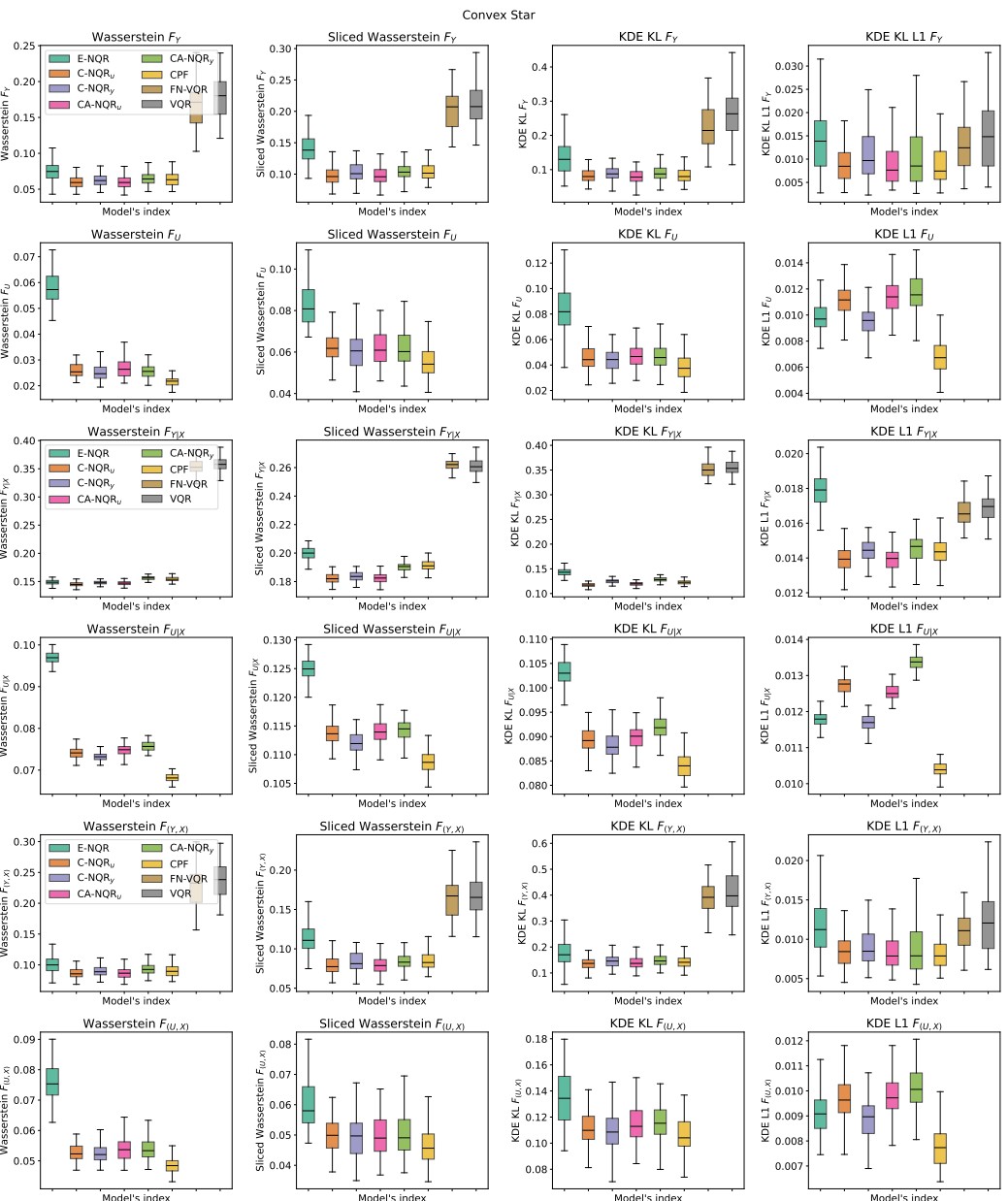

Figure 10: Full set of metrics for Convex Star dataset.

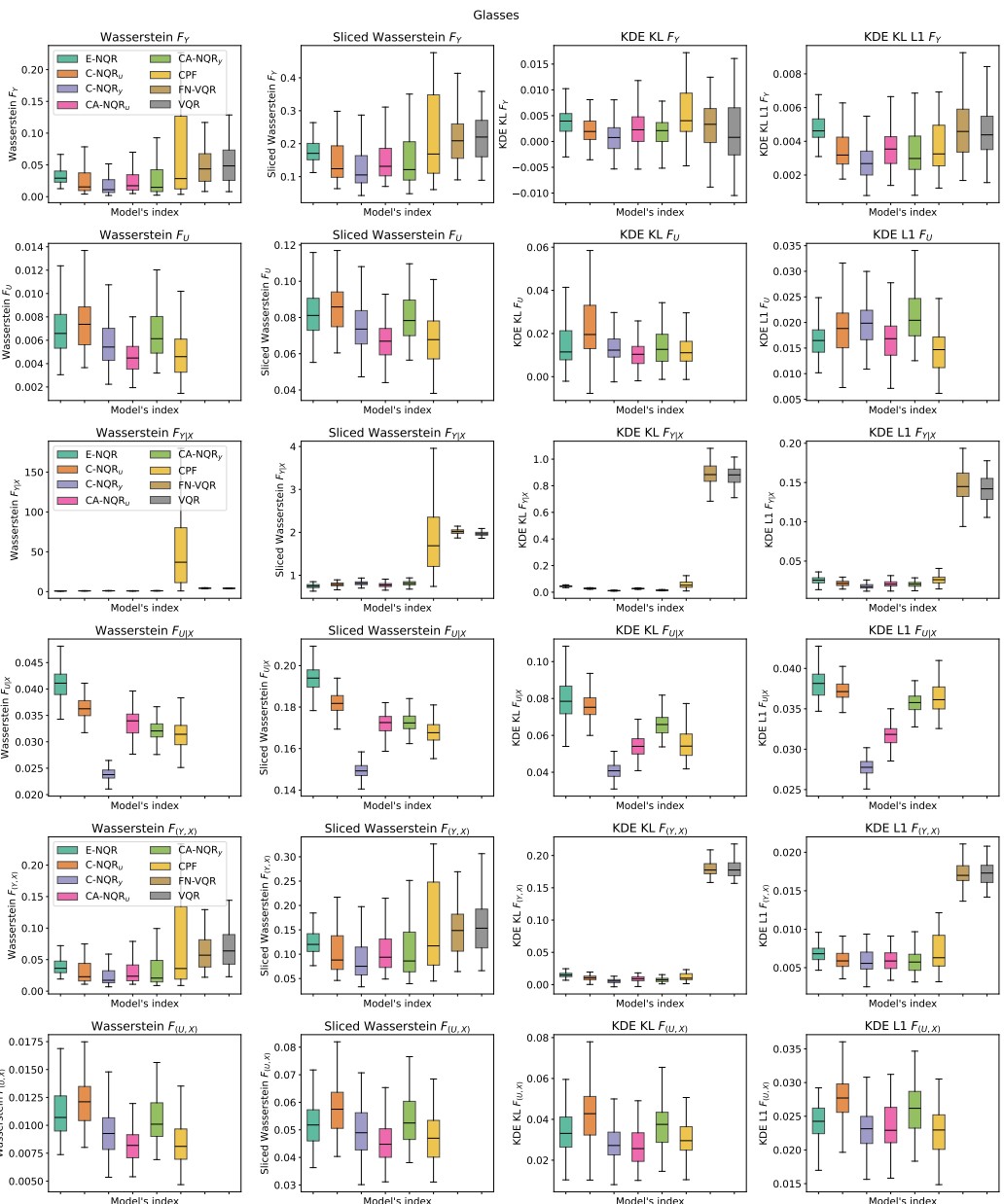

Figure 11: Full set of metrics for Glasses dataset.

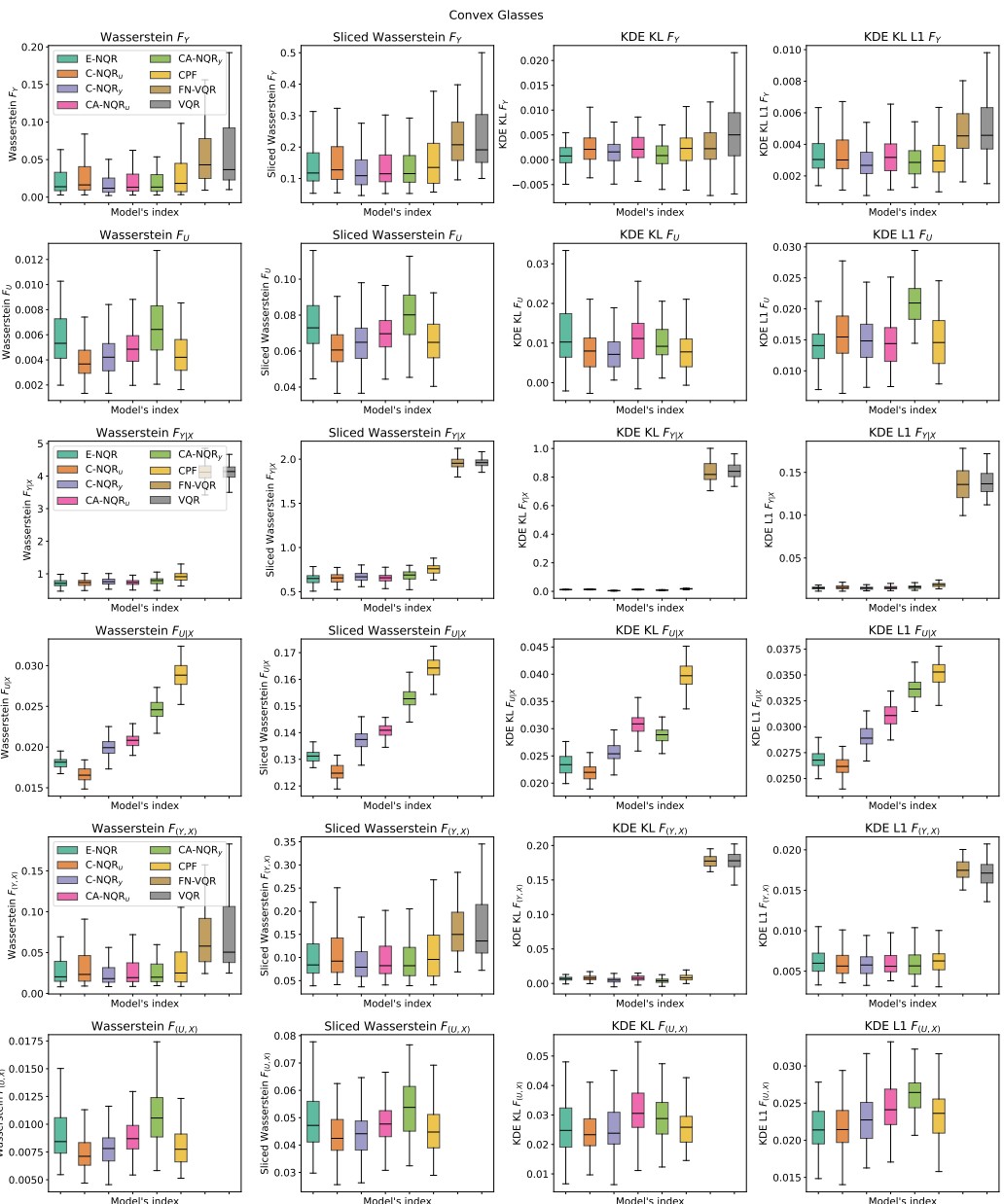

Figure 12: Full set of metrics for Convex Glasses dataset.

