# OpenReview forum: "Neural Optimal Transport Meets Multivariate Conformal Prediction"
_ICLR.cc/2026/Conference — ICLR 2026 Poster_

### Official Review · Reviewer_Rw7A · 2025-10-25

**Soundness:** 1
**Presentation:** 2
**Contribution:** 2
**Rating:** 2
**Confidence:** 4

**Summary:**

This work proposed a parametric approach to estimate conditional vector quantiles, where the push-forward is parameterized through the gradient of a (partially) convex neural network. An amortized strategy is adopted to reduce computation (by following the unconditional case) and an application to split conformal prediction is discussed. Experiments on small-scale synthetic dataset and regression benchmarks confirm the effectiveness of the proposed methods.

**Strengths:**

- a neural approach to estimate vector quantiles

- a more versatile approach for conformal prediction

**Weaknesses:**

- limited novelty: the majority of the contributions are a straightforward extension of prior works, with no significantly new ingredients

- possible gaps in Theorem 3: the proof on Line 1244-1246 needs more explanation. The proof seems to rely on the assumption that HPD superlevel sets are "optimal"? In Remark 3, how is the displayed equation in Theorem 3 satisfied? Here the left-hand side does not depend on y while the right-hand side does. Is this assumption too strong?

- writing can be improved. A lot of details are deferred to the appendix, including the description of the proposed methods (Algorithms 1, 2 and 3). At times, it is hard to appreciate what are the significant modifications on the authors' side.

**Questions:**

My main concerns are on the significance and correctness:

- most results appear to be rather straightforward adaptions of prior works (Sections 4 and 5). This is not to say there isn't anything new, but the writing makes it feel like these are rather incremental changes.

- I was not able to verify the proof of Theorem 3 and I think its assumptions may be too strong. In particular, for the elliptical example the authors need to verify the displayed equation in Theorem 3.

I was intrigued by the authors claim that "Neural OT as we propose here have never been scaled" (Line 321) and "never from a unique joint sampling using the framework of Carlier et al. as proposed in our work" (Line 340). A quick google search revealed the following work "Conditional Generative Quantile Networks via Optimal Transport" (https://openreview.net/forum?id=BBxeo2Vuvbq), which seemed to be based on Carlier et al. and already proposed a neural approach for conditional vector quantile estimation. This latter work did not discuss conformal prediction but perhaps is still very relevant.

Other comments:

Line 148: "In the multivariate setting, monotonicity requires the map to be the gradient of a convex function." How is monotonicity defined in the multivariate setting? Why is it needed and why does it require the map to be the gradient of a convex function?

Figure 3, right plot: the y-axis (log V)/d can be as low as -0.7 (even -5 for the second plot). For moderately large d this means the volume of the conformal set is exceedingly small. Is this expected?

Line 467: the resulting response dimensions are 16, 4, 2, and 2. The potential based OT approach is known to be difficult to scale (due to the evaluation of the conjugate potential). The authors' experiments seem to confirm this observation. How high the dimension can be for the proposed approach to remain effective?

---

> ### Author Response · Authors · 2025-11-20
> **Novelty**
>
> Our approach to conditional quantile modeling is **one of the main contributions of this paper**.
>
> We note that there are two crucial difference between our approach and [Sun *et al.*](https://openreview.net/forum?id=BBxeo2Vuvbq), namely **modeling** and **solving** for the convex potentional.
>
> **Modeling.** Indeed, both of our works start with the same shared dual optimization problem (corresponding to Eq. (2) in our paper).
> $$\min_{\varphi , \psi }\mathbb{E} \left[\varphi(V,X)\right]+\mathbb{E}\left[ \psi(Y,X)\right], $$
> where both $\psi$ and $\varphi$ are taken to be convex functions in their first parameter ($V$ and $Y$ above) and $\psi$ is the c-transform of $\varphi$ denoted $\varphi^*$ (semi-dual formulation).
>
> [Sun *et al.*](https://openreview.net/forum?id=BBxeo2Vuvbq) further assume the following structure
>
> $$\varphi(V,X) = \bar{\varphi}(V) + b(V)^T f(X),$$
>
> where different structural constraints may then be enforced on $\bar{\varphi}$ and $b$ to insure that $\varphi$ is convex in $V$. Similar approach is chosen in [Continuous Vector Quantile Regression, Vedula *et al.*](https://openreview.net/forum?id=DUZbGAXcyL). Both [Sun *et al.*](https://openreview.net/forum?id=BBxeo2Vuvbq) and [Vedula *et al*](https://openreview.net/forum?id=DUZbGAXcyL) model the potential as neural networks, similar to us.
>
> In our analysis, we impose no restrictive assumptions on the function $\varphi$. In particular, we do not assume that $\varphi$ possesses an affine structure with respect to a feature vector derived from the dependent variables. We employ a PICNN architecture specifically constructed to ensure convexity with respect to the variable $V$ for any fixed $X$. The network design is inspired by normalizing flow models with residual connections, yet its parametrization is explicitly constrained to preserve the required convexity structure.
>
> The network used in [Sun *et al.*](https://openreview.net/forum?id=BBxeo2Vuvbq) and [Vedula *et al.*](https://openreview.net/forum?id=DUZbGAXcyL) is therefore much less expressive for learning the dependence in $x$. See the experiments below.
>
>
> **Solving.** The second *main difference* between our approach and [Sun *et al.*](https://openreview.net/forum?id=BBxeo2Vuvbq) and [Vedula *et al.*](https://openreview.net/forum?id=DUZbGAXcyL) is in handling the inner optimization to compute the c-transform of $\varphi$.
>
> Since we parameterize $\varphi$ using PICNN, so this inner optimization step is actually (strongly) convex and can be solved efficiently using a combination of Amortized optimization used to initialize LBFGS to solve the semi dual problem:
> $$\min_{\varphi}\mathbb{E} \left[\varphi(V,X)\right]+\mathbb{E}\left[ \varphi^*(Y,X)\right].$$
>
>
> [Sun *et al.*](https://openreview.net/forum?id=BBxeo2Vuvbq) and [Vedula *et al.*](https://openreview.net/forum?id=DUZbGAXcyL) rely on empirical sampling to solve the inner maximization problem; that is, they sample realizations of $(V, Y, X)$ and set the inner optimization problem to the maximal value:
> $$\mathbb{E}\left[\bar{\varphi}(V) + b(V)^T f(X) +\sup_{v} v^T Y -\bar{\varphi}(v) - b(v)^T f(X)\right].$$
>
> While [Sun *et al.*](https://openreview.net/forum?id=BBxeo2Vuvbq) argue that the law of large numbers justifies this sampling approximation, in practice, the number of samples required to accurately estimate this maximum is intractable for high-dimensional settings.
>
> [Vedula *et al.*](https://openreview.net/forum?id=DUZbGAXcyL) choose to replace the inner maximization loop with a softmax operator, which cannot be fully theoretically justified in the setting they are studying and suffers from the same scaling problem as [Sun *et al.*](https://openreview.net/forum?id=BBxeo2Vuvbq).
>
> Therefore, our method for estimating the conditional quantile function is *fundamentally different* from those of [Sun *et al.*](https://openreview.net/forum?id=BBxeo2Vuvbq) and [Vedula *et al.*](https://openreview.net/forum?id=DUZbGAXcyL), and it consistently outperforms both approaches across the numerous benchmarks we have considered. The use of a *PICNN architecture*, rather than an affine function of hand-crafted features, provides a *more flexible and expressive representation* of the conditional quantile function. Moreover, our approach yields *explicit formulations of the pushforward and pullback operators*, which are essential for constructing conformal regions and for the modeling framework presented in our conformal prediction component.

---

> > ### Comment · Reviewer_Rw7A · 2025-11-24
> > **Re: Novelty**
> >
> > Thanks for the detailed response. I think it is fair to say there is some difference from prior works but the claims "Neural OT as we propose here have never been scaled" (Line 321) and "never from a unique joint sampling using the framework of Carlier et al. as proposed in our work" (Line 340) should be tuned down a bit, and the relevant works should be cited and discussed. Please do so in your revision.

---

> > > ### Author Response · Authors · 2025-11-25
> > > **State of the art**
> > >
> > > In the supplementary material, we provided a state-of-the-art review that we believe offers a comprehensive overview of optimal transport methods for conditional transport and quantile mapping. This area has received significant attention.
> > >
> > > Carlier’s work introduced a regression-based approach with a relatively strong structural constraint on the quantile function. This framework has since been adopted and extended by several authors, for example, through nonlinear variants that learn feature representations. We believe this point is made clearly in our discussion.
> > >
> > > Regarding scalability, our intention was to note that discrete optimal transport, which underlies many of these approaches, does not scale efficiently to high-dimensional settings – a limitation well known in the field.
> > >
> > > If the wording of that sentence seems too assertive, we are happy to revise it. We fully recognize that the topic is well developed, but within the space constraints of the main text, we can only devote about half a column to reviewing prior work. This is why we included a more detailed discussion in the supplementary material.

---

> > > > ### Author Response · Authors · 2025-11-25
> > > > **Cyclically Monotone**
> > > >
> > > > A subset $S \subseteq \mathbb{R}^d \times \mathbb{R}^d$ is said to be **cyclically monotone** if, for any finite collection of points  $\{(\mathbf{x}_1, \mathbf{y}_1), \ldots, (\mathbf{x}_k, \mathbf{y}_k)\} \subseteq S$, the following inequality holds:
> > > > $$
> > > > \langle \mathbf{y}_1, \mathbf{x}_2 - \mathbf{x}_1 \rangle + \langle \mathbf{y}_2, \mathbf{x}_3 - \mathbf{x}_2 \rangle + \cdots +
> > > > \langle \mathbf{y}_k, \mathbf{x}_1 - \mathbf{x}_k \rangle
> > > > \leq 0.
> > > > $$
> > > >
> > > > A finite subset  $$S = \{(\mathbf{x}_1, \mathbf{y}_1), \ldots, (\mathbf{x}_n, \mathbf{y}_n)\} \subseteq \mathbb{R}^d \times \mathbb{R}^d$$
> > > >
> > > > is cyclically monotone **if and only if** the inequality above holds for $k = n$. Equivalently, $S$ maximizes the empirical correlation $\sum_{i=1}^n \langle \mathbf{x}\_i, \mathbf{y}\_i \rangle$ among all pairings of $\{\mathbf{x}\_1, \ldots, \mathbf{x}\_n\}$ and $\{\mathbf{y}\_1,\ldots,\mathbf{y}\_n\}$, or, equivalently, **minimizes** the empirical distance $\sum_{i=1}^n \|\mathbf{y}_i - \mathbf{x}_i\|^2$.
> > > >
> > > >
> > > > In other words, a finite subset $S$ is cyclically monotone **if and only if** the pairs $(\mathbf{x}_i, \mathbf{y}_i)$ form the solution to the **optimal assignment problem** with cost $\|\mathbf{y}\_i - \mathbf{x}\_i\|^2$.
> > > > Rockafellar's Lemma (1966) establishes a relation between cyclical monotonicity and convex functions.
> > > >
> > > > #### (Rockafellar 1966, Theorem 1).
> > > > The subdifferential $\partial \psi$ of a convex function $\psi$ on $\mathbb{R}^d$ enjoys cyclical monotonicity. Conversely, any cyclically monotone set $S$ of $\mathbb{R}^d \times \mathbb{R}^d$ is contained in the subdifferential $\partial \psi$ of some convex function $\psi$ on $\mathbb{R}^d$.
> > > >
> > > > An interesting property is that any push-forward of a measure that admits a density with respect to the Lebesgue measure can be viewed as the composition of a cyclically monotone transformation and a measure-preserving transformation.
> > > >
> > > > #### Brenier’s Polar Factorization Theorem
> > > >
> > > > Let $\mu$ and $\nu$ be two probability measures on $\mathbb{R}^d$, with $\mu$ **absolutely continuous** with respect to the Lebesgue measure.
> > > > Let $T : \mathbb{R}^d \to \mathbb{R}^d$ be a measurable map that **pushes forward** $\mu$ onto $\nu$, i.e. $T\_{\star}\mu=\nu$.
> > > >
> > > >
> > > > Then, there exists a **unique** (up to $\mu$-almost everywhere equality) **cyclically monotone map**  $\nabla \varphi : \mathbb{R}^d \to \mathbb{R}^d$, where $\varphi$ is a convex function, such that $T = \nabla \varphi \circ S$ with $S : \mathbb{R}^d \to \mathbb{R}^d$ a **measure-preserving map**, that is $S_{\star} \mu = \mu$. Moreover, $\nabla \varphi$ is the **optimal transport map** pushing $\mu$ forward to $\nu$ for the **quadratic cost** $c(x, y) = \frac{1}{2} \|x - y\|^2$, and it minimizes $\int_{\mathbb{R}^d} \|T(x) - x\|^2 \, d\mu(x)$ among all measurable maps satisfying $T_{\star} \mu = \nu$.
> > > >
> > > > Brenier’s theorem states that any measurable transport map $T$ can be **factorized** as:
> > > > - a **measure-preserving rearrangement** $S$, which redistributes mass within $\mu$ without altering it, and
> > > > - a **cyclically-monotone map** $\nabla \varphi$, which moves $\mu$ optimally (in the quadratic sense) to $\nu$.
> > > >
> > > >
> > > > #### Key takeover:
> > > > When defining **multivariate quantiles**, we learn the **cyclically monotone pushforward** between the data distribution $\nu$ over $\mathbb{R}^d$ and the **standard normal gaussian distribution** denoted $\mu$ (we could take the uniform distribution on the disk or the multivariate normal with 0 mean and identity covariance). We assume that $\nu$ is absolutely continuous w.r.t. the Lebesgue measure.
> > > >
> > > > Assume that we stake MVN(0,I), we obtain the $(1 - \alpha)$ coverage sets, we take **balls of radius $\rho_\alpha$**; where $\rho_\alpha$ is the $1-\alpha$ quantile of a $\chi^2$ distribution with $d$ degrees of freedom and compute their **pre-images** under this map.
> > > >
> > > > If we use a transform which is **not** cyclically monotone, the regions that we will obtain using this method will of course have the correct coverage, but might have **wild shapes**, because the measure preserving transform (appearing in Brenier's polar factorization) can be arbitrarily **complex**. Indeed, there infinitely many pushforward map between $\mu$ and $\nu$, but only the cyclically monotone pushforward allows to define  a quantile function and the associated ranks and signs that are meaningful.
> > > >
> > > > We illustrate this phenomenom in dimension $d=2$.

---

> ### Author Response · Authors · 2025-11-20
> **Benchmarking**
>
> Importantly, previous approaches—such as the Vector Quantile Regression (VQR) of Carlier, the Continuous Quantile Network of Sun *et al.*, the Continuous Vector Quantile Regression of Vedula *et al.*, and the Fast Non-Linear Quantile Regression of Rosenberg—do not explicitly model the rank map $Q_{Y \mid X}(Y): \mathcal{Y} \to \mathcal{V}$.
>
> Instead, these methods rely on discrete formulation of the OT and pointwise maximization to reconstruct the map. While they can, in principle, be extended to continuous quantile region construction through interpolation methods, such extensions *suffer from scalability limitations in higher dimensions*.
>
> [Sun *et al.*](https://openreview.net/pdf?id=BBxeo2Vuvbq) and [Vedula *et al.*](https://openreview.net/forum?id=DUZbGAXcyL) only provide a continuous representation for the quantile map $Q_{V \mid X}(V): \mathcal{V} \to \mathcal{Y}$, which, as discussed above, is less accurate than our construction. It also remains unclear how to construct a rank map $Q_{Y \mid X}(Y)$ using approaches described in Sun et al. and Vedula *et al.*
>
>
> **Benchmarking.** We have added [Sun *et al.*](https://openreview.net/pdf?id=BBxeo2Vuvbq) and [Vedula *et al.*](https://openreview.net/forum?id=DUZbGAXcyL) to our *generative expressiveness benchmark* on synthetic datasets. Below, we report the comparison with C-NQR$_{U}$; the complete results table is provided in the main paper.
>
> Experiments on synthetic datasets *clearly demonstrate the superior representational power of our approach*, highlighting that previous methods (particularly those of Sun *et al.* and Vedula *et al.*) *struggle to capture the full diversity of conditional quantile structures*. These results further substantiate our claim that our method provides a *more expressive and principled modeling framework* for conditional quantile estimation.
>
>
> | Dataset        | C-NQR${_U}$ | Sun et al. | Vedula et al. |
> |----------------|---------:|-------------------:|-----------------------:|
> | Star           |    0.184 |              0.274 |                  0.443 |
> | Glasses        |    0.785 |              0.931 |                  1.170 |
> | Banana         |    0.072 |              0.237 |                  0.401 |
> | Convex Star    |    0.182 |              0.252 |                  0.448 |
> | Convex Glasses |    0.656 |              0.793 |                  0.953 |
> | Convex Banana  |    0.101 |              0.211 |                  0.425 |

---

> > ### Author Response · Authors · 2025-11-20
> > **Additional comments**
> >
> > > Figure 3, right plot: the y-axis (log V)/d can be as low as -0.7 (even -5 for the second plot). For moderately large d this means the volume of the conformal set is exceedingly small. Is this expected?
> >
> > Yes, this is due to the differrences between marginal and conditional coverage. Sharper conditional coverage necessitates larger set size on average.
> >
> > > Line 467: the resulting response dimensions are 16, 4, 2, and 2. The potential based OT approach is known to be difficult to scale (due to the evaluation of the conjugate potential). The authors' experiments seem to confirm this observation. How high the dimension can be for the proposed approach to remain effective?
> >
> > Clearly, the main computational bottleneck of our method lies in its reliance on the explicit computation of the c-transform, which requires solving an inner minimization problem during training.
> > Using a combination of amortized optimization to get good initial points and LBFGS, we have neverthless show in our experiments, that we successfully scaled the conformal prediction method to more than 50-dimensional settings using the re-ranked pullback approach, without modifying the training procedure on a M4-laptop with 16 Gb memory. This is a major advantage of using the continuous formulation of optimal transport (OT) rather than the discrete versions used in most existing works and in all conformal approaches for estimating VQRs, even those with affine parameterizations. In 50 dimensions, OT methods that discretize the latent space through grid sampling become completely impractical. Consequently, our approach is the only VQR estimation method that scales effectively with dimension, which we consider an important result.
> >
> > > writing can be improved. A lot of details are deferred to the appendix, including the description of the proposed methods (Algorithms 1, 2 and 3). At times, it is hard to appreciate what are the significant modifications on the authors' side.
> >
> > Our work makes contributions in two distinct areas: optimal transport and conformal methods. Clearly, both topics have given rise to a large body of research over the past decade, and it is a real challenge to provide an exhaustive and comprehensible state-of-the-art overview within the constraints of the main text. It is simply not feasible to compress such a volume of information into a few paragraphs. We acknowledge that our discussion of the differences with prior work could have been more detailed. However, given the 9-page limit, it is inherently challenging to provide a clear and thorough presentation of both our sophisticated algorithms and alternative approaches based on fundamentally different principles.
> >
> > For this reason, we chose to include part of the related work discussion in the appendix. However, we will revise the manuscript to cite additional key references in the main text in order to better highlight our essential contributions.

---

> ### Comment · Reviewer_Rw7A · 2025-11-24
> **Ignored questions**
>
> The authors seemed to have ignored one of my critical questions:
>
> - I was not able to verify the proof of Theorem 3 and I think its assumptions may be too strong. In particular, for the elliptical example the authors need to verify the displayed equation in Theorem 3.
>
> Let me repeat: I do not think your claim on the elliptical example is correct. Thus, Theorem 3 rests on an assumption that we do not know if it is ever satisfiable.
>
> The last paragraph of the proof of Theorem 3 seems to assume the optimality of HPD. If this is true you need to either explain it or give a proper citation. My guess is you can prove it using Neyman-Pearson.

---

> ### Author Response · Authors · 2025-11-24
> **Theorem 3 Clarifications**
>
> Apologies we missed posting the following note along the others.
>
> 1. Line 1244-1246 needs more explanation. The proof seems to rely on the assumption that HPD superlevel sets are "optimal"?
>
> Let $P$ be a probability measure on $\mathbb{R}^d$ that admits a density $p$ with respect to Lebesgue measure $\lambda$ (i.e., $dP=p\,d\lambda$).
> For $\tau\ge 0$, define the upper level set
> $$
> C_\tau := \{x\in\mathbb{R}^d:\; p(x)\ge \tau\},
> \qquad
> A_\tau := P(C_\tau)=\int_{C_\tau} p\,d\lambda .
> $$
> We prove that among all measurable sets $S\subseteq\mathbb{R}^d$ with $P(S)=A_\tau$, the set $C_\tau$ (more precisely, an HDR built from the superlevel set at threshold $\tau$) has minimal Lebesgue measure.
>
> ### Theorem (Minimal-volume property of highest–density regions)
> Fix $\tau\ge 0$. Let
> $$
> \mathcal{S}\_\tau := \{S\subseteq\mathbb{R}^d\ \text{measurable}: P(S)=A\_\tau \}.
> $$
> Then
> $$
> \lambda(S) \ge\ \lambda(C\_\tau^\star)\qquad \text{for all }S\in\mathcal{S}\_\tau,
> $$
> where $C_\tau^\star$ is any measurable set of the form
> $$
> C_\tau^\star \;=\; \{p>\tau\}\ \cup\ B,
> \qquad
> B\subseteq \{p=\tau\}\ \text{chosen so that}\ P(C_\tau^\star)=A_\tau.
> $$
> In particular, if $\lambda(\{p=\tau\})=0$, then $C_\tau^\star=\{p\ge\tau\}=C_\tau$ (up to $\lambda$-null sets), and it is the unique minimizer up to null sets.
>
> ---
>
> ### Proof
> Let $S\in\mathcal{S}\_\tau$ be arbitrary, so $\int\_S p\,d\lambda=\int\_{C_\tau^\star} p\,d\lambda=A\_\tau$.
> Consider the identity
> $$
> 0 \;=\; \int\_{\mathbb{R}^d} p\,( \mathbf{1}\_S - \mathbf{1}\_{C\_\tau^\star})\, d\lambda
> \;=\; \int\_{\mathbb{R}^d} (p-\tau)\,(\mathbf{1}\_S - \mathbf{1}\_{C\_\tau^\star})\, d\lambda
> \;+\; \tau \int_{\mathbb{R}^d} (\mathbf{1}\_S - \mathbf{1}\_{C\_\tau^\star})\, d\lambda.
> $$
> Rearranging gives
> $$
> \tau\bigl(\lambda(S)-\lambda(C\_\tau^\star)\bigr)
> \;=\; - \int\_{\mathbb{R}^d} (p-\tau)\,(\mathbf{1}\_S - \mathbf{1}\_{C\_\tau^\star})\, d\lambda.
> \tag{1}
> $$
>
> Split the domain into the disjoint sets
> $$
> D_+ := \{p>\tau\},\qquad D_0 := \{p=\tau\},\qquad D_- := \{p<\tau\}.
> $$
> By construction, $C_\tau^\star = D_+ \cup B$ with $B\subseteq D_0$.
> On $D_0$ the integrand $(p-\tau)$ vanishes, so the integral over $D_0$ is zero.
> Hence the right-hand side of (1) reduces to
> $$- \int\_{D\_+} (p-\tau)\,(\mathbf{1}\_S - \mathbf{1}\_{C\_\tau^\star})\, d\lambda
> \;-\; \int\_{D\_-} (p-\tau)\,(\mathbf{1}\_S - \mathbf{1}\_{C\_\tau^\star})\, d\lambda.$$
>
> Observe the signs:
> - On $D_+$, $p-\tau>0$ and $\mathbf{1}\_{C_\tau^\star}=1$. Thus
>   $(\mathbf{1}\_S - \mathbf{1}\_{C\_\tau^\star})\le 0$, so
>   $-(p-\tau)(\mathbf{1}\_S - \mathbf{1}\_{C\_\tau^\star})\ \ge\ 0$.
> - On $D\_-$, $p-\tau<0$ and $\mathbf{1}\_{C\_\tau^\star}=0$. Thus
>   $(\mathbf{1}\_S - \mathbf{1}\_{C\_\tau^\star})=\mathbf{1}\_S\ge 0$, so
>   $-(p-\tau)(\mathbf{1}\_S - \mathbf{1}\_{C\_\tau^\star})
>    = - (p-\tau)\mathbf{1}\_S \ \ge\ 0$.
>
> Therefore the right-hand side of (1) is nonnegative, yielding
> $$
> \tau\bigl(\lambda(S)-\lambda(C\_\tau^\star)\bigr) \;\ge\; 0.
> $$
> Since $\tau\ge 0$, we conclude $\lambda(S)\ge \lambda(C\_\tau^\star)$.
> If $\tau>0$, equality forces $\mathbf{1}\_S=\mathbf{1}\_{C\_\tau^\star}$ almost everywhere on $D\_+\cup D\_-$; any freedom can only lie on $D\_0=\{p=\tau\}$, exactly as stated.
> If $\tau=0$, the statement is trivial (every $S$ with full mass has the same probability and the minimizer is the essential support of $p$).
> This proves the claim.  $\square$
>
> ---
>
> ### Remarks
> 1. HDR interpretation.
>    The minimizing sets $C_\tau^\star$ are the highest–density regions: among all regions containing a given probability mass, they have the smallest Lebesgue measure.
>
> 2. Neyman–Pearson perspective (alternative proof sketch).
>    Minimizing $\lambda(S)$ under the constraint $\int_S p\,d\lambda = A_\tau$ is equivalent to maximizing
>    $\int_S p\,d\lambda - \tau \lambda(S) = \int_S (p-\tau)\,d\lambda$ over $S$,
>    which is solved by taking $S=\{p>\tau\}$ plus (if needed) a subset of $\{p=\tau\}$ to match mass—this is the standard likelihood–ratio (threshold) rule and mirrors the Neyman–Pearson lemma.
>
> 3. Uniqueness.
>    If $\lambda(\{p=\tau\})=0$, the minimizer is unique up to $\lambda$-null sets and equals $C_\tau=\{p\ge\tau\}$.
>
> 4. Regularity.
>    No smoothness of $p$ is required beyond measurability and integrability; measurability ensures the superlevel sets are measurable.

---

> ### Author Response · Authors · 2025-11-25
> **Cyclically Monotone Example**
>
> Assume that the data distribution $\nu$ is multivariate normal  with a non-singular covariance matrix $\Gamma$. We map it to the multivariate distribution with density $\tfrac{d \mu}{d \operatorname{Leb}}(u) = MVN(u; 0, I)$, where $\operatorname{Leb}$ is the Lebesgue measure.
>
> The cyclically motonone transform is $\nabla \Phi(u)= \Gamma^{-1/2} u$, where $\Gamma^{-1/2}$ is the Hermitian root-square of $\Gamma$ (and $\Phi(u)= (1/2) u^\top \Gamma^{-1/2} u$ is a convex function). The Brenier polar factorization theorem shows that any pushforward $\nu \rightarrow \mu$ can be decomposed as $T= \nabla \Phi \circ \sigma$, where $\sigma$ is a measure-preserving map for $\mu=N(0, I)$. This measure preserving maps is not cyclically monotone and therefore might affect the ranks used in the construction.
>
>
>  There are many ways to construct measure preserving maps of $\text{MVN}(0,I)$. We provide such a construction in $\mathbb{R}^2$. Let $(x(t),y(t)\_{t \geq 0})$ a two dimensional random process which evolves according to the the ODE:
> $$\dot{x}(t)= v\_{t,x}(x(t), \quad \dot{y}\_t= v\_{t,y}(y(t),
> $$
> where $v_t= (v\_{t,x},v\_{t,y})$ is the **velocity field**. Denote by $p\_t$ the pdf of $(x(t),y(t))$ at time $t$.
> The continuity equation (Liouville) shows that
> $$
> \nabla\_t p\_t + \nabla \cdot (p\_t v\_t) = 0
> $$
> The density is invariant under the flow of the ODE is $\nabla \cdot (p\_t v\_t)=0$ for all $t \geq 0$, in which case $p\_t= p\_0$ for all $t \geq 0$.
> Denote $w\_t= p\_t v\_t$. A canonical way to ensure $\nabla \cdot w_t=0$ is to set $w_{t,x}= \partial_y \Psi(x,y)$ and $w_{t,y}= - \partial_x \Psi(x,y)$, where $\Psi$ plays the role of an Hamiltonian. If we start from $p\_0$ the pdf of $MVN(0,I)$, the flow of the ODE
> $$
> \dot{x}(t) = \frac{1}{p\_0}(x(t),y(t)) \partial\_y \Psi(x(t),y(t)) \quad
> \dot{y}(t) = -\frac{1}{p\_0}(x(t),y(t)) \partial\_x \Psi(x(t),y(t))
> $$
> is measure preserving for $p\_0$, for any $t > 0$. In other words, if we define $(x(t),y(t))= \operatorname{F}\_{0 \rightarrow t}(x(0),y(0))$ the flow of the solutions of the ODE, ${\operatorname{F}\_{0 \rightarrow t}}_\star p\_0 = p\_0$. Along the flow of the solutions,
> $$
> \frac{d}{d t} \Psi(x(t),y(t))= \partial\_x \Psi(x(t),y(t)) \, \dot{x}(t) + \partial\_y \Psi(x(t),y(t)) \dot{y} = 0
> $$
> so the Hamiltonionan $\Psi$ is constant along the orbit: $\Psi(x(t),y(t))=C$. We illustrate this with the mixing
> $$
> \Psi(x,y)= (x^2+y^2)^{3/2}(1-x^2-y^2)y
> $$
>
> We put new section in the main text to discuss this topic and provide visualisations of described phenomena.
>
> In (A., B.), we show the level set obtained by applying the cyclically monotone transform. In (C.,D.) we show the level sets obtained using a non cyclically monotone transformation $T$, obtained by integrating the Hamiltonian flow until tile $t=1.5$.
>
> It is clear that a transformation that is not cyclically monotone does not define a notion of ranks similar to the one described in [Marc Hallin *et al.*](https://projecteuclid.org/journals/annals-of-statistics/volume-49/issue-2/Distribution-and-quantile-functions-ranks-and-signs-in-dimension-d/10.1214/20-AOS1996.full), and cannot be used to construct a conformalisation procedure similar to ours.

---

> > ### Comment · Reviewer_Rw7A · 2025-11-28
> >
> > Thank you for the further explanations! I have no question about the proof of Theorem 3 now: indeed Neymann-Pearson suffices. I suspect this is a known result and it perhaps makes sense to find an appropriate reference (if not, please include your proof).
> >
> > I have also overlooked the possibility of setting j_x to be a constant function of y. This makes the elliptical example, and hence the premise of Theorem 3, much more believable. I would suggest writing down the details in Remark 3 of the verification of each assumptions in Theorem 3. One inconsistency that I realized is that in the main paper, phi in F(u) = phi(|u|) was assumed to be strictly decreasing while this assumption was dropped in Theorem 4 (page 24). In any case, I would appreciate if you could verify why r \mapsto phi(r) j_x(r) is strictly decreasing for the elliptical example (where j_x is constant of r). Note that F is a cdf so it is not clear to me how strong the decreasing assumption on phi in F(u) = phi(|u|) is. (Reading Example 3 again I realize maybe you meant the pdf f = phi(|u|)?)
> >
> > Thanks for the detailed explanation on the monotonicity issue. I knew maximal cyclic monotonicity in optimal transport and Brenier's theorem. What I meant is the logic behind the statement "monotonicity requires the map to be the gradient of a convex function" is not entirely natural (rigorous) since you will have to strengthen monotonicity to maximal cyclic monotonicity in order for this statement to be true, and then you will have to explain why maximal cyclic monotonicity is desirable in the first place (whereas maximality is perhaps not easy to justify). It would be much simpler to just say there are many push-forwards and we chose the one given by the gradient of a convex function. I don't necessarily agree with statements like
> >
> > >  only the cyclically monotone pushforward allows to define a quantile function and the associated ranks and signs that are meaningful
> >
> > because it really depends on what one desires for a notion of rank and sign. For elliptical distributions I agree optimal transport is a better way to go. For others I am less sure. It is not like maximal cyclic monotonicity does not have issues: you can't compose them for example (so going from a to b to c could be different from going from a to c). Anyway, this is a minor issue, and I think it is perfectly reasonable to choose the optimal transport approach (just don't already exclude other possibilities?).

---

> > > ### Author Response · Authors · 2025-11-28
> > > **"only cyclically monotone"**
> > >
> > > Of course, there is no single universally accepted extension of the notion of rank or quantile in the multivariate setting. This point has been discussed extensively in numerous recent papers (including work by Hallin, Konen, del Barrio, Ghosal, among others). Although there is now a broad consensus around the use of optimal-transport–based ranks and the associated center-outward distribution function, one can certainly imagine situations where this notion of rank may not be entirely appropriate—particularly for highly multimodal distributions (even though our construction still applies in such cases).
> > >
> > > It is worth emphasizing that the proposed VQR framework enables conditional density estimation in a manner that differs markedly from ‘classical’ conditional pdf estimators. In our approach, the conditioning is learned through amortization of a normalizing flow constrained to be cyclically monotone. Consequently, if one is not fully satisfied with this particular notion of multivariate rank, one can still rely on VQR to obtain HPD regions directly. In that scenario, however, the cyclic monotonicity constraint on the flow is no longer essential, since the objective is solely to model the conditional density.
> > >
> > > Our empirical observation is that confidence regions constructed from the ranks tend to exhibit greater robustness than those based on HPDs (better worst-slab coverageà. At this stage, however, this remains an experimental finding, and we do not yet have a clear theoretical explanation for this behavior. We agree that this point deserves further investigation, and we will elaborate on it in the revised version.

---

> > > ### Author Response · Authors · 2025-12-03
> > > **Remarks on Theorem 3 and Elliptical Examples.**
> > >
> > > We appreciate that the provided proof and details have resolved the concerns regarding the correctness of Theorem 3 and the validity of the elliptical example.
> > >
> > > Notation (PDF vs. CDF). In theorem 3 and 4, \phi(|u|)$ referred to the probability density function (PDF), not the CDF. We will clarify this to avoid ambiguity.  Thus, the assumption that $\phi$ is strictly decreasing radially is met for many elliptical distributions.
> > >
> > > Theorem 4 in the appendix was intended as a restatement for theorem 3 in the main text. Thus, theorem 4 relies on the same assumptions as Theorem 3. We will fix this inconsistency.

---

> ### Author Response · Authors · 2025-11-25
> **Theorem 3 Clarifications**
>
> We now proceed to verifying the assumptions of Theorem 3 for the elliptical case.
>
> Fix $x$. Assume the reference $U$ has spherical density $f_U(u) = \phi(|u|)$ for $u \in \mathbb{R}^{d_y}$, with $\phi$ continuous and strictly decreasing. Here $|\cdot|$ denotes the Euclidean norm. Let $Y \mid X = x$ be elliptical with location $m(x)$ and positive definite scatter matrix $\Sigma(x)$, obtained as $Y \mid X = x$ has the same distribution as $T_x(U)$, where $T_x(u) = m(x) + \Sigma(x)^{1/2} u$.
>
> The map $T_x$ is the gradient of the convex quadratic $\psi_x(u) = \tfrac12 u^\top \Sigma(x)^{1/2} u + m(x)^\top u$, so by Brenier–Knott–Smith theory for the quadratic cost, $T_x$ is the optimal transport from $F_U$ to $F_{Y \mid X = x}$.
>
> Denote the inverse map as $S_x(y) = \Sigma(x)^{-1/2} (y - m(x))$. Its Jacobian is constant: $\nabla_y S_x(y) = \Sigma(x)^{-1/2}$ and therefore $\text{det}(\nabla_y S_x(y)) = \text{det}(\Sigma(x)^{-1/2}) =: c_x > 0$.
>
> Theorem 3 assumes the existence of a function $j_x : [0,\infty) \to (0,\infty)$ such that $\text{det}(\nabla\_y Q^{-1}\_{Y|X}(y,x)) = j\_x(|Q^{-1}\_{Y|X}(y,x)|)$.
>
> In the elliptical case we have $Q^{-1}_{Y|X}(y,x) = S_x(y)$ and $\text{det}(\nabla_y S_x(y)) = c_x$, so we can simply set $j_x(r) = c_x = \text{det}(\Sigma(x)^{-1/2})$ for all $r \ge 0$.
>
> Finally, since $\phi$ is strictly decreasing and $j_x(r) = c_x > 0$ is constant, the product $r \mapsto \phi(r) j_x(r) = c_x \phi(r)$ is strictly decreasing in $r$, as required.

---

> > ### Comment · Reviewer_Rw7A · 2025-11-28
> > **Re: Theorem 3 Clarifications**
> >
> > For some reason I did not see this response when I responded earlier (perhaps I forgot to refresh the page). Apologies.

---

### Official Review · Reviewer_XoMS · 2025-10-29

**Soundness:** 3
**Presentation:** 2
**Contribution:** 3
**Rating:** 6
**Confidence:** 5

**Summary:**

This paper extends univariate conformal quantile regression (CQR) to the multivariate setting through the use of transport-based quantiles. To achieve this, the authors propose implementing optimal transport (OT) maps within vector quantile regression via Neural OT. The approach relies on input convex neural networks (ICNNs) to enforce the monotonicity of conditional quantiles or ranks. In addition, amortized optimization is employed to fasten the repeated computations inherent to conditional optimal transport.

**Strengths:**

This paper effectively integrates recent advances in computational optimal transport into the study of transport-based quantiles. It leverages Partially ICNNs (PICNNs) from Neural OT, amortized optimization for efficient computation of dual conjugates, and entropic regularization. These methodological choices are particularly relevant for enabling the use of transport-based quantiles in machine learning applications, where computational efficiency is crucial. The application of Neural OT to conformal quantile regression is both well-motivated and timely. I also appreciate the level of detail provided in the appendices regarding the different Neural OT implementations and the comprehensive presentation of results across all evaluation metrics.

**Weaknesses:**

My main concern is that the introduction may give a somewhat misleading impression of the existing literature. The incorporation of Neural OT into transport-based quantiles is not entirely new (see, for example, Continuous VQR, Vedula et al., 2023). Since this is claimed to be the first main contribution, a more explicit discussion of these related works would be valuable, in particular, to clarify the computational differences between the approach presented in this paper and prior methods. Although these papers are properly cited in the Appendix, the comparison should appear more prominently in the main text to help readers understand the methodological positioning.

The connection between transport-based quantiles and conformal prediction is also not entirely new, and the framing of the second contribution could be clarified. As currently written, it appears quite close to the contributions of recent works such as Thurin et al. (2025) and Klein et al. (2025). My understanding is that the approach presented in this paper generalizes conformal quantile regression by learning the conditional distribution Y∣X through transport-based quantiles, in the spirit of Feldman et al. (2023). In contrast, the perspective adopted in Thurin et al. (2025) and Klein et al. (2025) is to combine multivariate scores with multivariate quantiles, which represents a different line of research. This distinction is not clearly reflected in the introduction, although it does emerge in the main text and seems to motivate your re-ranking variants.

In addition, the experimental section would benefit from including numerical comparisons with related approaches, such as Feldman et al. (2023), to better position the method empirically.

**Questions:**

**PART I - Framing contributions in the literature**

I.1 *l.49 “While well studied in the univariate case, multivariate extensions are less developed and often reduce to coordinate methods that ignore the geometry of joint distributions (Dheur et al.)”*: The paper by Dheur et al. actually discusses several possible ways to extend conformal prediction (CP) to multivariate regression. It therefore does not seem to be the most appropriate reference to support the claim that existing methods “reduce to coordinate approaches.” You might instead cite works that genuinely adopt coordinate-wise strategies.
In addition, the paragraph currently gives the impression that transport-based quantiles are the only viable direction for multivariate CP, which is not the case, as you yourself note later (lines 343–349) and in the extended related-work section. The introduction should be revised to explicitly mention the existence of alternative multivariate CP frameworks.

I.2 *l.815:”Despite these advances, none of the above techniques exploits the full geometric structure of multivariate quantiles”*: It would help to clarify what is meant by “the full geometric structure.” One could argue that transport-based quantiles belong to the L-CP family of conformity scores introduced by Dheur et al. (2025). Indeed, your method also maps data to a latent space (albeit of the same dimension), similarly to Fang et al. (2025) or Feldman et al. (2023). These approaches appear complementary rather than mutually exclusive, so a more nuanced comparison would strengthen the framing.
For example, the monotonicity of the transport map in your model preserves the ordering of multivariate samples, this property distinguishes your method from the above works. It would be interesting to explain in which settings this specific feature provides a tangible advantage.

I.3 *On Neural OT for multivariate quantiles*: The work of Vedula et al. (2023) on Continuous VQR appears closely related, as it also implements PICNNs for Neural OT. However, the similarities and distinctions between your approach and theirs are not sufficiently discussed. We recommend adding a short paragraph explicitly positioning your method with respect to prior studies that apply Neural OT to transport-based quantiles, possibly including targeted numerical comparisons if computational differences are significant.
You may also wish to cite recent references such as [3], which learns multivariate quantiles with PICNNs for time series, and [2], which uses amortized optimization for repeated OT computations. Clarifying these links would considerably improve how your contribution is situated within the broader literature.

**Part II - Numerical comparisons**

II1. Providing a comparison between different implementation variants is a valuable practice that can guide future practitioners. However, I have several questions regarding the results presented in Section 7.1.
In the tables, it is unclear what formatting (bold or underlined) denotes — could you clarify this in the captions or text? More importantly, what are the main takeaways from these experiments? You compare several Neural OT–based implementations of conformal quantile regression, but it is not evident which one should be preferred in practice.
In Table 1, for instance, are EC-NQR and CPF the best-performing methods overall? Is there a meaningful difference between learning the quantile map and the rank map, or do these approaches yield comparable results? Also, why is the AC-NQR variant used in the experiments for conformal prediction? A short Discussion paragraph after the results section would greatly help interpret these findings and highlight the practical implications of the comparisons.

II2. Furthermore, some of the experimental comparisons seem to involve methods that are not directly comparable. For example, in Figure 4, you compare your CQR variants based on signed residuals (PBS and RPBS) with your main proposals (PB and RPB). Yet these two groups of methods do not rely on the same base model: PB and RPB employ a neural network, whereas other works use a random forest. It is therefore not surprising that PB and RPB yield lower volumes, as neural networks generally achieve higher predictive accuracy. A fairer comparison would restrict attention to PBS and RPBS versus ELL and OTCP(+), which share a similar setup.

II3. In line with this point, it would also be natural to compare your method with Feldman et al. (2023), since their approach uses a neural network–based quantile regression as the base model — arguably a closer alternative to your main proposals (PB and RPB). Finally, since your re-ranking approach falls within the L-CP framework introduced by Dheur et al. (2025), a comparison with some of the methods considered in that reference would also be informative.

**PART III - Clarification**

III1. The denomination “Conditional vector quantile regression” used in the abstract seems inappropriate. Conditional quantiles refer to the quantiles of a distribution conditional on covariates, while quantile regression designates the learning of such conditional quantiles from data. This latter notion corresponds to the focus of your work, together with a new algorithmic framework to achieve it. Therefore, the term vector quantile regression would be a more accurate and conventional choice.

III2. Theorem 1 (iii): you write a probability at the end of your sentence that I do not understand:  what is the meaning of $\leq$ between two vectors? What is F_U(u), when F_U is defined as a probability measure? I assume that the point (iii) should be a sentence similar to the point (i): there exists a mapping that pushes the conditional distribution Y/X=x to the reference distribution F_U. Also, Assumption (2) is not customary: [1] defines the rank function in a semi-discrete setting, a fact that might appear in a remark.

[1] Ghosal, Promit, and Bodhisattva Sen. "Multivariate ranks and quantiles using optimal transport: Consistency, rates and nonparametric testing." The Annals of Statistics 50.2 (2022): 1012-1037

[2] Amos, Brandon, et al. "Meta optimal transport." Proceedings of the 40th International Conference on Machine Learning. 2023

[3] Kan, Kelvin, et al. "Multivariate quantile function forecaster." International Conference on Artificial Intelligence and Statistics. PMLR, 2022.

---

> ### Author Response · Authors · 2025-11-26
> **Part I - Q1., Q2.**
>
> We sincerly thank the reviewer for valuable comments. We largely agree with the suggestions given and incorporate them in the Introduction, Related work, Extended State of the art and Remark in the appendix.
>
> > Part I - Q.1 l.49 The paper by Dheur et al. actually discusses several possible ways to extend conformal prediction (CP) to multivariate regression.
>
> We have updated the introduction to explicietly mention different developped approaches to multidimensional conformal prediction.
>
> > Part I - Q.2 It would be interesting to explain in which settings this specific feature provides a tangible advantage.
>
> We have added a short paragraph highlighting the important implications of cyclical monotonicity in Related Work and now cite Kan et al. in the extended state-of-the-art. We also mention more explicitly other works on multidimensional conformal prediction, so our contribution is more prominent.
>
> Regarding the L-CP framework introduced by [Dheur *et al.*](https://arxiv.org/pdf/2501.10533), we argue that, to obtain a meaningful quantile mapping and a notion of ranks, one must employ a cyclically monotone transformation. Thus, a diffeomorphism to a latent space does not suffice; cyclical monotonicity goes beyond preserving the ordering of multivariate samples.
>
> To see this, one can construct a bijective, non-cyclically monotone mapping that will create very non-linear quantile regions.
>
> We illustrate this phenomenom in dimension $d=2$.
>
> Assume that the data distribution $\nu$ is multivariate normal  with a non-singular covariance matrix $\Gamma$. We map it to the multivariate distribution with density $\tfrac{d \mu}{d \operatorname{Leb}}(u) = MVN(u; 0, I)$, where $\operatorname{Leb}$ is the Lebesgue measure.
>
> The cyclically motonone transform is $\nabla \Phi(u)= \Gamma^{-1/2} u$, where $\Gamma^{-1/2}$ is the Hermitian root-square of $\Gamma$ (and $\Phi(u)= (1/2) u^\top \Gamma^{-1/2} u$ is a convex function). The Brenier polar factorization theorem shows that any pushforward $\nu \rightarrow \mu$ can be decomposed as $T= \nabla \Phi \circ \sigma$, where $\sigma$ is a measure-preserving map for $\mu=N(0, I)$. This measure preserving maps is not cyclically monotone and therefore might affect the ranks used in the construction.
>
>  There are many ways to construct measure preserving maps of $\text{MVN}(0,I)$. We provide such a construction in $\mathbb{R}^2$. Let $(x(t),y(t)\_{t \geq 0}$ a two dimensional random process which evolves according to the the ODE:
> $$\dot{x}(t)= v\_{t,x}(x(t), \quad \dot{y}\_t= v_{t,y}(y(t),
> $$
> where $v_t= (v\_{t,x},v\_{t,y})$ is the **velocity field**. Denote by $p_t$ the pdf of $(x(t),y(t)$ at time $t$.
> The continuity equation (Liouville) shows that
> $$
> \nabla\_t p\_t + \nabla \cdot (p\_t v\_t) = 0
> $$
> The density is invariant under the flow of the ODE is $\nabla \cdot (p_t v_t)=0$ for all $t \geq 0$, in which case $p_t= p_0$ for all $t \geq 0$.
> Denote $w_t= p\_t v\_t$. A canonical way to ensure $\nabla \cdot w\_t=0$ is to set $w\_{t,x}= \partial\_y \Psi(x,y)$ and $w\_{t,y}= - \partial\_x \Psi(x,y)$, where $\Psi$ plays the role of an Hamiltonian. If we start from $p\_0$ the pdf of $MVN(0,I)$, the flow of the ODE
> $$
> \dot{x}(t) = \frac{1}{p\_0}(x(t),y(t)) \partial\_y \Psi(x(t),y(t)) \quad
> \dot{y}(t) = -\frac{1}{p\_0}(x(t),y(t)) \partial\_x \Psi(x(t),y(t))
> $$
> is measure preserving for $p_0$, for any $t > 0$. In other words, if we define $(x(t),y(t))= \operatorname{F}\_{0 \rightarrow t}(x(0),y(0))$ the flow of the solutions of the ODE, $\operatorname{F}\_{0 \rightarrow t}  p_0 = p_0$. Along the flow of the solutions,
> $$
> \frac{d}{d t} \Psi(x(t),y(t))= \partial\_x \Psi(x(t),y(t)) \, \dot{x}(t) + \partial\_y \Psi(x(t),y(t)) \dot{y} = 0
> $$
> so the Hamiltonionan $\Psi$ is constant along the orbit: $\Psi(x(t),y(t))=C$. We illustrate this with the mixing
> $$
> \Psi(x,y)= (x^2+y^2)^{3/2}(1-x^2-y^2)y
> $$
>
> We add and extended remark to the appendix G. of our work to further support our argument. You can also find a visualistaion of a proposed flow in the appendix G.
>
> In (A., B.), we show the level set obtained by applying the cyclically monotone transform. In (C.,D.) we show the level sets obtained using a non cyclically monotone transformation $T$, obtained by integrating the Hamiltonian flow until tile $t=1.5$.
>
> It is clear that a transformation that is not cyclically monotone does not define a notion of ranks similar to the one described in [Marc Hallin *et al.*](https://projecteuclid.org/journals/annals-of-statistics/volume-49/issue-2/Distribution-and-quantile-functions-ranks-and-signs-in-dimension-d/10.1214/20-AOS1996.full), and cannot be used to construct a conformalisation procedure similar to ours.

---

> > ### Author Response · Authors · 2025-11-26
> > **Part I - Q3.**
> >
> > > Part I Q3. - Framing contributions in the literature
> >
> > Regarding the connection with [Sun *et al.*](https://openreview.net/pdf?id=BBxeo2Vuvbq) and [Vedula *et al.*](https://openreview.net/forum?id=DUZbGAXcyL). We acknowledge that [Sun *et al.*](https://openreview.net/pdf?id=BBxeo2Vuvbq) and [Vedula *et al.*](https://openreview.net/forum?id=DUZbGAXcyL) also propose approaches to construct a *continuous* quantile map $(Q_{U \mid X}(U): \mathcal{U} \to \mathcal{Y})$. However, for the conformal construction, a *rank map* $(Q_{Y \mid X}(Y): \mathcal{Y} \to \mathcal{U})$ is required instead. Moreover the works of [Carlier et al.](https://arxiv.org/abs/1406.4643) and subsequent developments ([Rosenberg et al.](https://arxiv.org/abs/2205.14977), [Sun et al.](https://openreview.net/pdf?id=BBxeo2Vuvbq), and [Vedula et al.](https://openreview.net/forum?id=DUZbGAXcyL)) are based on a linear parameterization of the quantile function as a linear combinations of the covariates or features computed from these covariates.
> >
> > In principle, one could attempt to invert the pointwise correspondence produced by VQR by taking a maximum over the discretization grid, but this approach would scale poorly.
> >
> > To further support the claim we have added [Sun *et al.*](https://openreview.net/pdf?id=BBxeo2Vuvbq) and [Vedula *et al.*](https://openreview.net/forum?id=DUZbGAXcyL) to our *generative expressiveness benchmark* on synthetic datasets. Below, we report the comparison with C-NQR$_{U}$; the complete results table is provided in the main paper.
> >
> > Experiments on synthetic datasets *clearly demonstrate the superior representational power of our approach*, highlighting that previous methods (particularly those of Sun *et al.* and Vedula *et al.*) *struggle to capture the full diversity of conditional quantile structures*. These results further substantiate our claim that our method provides a *more expressive and principled modeling framework* for conditional quantile estimation.
> >
> > | Dataset        | C-NQR${_U}$ | Sun et al. | Vedula et al. |
> > |----------------|---------:|-------------------:|-----------------------:|
> > | Star           |    0.184 |              0.274 |                  0.443 |
> > | Glasses        |    0.785 |              0.931 |                  1.170 |
> > | Banana         |    0.072 |              0.237 |                  0.401 |
> > | Convex Star    |    0.182 |              0.252 |                  0.448 |
> > | Convex Glasses |    0.656 |              0.793 |                  0.953 |
> > | Convex Banana  |    0.101 |              0.211 |                  0.425 |
> >
> > We have updated the exprimental part accordingly.
> >
> > It could be argued that [Meta Optimal Transport Amos *et al.*](https://arxiv.org/pdf/2206.05262) addresses a form of conditional optimal transport; however, its connection to our setting is sufficiently indirect that we do not include a direct comparison in the main text. By contrast, [Kan *et al.*](https://arxiv.org/pdf/2202.11316) seem to be learning some form of convex flow to approximate quantile of auto-regressive process and hence are more closely related to our approach, and we now cite this work in the extended state of the art.

---

> ### Author Response · Authors · 2025-11-26
> **Part II - Q4, Q6**
>
> > Part II - Q4.
>
> We woud like to thank the reviewer for valuable suggestions that allow to improve the clarity of our presentation. In particular, we mark the best result with bold text and the second best result with underscore in the tables. The corresponding sentence was added to the caption of the tables. We also add a small discussion on implications of conducted experiments in the experimental section.
>
> > Part II - Q6.
>
> Regarding the comparison with [Feldman *et al.*](https://arxiv.org/abs/2110.00816), the method relies on sampling and a subprocedure to determine the quantile region in latent space, making it difficult to scale to higher dimensions. We therefore present results we were able to compute for the smallest of our datasets. We see that our method performs on par with or better than PCP or ST-DQR, while being computationally more efficient. We report *WSC - worst slab coverage* and $\text{log-vol-d} = \frac{\log(V)}{d}$.
>
> ## Bio dataset
>
> | Method name                 | WSC (α=0.1) | (α=0.2)   | (α=0.3)   | log-vol-d (α=0.1) | (α=0.2)    | (α=0.3)    |
> | ------------------------    | --------------------------- | --------- | --------- | ----------------- | ---------- | ---------- |
> | ST-DQR ( [Feldman *et al.*](https://arxiv.org/abs/2110.00816) ) | 0.874                       |  0.659  | 0.534      | -0.238            |  -0.533          | -0.697           |
> | PCP ( [Wang *et al.*](https://arxiv.org/abs/2206.06584) )       | 0.796                       | 0.627     | 0.496     | -1.487            | -1.605     | -1.634     |
> | PB ( Ours )                 | 0.872                       | 0.784     | 0.677     | -0.322            | -0.285     | -0.285     |
>
> ## Blog dataset
>
> | Method name              | WSC (α=0.1) | (α=0.2) | (α=0.3)  | log-vol-d (α=0.1) | (α=0.2) | (α=0.3) |
> | ------------------------ | --------------------------- | ------- | -------- | ----------------- | ------- | ------- |
> | ST-DQR  ( [Feldman *et al.*](https://arxiv.org/abs/2110.00816) ) | 0.815              | 0.654   | 0.574    | 0.022             | -0.279   | -0.449     |
> | PCP ( [Wang *et al.*](https://arxiv.org/abs/2206.06584) )    |    0.771                    | 0.612   | 0.517    | -1.195            | -2.094  | -2.129  |
> | PB                       |    0.878                    | 0.751   | 0.666    | -0.641            | -0.721  | -0.744  |
>
> ## Sgemm dataset
>
> | Method name            | WSC (α=0.1) | (α=0.2) | (α=0.3)  | log-vol-d (α=0.1) | (α=0.2) | (α=0.3) |
> | ---------------------- | --------------------------- | ------- | -------- | ----------------- | ------- | ------- |
> | ST-DQR  ( [Feldman *et al.*](https://arxiv.org/abs/2110.00816) ) | 0.691                  | 0.528   | 0.701    | -1.206            | -1.274  | 1.822   |
> | PCP ( [Wang *et al.*](https://arxiv.org/abs/2206.06584) )   | 0.700                       | 0.534   | 0.415871 | -                 | -       | -       |
> | PB ( Ours )            | 0.887                       | 0.806   | 0.683    | -5.134            | -5.354  | -4.880  |

---

> > ### Author Response · Authors · 2025-11-26
> > **Part III - Q7., Q8.**
> >
> > > Part III - Q7.
> >
> > We thank the reviewer for attentiveness and agree that vector quantile regression suits better the object we are estimating. The text has been updated accordingly.
> >
> > > Part III - Q8.
> >
> > We want to clarify the notations:
> > 1. The inequality between vectors should be understood in coordinatewise sense.
> > 2. By $F_U(u)$ we mean distribution function (and not a probability measure).
> >
> > Thus, $F_U(u) = F_U(u_1, \dots, u_d) = \mathbb{P}(U_1 \le u_1, \dots, U_d \le u_d) = \mathbb{P}(U \le u)$. We made a corresponding remark at the beginning of Section 3.
> >
> > Regarding Assumption 2, we certainly agree that continuity is not needed and could be relaxed. We now discuss this in Appendix B.

---

> ### Author Response · Authors · 2025-12-01
> **Part II - Q5.**
>
> Towards a more fair comparison.
>
> We thank the reviewer for discussing this important methodological nuance. Indeed, the proposed methods differ in the kind of base model they use. While we agree that PBS and RPBS may be considered more related to ELL and OTCP, a fair comparison becomes tricky when more methods are taken into account. For example, we provide additional comparison with PCP([Wang *et al.*](https://arxiv.org/abs/2206.06584)) and are working on including ST-DQR ([*et al.*](https://arxiv.org/abs/2110.00816)).
>
> Since all considered methods are based on some form of (split) conformal prediction, we can naturally separate the calibration procedure from the score computation.
>
> PCP relies on a base conditional generative model and constructs the prediction set as a union of balls of a shared radius around the sampled points in $Y$ space. Thus, score computation / set inclusion test requires sampling, which for higher dimensions poses computation issues and precision issues of the test. Authors suggest 1000 samples (forward passes) for $Y$ of dimensions 2.
>
> ST-DQR, on the other hand, relies on two base models: a Conditional VAE and another network for DQR ([Paindaveine *et al.*](https://doi.org/10.1016/j.jmva.2010.08.004)). Prediction set is obtained as intersection of half-spaces, thus computing the score / testing inclusion relies on sampling once again. Authors suggest 10 000 samples (forward passes) for $Y$ of dimensions 2.
>
> In comparison, our approach requires querying the model just once. Nevertheless, both PCP and ST-DQR bare similarities to our methods and are rightfully included in the comparison.
>
> Getting back to the fairness of comparison. After highlighting that methods differ in more aspects than just the use of neural network as a base model, we would like to provide more rationale for our comparison strategy. First, we tried to standardize the calibration procedures, as some of them are shared by multiple methods. Note that the calibration sets are also of equal size for all methods. Second, we control the number of parameters-to-traing data ratio of the underlying base models, aiming to have no more than $10\\%$. Given the many aspects in which the compared methods differ, we argue that this approach is the right step towards a fair comparison. We will include these remarks about our comparison and also mention the similarities of PBS, RPBS and ELL, OTCP(+) suggested by the reviewer in the revised text.

---

### Official Review · Reviewer_Vkyx · 2025-10-31

**Soundness:** 3
**Presentation:** 3
**Contribution:** 2
**Rating:** 4
**Confidence:** 3

**Summary:**

This paper combines neural optimal transport with conditional vector quantile regression to improve multivariate prediction and uncertainty estimation. The authors use convex neural networks to model transport maps, suggest employing amortized optimization to improve efficiency, and apply entropic regularization for stability. They then use the resulting quantile maps to build conformal prediction sets that represent uncertainty in multiple dimensions. While the approach brings together several existing ideas, it is implemented carefully and evaluated on both synthetic and real datasets.

**Strengths:**

1.	This work introduces a new, novel method (NQR) that connects neural optimal transport with VQR.
2.	Optimality guarantee (Theorem 3).
3.	The writing is generally clear and well-structured.
4.	The experimental results show that the proposed approach yields better performance compared to the baselines.

**Weaknesses:**

1.	The paper’s novelty is moderate rather and not groundbreaking. Most of its core components, such as PICNN/ICNN (Makkuva et al., 2020; Amos et al., 2017), amortized optimization (Amos, 2023), entropic regularization (Genevay et al., 2016), and recent reranking conformal methods (Thurin et al., 2025; Klein et al., 2025), have already been explored in earlier work. The main contribution lies in how these ideas are combined and adapted for the conditional setting.
2.	The assumptions of Theorem 3, which is the main theoretical result of this paper, are hard to verify in practice.
3.	The conformal prediction experiments should include comparisons with VQR (Carlier et al., 2016, 2017, 2020), STDR (Feldman et al. 2023) and PCP (Wang et al. 2023).

**Questions:**

1.	How sensitive are the results to violations of the radial Jacobian assumption in Theorem 3?
2.	How does the proposed approach compare against modern conditional flow or diffusion models for conditional sampling and prediction sets?
3.	Can the proposed approach handle large output dimensions, e.g., $d_y > 20$?
4.	This paper introduces multiple methods: pb, rpb and hdp. Is there a suggestion for which of them to use in practice?

---

> ### Author Response · Authors · 2025-11-20
> **Novelty**
>
> Our paper has **two main contributions**.
>
> **First contribution.** We adapt partially input convex neural networks (PICNNs) to estimate vector quantile functions, whis is an application that, to our knowledge, has not been explored. While PICNNs are common in optimal transport, they have not been used for quantile estimation. Earlier work (e.g., [Carlier *et al.*](https://arxiv.org/abs/1406.4643), [Marc Hallin *et al.*](https://projecteuclid.org/journals/annals-of-statistics/volume-49/issue-2/Distribution-and-quantile-functions-ranks-and-signs-in-dimension-d/10.1214/20-AOS1996.full), [Victor Chernozhukov *et al.*](https://arxiv.org/abs/1412.8434), [Eustasio del Barrio *et al.*](https://www.tandfonline.com/doi/epdf/10.1080/01621459.2024.2366029?needAccess=true)) typically assumes a linear dependence between the latent variable and the covariates and relies on discrete solvers to obtain pointwise solutions, whereas [Eustasio del Barrio *et al.*](https://www.tandfonline.com/doi/epdf/10.1080/01621459.2024.2366029?needAccess=true) employ kernel estimators in combination with discrete optimal transport. More recent methods ([Rosenberg *et al.*](https://arxiv.org/abs/2205.14977), [Sun *et al.*](https://openreview.net/pdf?id=BBxeo2Vuvbq), and [Vedula *et al.*](https://openreview.net/forum?id=DUZbGAXcyL)) learn feature maps but still impose essentially linear constraints. In contrast, *we estimate the vector quantile without structural assumptions on the dual potentials*, using a fully neural parameterization. To the best of our knowledge, this has not been demonstrated previously.
>
> **Second contribution.** We leverage this extension to improve existing methods for constructing multidimensional conformal prediction regions. [Klein *et al.*](https://arxiv.org/abs/2502.03609) propose an unconditional procedure for constructing multidimensional conformal regions; however, their procedure does not construct conditional quantiles. Extending their approach to achieve approximate conditional coverage requires solving a Vector Quantile Regression (VQR) problem, as we do.
>
> [Thurin *et al.*](https://arxiv.org/abs/2501.18991) are most similar to our setting because they construct conformal sets with approximate conditional conformal guarantees. In contrast to our approach, they use a discrete formulation of optimal transport with weights computed using KNN directly on the covariates. However, our approach is more flexible and offers several advantages.
> - It learns an explicit, differentiable map between data and latent spaces, along with its inverse, which [Thurin *et al.*](https://arxiv.org/abs/2501.18991) do not provide. This enables direct visualization, more accurate volume computations, and explicit construction of quantile regions.
> - KNN methods are known to be severely affected by the dimension of the covariate space – this effect could be mitigated by feature engineering, but [Thurin *et al.*](https://arxiv.org/abs/2501.18991). do not address this aspect.
> - [Thurin *et al.*](https://arxiv.org/abs/2501.18991) use a discrete optimal transport solver that requires sampling a reference grid in latent space, which also scales poorly with the dimension of the dependent variable. In contrast, our method avoids any space discretization, making quantile construction better suited to higher-dimensional X and Y. In our first set of experiments, we intentionally limited the number of features to 50 across datasets, and the dimension of $Y$ never exceeded 16.

---

> > ### Author Response · Authors · 2025-11-20
> > **Comparison with Thurin et al.**
> >
> > To illustrate the impact of the dimension of the covariates and the dependent variables, we have added pedestrian trajectories from a Japanese shopping center (https://dil.atr.jp/crest2010_HRI/ATC_dataset/), using 17 time steps (96 features) as covariates X and 5 future time steps (15 variables) as targets Y. We compare worst-slab coverage between the methods of [Thurin *et al.*](https://arxiv.org/abs/2501.18991) and our neural network estimator; our method consistently outperforms both their baseline and their localized variant.
> >
> > We have summarized the results in the table below. Details of the experiment and the various hyperparameter choices are now included in the supplement, with an illustration in the main text. The table shows that our method is significantly more accurate than that of [Thurin *et al.*](https://arxiv.org/abs/2501.18991) in terms of conditional coverage, even in this relatively simple setting.
> >
> > **Achieved worst-slab coverage vs requested (higher is better)**
> >
> > | Method | Coverage : 0.9 | 0.8 | 0.7 |
> > |---|---:|---:|---:|
> > | [Thurin] OT-CP-Global | 0.788 | 0.573 | 0.418 |
> > | [Thurin] OT-CP-Local  | 0.873 | 0.718 | 0.599 |
> > | [Our method] PB       | 0.876 | 0.744 | 0.629 |
> > | **[Our method] RPB**  | **0.884** | **0.775** | **0.672** |
> >
> >  We also compare the volumes of the resulting regions, reporting $\frac{1}{d}\log V$ to mitigate the exponential scaling of volume in high dimensions. Across all target coverages, the PB method attains substantially smaller volumes than the approaches of Thurin et al. while still achieving superior worst-slab coverage (see table above).
> >
> > **log-d-volumes of the predicted region**
> > | Method | Coverage : 0.9 | 0.8 | 0.7 |
> > |---|---:|---:|---:|
> > | [Thurin] OT-CP-Global | 1.537 | 1.426 | 1.284 |
> > | [Thurin] OT-CP-Local  | -0.759 | -0.778 | -0.833 |
> > | [Our method] PB       | -1.952 | -2.848 | -3.108 |
> > | [Our method] RPB  | 1.534 | 1.430 | 1.268 |

---

> > > ### Author Response · Authors · 2025-11-20
> > > **Comment on VQR by Carlier et al.**
> > >
> > > Regarding the comparison with VQR, we note that the work of Carlier *et al.* and subsequent developments based on its assumptions are based (i) on a discrete version of OT (ii) a linear parameterization of the quantile function as a linear combinations of the covariates or features computed from these covariates. We acknowledge that [Sun *et al.*](https://openreview.net/pdf?id=BBxeo2Vuvbq) and [Vedula *et al.*](https://openreview.net/forum?id=DUZbGAXcyL) also propose approaches to construct a *continuous* quantile map $(Q_{U \mid X}(U): \mathcal{U} \to \mathcal{Y})$, still using an affine parameterization of the quantile function. However, for the conformal construction, a *rank map* $(Q_{Y \mid X}(Y): \mathcal{Y} \to \mathcal{U})$ is required instead.
> > >
> > > In principle, one could attempt to invert the pointwise correspondence produced by VQR by taking a maximum over the discretization grid, but this approach would scale poorly. Moreover, VQR methods rely on the assumption of linear dependence in the quantile function, which is very restrictive and typically yields much weaker results than those obtained either by [Thurin et al.](https://arxiv.org/abs/2501.18991), [Eustasio Del Barrio et al](https://arxiv.org/abs/2204.11756) and also with our method. For these reasons, we did not include discrete VQR solvers in our conformal benchmark. And believe that the combination of generative modeling experiments in Table 1 and the comparison with optimal-transport–based methods in Table 3 provides a comprehensive overview of our method’s strengths.

---

> ### Author Response · Authors · 2025-11-20
> **Experiments with VQR by Carlier et al. and PCP by Wang et al.**
>
> Per the reviewers’ request, we added VQR-based conformalization baselines to our benchmarks on the Bio, Blog, and SGEMM datasets with PCP method introduced by *Wang et al.*. *The method of [Carlier *et al.*](https://arxiv.org/abs/1406.4643) does not scale to scm20d: the target dimension is 16, so even a coarse grid with two points per coordinate contains $2^{16}=65{,}536$ cells.* Such a grid is too sparse to yield accurate conditional quantile estimates and cannot support efficient computation for a reliable rank map $Q_{Y\mid X}(Y): \mathcal{Y}\to\mathcal{U}$.
>
> We report conditional coverage and $\frac{1}{d}\log V$ (Log-d-volume) for conformal sets constructed via the VQR baseline of [Carlier *et al.*](https://arxiv.org/abs/1406.4643), Want et al. method and for PB method introduced in our work.
>
> ### bio dataset
> | method_name              | worst_slab_coverage (α=0.1) | (α=0.2)   | (α=0.3)   | log_vol_d (α=0.1) | (α=0.2)    | (α=0.3)    |
> | ------------------------ | --------------------------- | --------- | --------- | ----------------- | ---------- | ---------- |
> | VQR ( Carlier *et al.* ) | 0.896                       | 0.736     | 0.662     | 2.778             | 2.663      | 2.566      |
> | PCP ( Wang *et al.* )    | 0.796                       | 0.627     | 0.496     | -1.487            | -1.605     | -1.634     |
> | ST-DQR ( Feldman *et al.* ) | 0.874                       | 0.659     | 0.534     | -0.238            | -0.533     | -0.6969    |
> | PB ( Ours )          | 0.872                   | 0.784 | 0.677 | -0.322        | -0.285 | -0.285 |
>
> ### blog dataset
> | method_name              | worst_slab_coverage (α=0.1) | (α=0.2) | (α=0.3)  | log_vol_d (α=0.1) | (α=0.2) | (α=0.3) |
> | ------------------------ | --------------------------- | ------- | -------- | ----------------- | ------- | ------- |
> | VQR ( Carlier *et al.* ) | 0.881                    | 0.761   | 0.672    | 2.381             | 2.328   | 2.269   |
> | PCP ( Wang *et al.* )    | 0.771                       | 0.612   | 0.517111 | -1.195            | -2.094  | -2.129  |
> | ST-DQR ( Feldman *et al.* ) | 0.815                    | 0.654   | 0.574    | -0.022            | -0.279  | -0.449  |
> | PB                       | 0.878                       | 0.751   | 0.666    | -0.641            | -0.721  | -0.744  |
>
> ### sgemm dataset
> | method_name            | worst_slab_coverage (α=0.1) | (α=0.2) | (α=0.3)  | log_vol_d (α=0.1) | (α=0.2) | (α=0.3) |
> | ---------------------- | --------------------------- | ------- | -------- | ----------------- | ------- | ------- |
> | VQR ( Carlier et al. ) | 0.819847                    | 0.653   | 0.653    | 2.063             | 1.998   | 1.925   |
> | PCP ( Wang *et al.* )  | 0.700                       | 0.534   | 0.415871 | -                 | -       | -       |
> | ST-DQR ( Feldman *et al.* ) | 0.691                  | 0.528   | 0.701    | -1.206            | -1.274  | 1.822   |
> | PB ( Ours )            | 0.887                       | 0.806   | 0.683    | -5.134            | -5.354  | -4.880  |
>
>
>
> Across coverage targets and datasets, VQR-based conformalization yields substantially larger volumes than our method and exhibits inferior conditional coverage. The PCP baseline, while achieving small volumes, fails to produce valid conditional conformal regions. We further observe that the performance of the VQR baseline deteriorates with increasing dimensionality, as reflected in the SGEMM results. Collectively, these findings reinforce the advantages of our approach.
>
> *We are still working on adding Feldman method to the benchmark.*
> After adding results on ST-DQR method of Feldman et al., we observe that PB outperforms it in both conditional coverage and volume of the prediction sets. Similar to VQR, ST-DQR relies on a grid in $\mathcal{Y}$ and suffers from the same poor computational performance scaling.

---

> > ### Author Response · Authors · 2025-11-20
> > **Sensitivity of radial Jacobian assumption in Theorem 3**
> >
> > *   **Validity is Robust:** First, we emphasize that the validity of our confidence sets (i.e., $\mathbb{P}(Y \in \hat{C}) \ge 1-\alpha$) is **not sensitive** to this assumption. Because we employ Split Conformal Prediction, marginal coverage is guaranteed in finite samples regardless of the underlying distribution geometry or the quality of the estimated map.
> > *   **Stability:** Regarding volume efficiency, the Radial Jacobian assumption is satisfied by the broad family of **Elliptical distributions** (including Gaussian). This aligns with standard assumptions in recent literature, such as  Braun et al. (2025) (*arXiv:2507.20941*). Beyond exact elliptical distributions, Optimal Transport maps are known to be quantitatively stable (see, e.g., Delalande & Mérigot, 2021, *arXiv:2103.05934*, noting that constants heavily depend on dimension and assumptions). Consequently, if the true conditional distribution is close to satisfying the assumptions of Theorem 3, the volume of the set produced by our approach can be shown close to the optimal set. Thus, for mild violations, the efficiency does not completely colapse.
> > *   **Topological Limitation (Multimodality):** We acknowledge, however, a topological limitation. If the conditional distribution is strongly multimodal (e.g., a mixture of widely separated Gaussians), the theoretical volume-optimal set is disjoint. Since our method produces pullbacks of balls via continuous maps, the resulting sets ($\mathcal{C}^{\text{pb}}$ and $\mathcal{C}^{\text{rpb}}$) are necessarily connected. In such cases, our method cannot recover the disjoint HPD region and will instead produce a connected set. This is a limitation shared by any method producing connected sets.

---

> > > ### Author Response · Authors · 2025-11-20
> > > **Additional comments**
> > >
> > > > The assumptions of Theorem 3, which is the main theoretical result of this paper, are hard to verify in practice.
> > >
> > > The assumptions of Theorem 3 are indeed hard to verify from finite data. Nonetheless, they do cover a standard modelling choice. In particular,  conditionally on $X=x$, the error of $Y$ is well-approximated by an $x$-dependent anisotropic (elliptical) Gaussian. In addition, the experiments show that the method behaves well on real where these assumptions may typically not hold or hold in an approximate way.
> > >
> > > >Q2. How does the proposed approach compare against modern conditional flow or diffusion models for conditional sampling and prediction sets?
> > >
> > > The objective of our generative experiments is vector quantile regression (VQR), not generic conditional density modeling. Consequently, comparisons with models that do not solve VQR have limited interpretability. Our estimator enforces cyclic monotonicity of the transport map, which yields uniquely defined multivariate ranks required for constructing conformal sets. Standard generative models (VAEs, conditional flows, and diffusion models) do not parameterize quantile operators; strong scores on generation metrics therefore do not directly reflect performance on VQR. For conformal prediction, it is essential that the learned operator implement VQR.
> > >
> > > Despite these structural constraints, our method performs comparably to modern normalizing-flow baselines parameterized as gradients of convex potentials, e.g., [Chin-Wei Huang *et al.* (2020)]( https://arxiv.org/abs/2012.05942).
> > >
> > > > Q3. Can the proposed approach handle large output dimensions, e.g., ?
> > >
> > > We have successfully run the method in 30 dimensions in our experiments without any changes to the training procedure, but with significant increase in training time. We have considered [ATC](https://dil.atr.jp/crest2010_HRI/ATC_dataset/) dataset, containing trajectories of pedestrians in shopping center in Japan. *Re-ranked pull back method achives 0.89 marginal coverage and 0.86 worst-slab coverage on prediction of 10 next timesteps with 60 timesteps context and requested coverage level of 0.9.* We acknowledge that the proposed approach cannot be adapted to more complex modalities, such as images or text generation, in its current form. It opens an opportunity for further development, notably with recent advances in computational optimal transport in the works by [Nazar Buzun, Maksim Bobrin](https://arxiv.org/abs/2403.03777) or [Théo Uscidda, Marco Cuturi](https://arxiv.org/abs/2302.04953), which enable solving transport problems with non-convex parameterizations and may extend quantile solutions to higher dimensions and more complex modalities.
> > >
> > > > Q4. This paper introduces multiple methods: pb, rpb and hdp. Is there a suggestion for which of them to use in practice?
> > >
> > > - **HPD (high probability density sets).** Conceptually, hdp is the *optimal* construction: if rank map (and its Jacobian) were known exactly, the resulting sets would coincide with true HPD regions and minimize volume among all sets with coverage $1-\alpha$. In practice, however, we tried multiple approaches and found that  estimating $\det \nabla_y \widehat{Q}^{-1}_{Y\mid X}$ is numerically unstable and expensive, especially in higher dimensions, and our experiments show that this leads to worse coverage–volume trade-offs than the simpler alternatives.
> > >
> > > - **PB / RPB (pullback and reranked pullback).** Both pb and rpb avoid density/Jacobian estimation and are much more stable. pb is the simplest and works well when the conditional distribution is close to radially symmetric in rank space. rpb adds a reranking step to correct deviations from the reference $F_U$ and is therefore more robust under misspecification. In our experiments, rpb often offered the best empirical coverage–efficiency trade-off.
> > >
> > >  As a conclusion we would say that if the most accurate conditional coverage is requiered and having small prediction sets is not crucial one can use RPB method. However if one aims to have smallest prediction sets, and can tolerate less accurate conditinal coverage, using PB method is recommended.

---

### Official Review · Reviewer_6k36 · 2025-11-07

**Soundness:** 3
**Presentation:** 2
**Contribution:** 2
**Rating:** 6
**Confidence:** 3

**Summary:**

The general methodology is to rely on mutlivariate quantiles of Hallin et al. to define a multvariate score, converted into univariate score, from which conformal prediction tools can be applied.

The novelty is the nature of the optimal transport solver used to compute the center outward map. Instead of relying on entropic-regularized solvers (Sinkhorn) like Klein et al, authors rely on the gradient of a partial input convex neural network (PICNN). The training leverages the amortization trick of Amos et al to accelerate the fitting of the dual potential.

The method is benchmarked on a few multivariate regression datasets, like the ones introduced by Rosenberg et al. The average Sliced-2 wasserstein-2 distance is computed to benchmark the method, and other metrics like coverage or log-volume of prediction sets are compared against competitors works or baselines.

**Strengths:**

### Motivation

The paper extend the work of Thurin et al; Klein et al using PICNN, a parameterized approach.

###  Exposition

The paper makes a good job at introducing the necessary tools (center outward quantiles, neural dual solvers, conformal prediction).

### PICNN training

ICNN (and PICNN) training can be challenging. Various variants are tested. The results of Figure 3 and 4, show strong results: this is my main motivation for leaning toward acceptance. Fig 1. confirms that this PICNN can fit complex shapes. Since the work is done in low dimensions (the Y remaining much smaller than the X space) PICNN don't suffer as much from the curse of dimensionality.

**Weaknesses:**

### Novelty on the multivariate conformal prediction part

This is similar to the approach of two concurrent works (Thurin et al, Klein et al) that this paper acknowledges: using multivariate quantiles of Hallin et al, fit the (empirical) center outward map (or an approximation of thereof), and converting these generalized (multivariate) quantiles into 1D score .

The main novelty is reliance on PICNN/ICNN instead of cubic linear solvers (Thurin et al) or entropic solvers (Klein et al), using other tricks documented in W2 Monge map learning.

### Clarity

I had some issues other than navigate the experimental part, that lacked clarity at times. e.g Table 1 is not directly related to conformal prediction, it's simply measuring the distance between $T_{\sharp}P$ and $Q$ (ie it's a sanity check). Also some issues with Fig 2 (see below).

Similarly, it's not clear to me which message Fig 2. is trying to convey.

**Questions:**

### Fig. 2

Why is the Wasserstein distance reported as negative?

### Pareto front

You write:

> The re-ranking step of RPB and RPBS allows to achieve a slightly
sharper conditional coverage, but the increase in prediction sets volume make it a questionable trade-
off.

I think this observation mandates a Pareto front, to better position the relative strengths of each method.

---

> ### Author Response · Authors · 2025-11-18
> **Novelty on the multivariate conformal prediction part**
>
> We are grateful to the reviewer for their insightful comments and suggestions, which have greatly contributed to strengthening our work.
>
> Klein et al. propose an unconditional procedure for constructing multidimensional conformal regions; however, their procedure does not construct conditional quantiles, as the entire paper focuses on marginal coverage guarantees. Extending their approach to achieve approximate conditional coverage requires solving a Vector Quantile Regression (VQR) problem, as we do. An initial approach in this direction was taken by [Eustasio del Barrio et al.](https://www.tandfonline.com/doi/epdf/10.1080/01621459.2024.2366029?needAccess=true), who used a kernel-based estimator of the conditional density in combination with the discrete weighted version of optimal transport. A distinct line of research was introduced by [Carlier et al.](https://arxiv.org/abs/1406.4643), who developed the concept of Vector Quantile Regression. Most early work in this area focused on vector quantiles that are linear functions of covariates (or features computed from the covariates). More recently, this work was extended by [Rosenberg et al.](https://arxiv.org/abs/2205.14977), [Sun et al.](https://openreview.net/pdf?id=BBxeo2Vuvbq), and [Vedula et al.](https://openreview.net/forum?id=DUZbGAXcyL), who all propose learning the feature maps while still imposing the same linear constraints. In contrast, we estimate the vector quantile without assumptions on the form of the dual potentials, using a fully neural parameterization. In our work, the quantile function is parameterized as the gradient of a partially input convex neural network. To the best of our knowledge, this has not been demonstrated previously.
>
> Thurin et al. are most similar to our setting because they construct conformal sets with approximate conditional conformal guarantees. In contrast to our approach, they use a discrete formulation of optimal transport with weights computed using KNN directly on the covariates. However, our approach is more flexible and offers several advantages. (i) It learns an explicit, differentiable map between data and latent spaces, along with its inverse, which Thurin et al. do not provide. This enables direct visualization, more accurate volume computations, and explicit construction of quantile regions. (ii) KNN methods are known to be severely affected by the dimension of the covariate space – this effect could be mitigated by feature engineering, but Thurin et al. do not address this aspect. (iii) Thurin et al. use a discrete optimal transport solver that requires sampling a reference grid in latent space, which also scales poorly with the dimension of the dependent variable. In contrast, our method avoids any space discretization, making quantile construction better suited to higher-dimensional X and Y. In our first set of experiments, we intentionally limited the number of features to 50 across datasets, and the dimension of $Y$ never exceeded 16.
>
> To illustrate the impact of the dimension of the covariates and the dependent variables, we have added pedestrian trajectories from a Japanese shopping center (https://dil.atr.jp/crest2010_HRI/ATC_dataset/), using 17 time steps (96 features) as covariates X and 5 future time steps (15 variables) as targets Y. We compare worst-slab coverage between the methods of Thurin et al. and our neural network estimator; our method consistently outperforms both their baseline and their localized variant.
>
> We have summarized the results in the table below. Details of the experiment and the various hyperparameter choices are now included in the supplement, with an illustration in the main text. The table shows that our method is significantly more accurate than that of Thurin et al. in terms of conditional coverage, even in this relatively simple setting.
>
> ### Achieved worst-slab coverage vs requested (higher is better)
>
> | Method | Coverage : 0.9 | 0.8 | 0.7 |
> |---|---:|---:|---:|
> | [Thurin] OT-CP-Global | 0.788 | 0.573 | 0.418 |
> | [Thurin] OT-CP-Local  | 0.873 | 0.718 | 0.599 |
> | [Our method] PB       | 0.876 | 0.744 | 0.629 |
> | **[Our method] RPB**  | **0.884** | **0.775** | **0.672** |
>
>  We also compare the volumes of the resulting regions, reporting $\frac{1}{d}\log V$ to mitigate the exponential scaling of volume in high dimensions. Across all target coverages, the PB method attains substantially smaller volumes than the approaches of Thurin et al. while still achieving superior worst-slab coverage (see table above).
>
> ### log-d-volumes of the predicted region
>
> | Method | Coverage : 0.9 | 0.8 | 0.7 |
> |---|---:|---:|---:|
> | [Thurin] OT-CP-Global | 1.537 | 1.426 | 1.284 |
> | [Thurin] OT-CP-Local  | -0.759 | -0.778 | -0.833 |
> | [Our method] PB       | -1.952 | -2.848 | -3.108 |
> | [Our method] RPB  | 1.534 | 1.430 | 1.268 |

---

> ### Author Response · Authors · 2025-11-18
> **Clarity**
>
> Regarding Tables 1–2 and Figure 2, the specific method we use to construct conditional quantiles has not, to our knowledge, been explored in previous work. Our approach can be seen as a form of amortized optimal transport and is novel. We argue that accurate vector quantile construction is essential for obtaining reliable conformal regions with small volume.
>
> We also include experiments that assess the generative capabilities of our method compared to existing solutions for conditional quantiles and conditional optimal transport.
>
> The experiments yield two main takeaways. First, on the simple synthetic datasets in Table 1, our method outperforms competing constructions of conditional quantile functions that rely on discrete OT and assume linearity of the VQR as function of (learned) feature. Second, Table 2 omits discrete quantiles because they do not provide an inverse operator – that is, a map from $\(Y\mid X\)$ back to $\(U\)$.
> Figure 2 provides a complementary comparison: our approach outperforms previous methods for constructing conditional quantiles, and we also examine an entropy-regularized variant inspired by Sinkhorn iterations. Implementing this requires estimating the gradient of a log-integral through sampling; the naïve estimator is biased, and Figure 2 shows that this naïve entropic solver performs worse, while the Amos-style amortized solution remains stable and accurate.
>
> Overall, Tables 1–2 and Figure 1 are necessary to demonstrate that the proposed quantile construction produces accurate quantile sets and positions our method relative to competitive generative baselines.
>
> Clarification for Figure 1: it reports log Wasserstein distances; negative values indicate very small distances. We have updated the text to make this explicit.

---

> > ### Author Response · Authors · 2025-11-18
> > **Pareto front**
> >
> > We appreciate the reviewer’s comment about analyzing the Pareto front. We agree that this is an interesting direction and intend to include these results. However, running the necessary simulations will take some time, so we will provide the results later (before of the end of the discussion).

---

### Meta-Review · Area_Chair_rbjY · 2026-01-11

**Summary:**

This paper considers the problem of conformal prediction based uncertainty quantification for multivariate output setting by combining neural optimal transport and conditional vector quantile regression. The approach employs convex neural networks to model transport maps and uses the resulting quantile maps to build uncertainty sets. Experiments are performed on both synthetic and real-world datasets to demonstrate its efficacy.

The reviewers' acknowledged the contribution that brings together existing ideas to solve this problem, but also raised some critical concerns:
1. Positioning of the work in the context of current state of knowledge (for this problem setting and beyond)
2. Novelty given that it combines many existing ideas
3. Validity of assumptions behind Theorem 3 in practice and proof of Theorem 3.
4. Empirical comparisons and some clarification questions

The author rebuttal addressed concerns #3 and #4 mostly satisfactorily and somewhat satisfactorily for #1 and #2. The AC did search for the related work and came across one paper that is relevant:
ICLR-2025 paper https://openreview.net/forum?id=Nfd7z9d6Bb

The paper is a borderline (mostly combines existing ideas but executed well, strong tone for claims as pointed out by some reviewers', and positioning of the paper w.r.t state of the art which is improved but still has some room). If the paper is accepted, I strongly encourage the authors' to revise the paper to address these issues in the final version.

**Reviewer Concerns:**

1. Positioning of the work in the context of current state of knowledge (for this problem setting and beyond)
2. Novelty given that it combines many existing ideas
3. Validity of assumptions behind Theorem 3 in practice and proof of Theorem 3.
4. Empirical comparisons and some clarification questions

The author rebuttal addressed concerns #3 and #4 mostly satisfactorily and somewhat satisfactorily for #1 and #2. The AC did search for the related work and came across one paper that is relevant:
ICLR-2025 paper https://openreview.net/forum?id=Nfd7z9d6Bb

**Reviewer Scores:**

XoMS: 6
Vkyx: 4 => 6
Rw7A: 2 => 4
6k36: 6

---

### Decision · Program_Chairs · 2026-01-26

Accept (Poster)